# Adversarial Subspace Generation for Outlier Detection in High-Dimensional Data

Jose Cribeiro-Ramallo                                    *jose.cribeiro@kit.edu*
*Karlsruhe Institute of Technology*

Federico Matteucci                                       *federico.matteucci@kit.edu*
*Karlsruhe Institute of Technology*

Paul Enciu                                               *paul.enciu@student.kit.edu*
*Karlsruhe Institute of Technology*

Alexander Jenke                                          *alexander.jenke@student.kit.edu*
*Karlsruhe Institute of Technology*

Vadim Arzamasov                                          *vadim.arzamasov@kit.edu*
*Karlsruhe Institute of Technology*

Thorsten Strufe                                          *thorsten.strufe@kit.edu*
*Karlsruhe Institute of Technology*

Klemens Böhm                                             *klemens.boehm@kit.edu*
*Karlsruhe Institute of Technology*

**Reviewed on OpenReview:** *https://openreview.net/forum?id=k7QsjiRE17*

## Abstract

Outlier detection in high-dimensional tabular data is challenging since data is often distributed across multiple lower-dimensional subspaces — a phenomenon known as the Multiple Views effect (MV). This effect led to a large body of research focused on mining such subspaces, known as *subspace selection*. However, as the precise nature of the MV effect was not well understood, traditional methods had to rely on heuristic-driven search schemes that struggle to accurately capture the true structure of the data. Properly identifying these subspaces is critical for unsupervised tasks such as outlier detection or clustering, where misrepresenting the underlying data structure can hinder the performance. We introduce Myopic Subspace Theory (MST), a new theoretical framework that mathematically formulates the Multiple Views effect and writes subspace selection as a stochastic optimization problem. Based on MST, we introduce **V**-GAN, a generative method trained to solve such an optimization problem. This approach avoids any exhaustive search over the feature space while ensuring that the intrinsic data structure is preserved. Experiments on 42 real-world datasets show that using **V**-GAN subspaces to build ensemble methods leads to a significant increase in one-class classification performance — compared to existing subspace selection, feature selection, and embedding methods. Further experiments on synthetic data show that **V**-GAN identifies subspaces more accurately while scaling better than other relevant subspace selection methods. These results confirm the theoretical guarantees of our approach and also highlight its practical viability in high-dimensional settings.

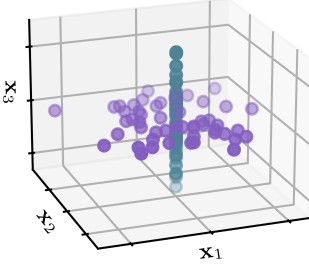 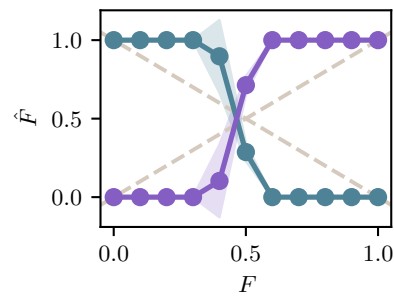

(a) Population from example 1. (b) SotA for subspace search

Figure 1: **(a)** Population from example 1 and the performance of the SotA for subspace search in it. We colored in blue those in subspace $U_1$ and purple those in subspace $U_2$. **(b)** The normalized weights $\hat{F}$ and $1 - \hat{F}$ assigned by GMD (Trittenbach and Böhm, 2019) to subspaces $S_1$ and $S_2$ should be as close as possible to $F$ and $1 - F$, i.e., the dashed grey lines.

# 1 Introduction

High-dimensional data, such as images, text, or some tabular datasets, constitutes much of the available data on the internet, in medical domains, and even in the private sector. Especially when high-dimensional data can exhibit multiple complex relations between its features. Outlier Detection (OD), as well as other downstream tasks, can greatly benefit from correctly exploiting these relations to achieve more accurate results (Aggarwal, 2017; Trittenbach and Böhm, 2019). A popular research direction in the literature is to search for subspaces maximizing a given quality metric. The multiple subspaces are later employed either to study complex interactions between features or to build an ensemble of models, with each member in a different subspace (Aggarwal, 2017).

Methods for obtaining subspaces work in two ways. One type extracts a single subspace that better represents all data — like embedding and feature selection methods (Balın et al., 2019; Meilă and Zhang, 2023; Healy and McInnes, 2024). These methods assume that the data lies on a single, low-dimensional subspace preserving its properties, such as point distances, topology, or notably, the underlying distribution. As discussed in Example 1, however, a single, low-dimensional subspace might not be enough to characterize the data. It is therefore common in unsupervised tasks to assume that the data instead lies on multiple subspaces — known as the *Multiple Views effect (MV)* (Keller et al., 2012). Subspace selection methods handle this latter scenario by providing a list of interesting subspaces and, hence, better preserving the relationships within the data (Keller et al., 2012; Trittenbach and Böhm, 2019; Agrawal et al., 2005).

The extra information provided by mining for multiple subspaces can be crucial for unsupervised tasks with little prior information, like clustering and outlier detection (Keller et al., 2012; Qu et al., 2023; Cribeiro-Ramallo et al., 2024). Particularly, one can create ensembles of outlier detection methods, like LOF (Breunig et al., 2000), by training multiple detectors, each on a lower-dimensional projection of the data. Subspace selection methods that provide projections into each subspace are known as *subspace search methods*. The most common approach, as discussed in (Aggarwal, 2017, Chapter 4), is to search for those feature subspaces maximizing some heuristic quality metric. The obtained subspaces are hence necessarily axis-parallel. Despite this limitation, this approach proved to be an effective technique for some downstream tasks, including outlier detection. The usage of a heuristic as the quality metric, however, does not guarantee that the extracted subspaces preserve the data's properties, such as its distribution. As we elaborate in the following example, this can happen even in simple settings.

**Example 1.** *Consider a population as in Figure 1a, where 3-dimensional data lies with probability $F$ on the $x_1$-$x_2$ plane and with probability $1 - F$ on the $x_3$ line. We refer to these two subspaces as $S_1$ and $S_2$*

Table 1: Summary of existing methods and their capabilities.

|  | Multi-subspace | Represent | Project |
|---|---|---|---|
| Feature Selection | ✗ | ✓ | ✓ |
| Embedding Methods | ✗ | ✓ | ✓ |
| Subspace Search | ✓ | ✗ | ✓ |
| Subspace Discovery | ✓ | ✓ | ✗ |
| **Subspace Generation** (Ours) | ✓ | ✓ | ✓ |

*respectively. The data exhibits Multiple Views, as it lies within $S = S_1 \cup S_2 \subset \mathbb{R}^3$. As the data lies on two subspaces, methods that return a single subspace are not able to correctly represent it. On the other hand, methods that return a list of subspaces should be able to (1) identify $S_1$ and $S_2$ as the relevant subspaces and (2) assign them scores proportional to $F$ and $1 - F$. However, as shown in 1b, the state-of-the-art (GMD (Trittenbach and Böhm, 2019)) fails to identify both subspaces and to assign them an accurate score. Indeed, the retrieved subspaces approximately degenerate to the sole $S_1$ if $F < 0.5$ and to $S_2$ otherwise.*

Our goal is to avoid using a heuristic quality metric while guaranteeing that the underlying data distribution is preserved. This presents the following challenges. (1) To the best of our knowledge, the only previous attempt at formalizing the Multiple Views effect focused on proving the efficacy of subspace ensembles for tabular Outlier Detection (OD) (Cribeiro-Ramallo et al., 2024). Consequently, their theory is not directly usable to find subspaces, nor does it extend to arbitrary data types; we will elaborate on this in Section 3.1. (2) Even with a theory able to recognize the subspaces relevant for MV, these latter live in an exponential search space — the power set of the set of features. An exhaustive search is therefore unfeasible. The designed method should be able to find relevant subspaces and approximate their weights while avoiding searching in such a power set.

Our contributions are the following: (1) We generalize the theory from Cribeiro-Ramallo et al. to be both applicable to general data types and to allow us to obtain subspaces in practice. In our revised theory, subspaces can be obtained by solving a stochastic optimization problem, while we can provide guarantees on the underlying data distribution being preserved. (2) To solve the optimization problem, we propose the generative network **V**-GAN, whose goal is to generating projections into the desired subspaces. We prove that its loss function is optimizable and that the network converges to the desired global optimum under mild conditions. (3) We validate our theoretical results in practice using synthetic data, showing how **V**-GAN can extract the predicted subspaces by our theory. (4) We study the quality of **V**-GAN subspaces by training an ensemble of outlier detection methods using them, and testing their performance. In particular, we show how **V**-GAN's subspaces consistently lead to ensembles that are significantly better than all their competitors on 42 real-world benchmark datasets from (Han et al., 2022). (5) Finally, we provide the code for all of our experiments and methods[1].

## 2 Related Work

This section briefly overviews the subspace selection field together with its subfields. Table 1 includes a summary of the discussion.

A classic approach to dealing with high-dimensional data is to assume that data lies on a lower-dimensional manifold and to provide a projection into it. This can be done by removing unwanted features (Balın et al., 2019) or finding other (not necessarily) orthogonal transformations (Jones and Artemiou, 2021; Meilă and Zhang, 2023). These transformations always focus on preserving the original distribution of the data in order to use it in a given downstream task. For unsupervised downstream tasks, it is common to use a more general assumption on the data, known as *Multiple Views effect* (MV). Under MV, data lies in a collection of subspaces $\{S_i\}_{i=1}^{n_s}$, rather than in a single well-behaved one (Keller et al., 2012; Elhamifar and Vidal, 2013). Subspace selection methods aim to obtain a set of projections $\{\pi_i\}_{i=1}^{n_s}$ into each subspace, ensuring that any new datapoint can be projected into these subspaces (Aggarwal, 2017; Keller et al., 2012). The

---

[1]https://github.com/jcribeiro98/V-GAN

number of subspaces, $n_s$, is assumed unknown beforehand (Agrawal et al., 2005), in contrast to other classical approaches employed in subspace clustering (Wang et al., 2009; Vidal et al., 2012). Using these subspaces, one can build powerful ensembles for outlier detection (Trittenbach and Böhm, 2019) or obtain more precise clusters (Qu et al., 2023). There exist two big families of methods in subspace selection, as we will elaborate in what follows.

**Subspace Search.** Subspace search methods assume that the subspaces conforming the data are feature subspaces, i.e., axis-parallel projections of the data. Thanks to this, they work on a finite search space $\mathcal{P}(\{1, ..., d\})$ simplifying the search scheme. More specifically, these methods explicitly output the subspaces that maximize a certain quality metric in $\mathcal{P}(\{1, ..., d\})$. As the subspaces are explicitly known, one can trivially *project* the data into each subspace for the desired downstream task using axis-parallel projections. They are popular for outlier detection, as they allow the use of subspaces to create ensembles with off-the-shelf outlier detectors (Aggarwal, 2017). These methods are also used for clustering, like in (Agrawal et al., 2005; Cheng et al., 1999), although they are typically coupled with a particular clustering algorithm. However, as discussed earlier, the main drawbacks of these methods are the cardinality of $\mathcal{P}(1, ..., d)$ and the selection of a quality metric. Although some efforts address the search space problem, no work in the subspace search literature offers a theoretical definition of what an 'important' subspace is. Current methods rely on heuristic quality metrics that do not guarantee the selected subspaces accurately preserve the data's properties. For instance, a method like *Enclus* (Cheng et al., 1999) relies on subspace search based on the entropy metric. Therefore, *Enclus* would select the subspaces with minimal entropy, missing potential subspaces if the assumption of low entropy subspaces does not hold — like for the population from Example 1. This leads to a collection of subspaces that do not *represent* the data correctly.

**Subspace Discovery.** Given the problems with subspace search, a second group of methods adopted a different approach for subspace selection. That is, they do not assume that the subspaces are necessarily feature subspaces, nor do they explicitly output the projections themselves. Instead, they solely focus on identifying which data points are likely to belong to the same subspace to output an adjacency matrix based on it. This relationship matrix is then typically used as a graph adjacency matrix for spectral clustering. Authors successfully used these methods to develop clustering techniques for various applications, including face, motion, and sentiment recognition (Qu et al., 2023; Elhamifar and Vidal, 2013). However, since they rely on a relationship matrix between the clustered points, these methods focus on projecting the clustered data only. Therefore, they cannot *project* new data points into the subspaces without retraining, which limits the use of subspace discovery methods for outlier detection. While there are adaptations to handle incoming data points, they typically treat data in an online manner, continuously updating the projections with each new datapoint inferred (Kong et al., 2011; Langone et al., 2014; Sui et al., 2022). Consequently, this adaptation falls under the category of online machine learning. As we treat data in an offline manner, this adaptation is outside of our scope (Ben-David et al., 1997).

**Previous Descriptions of Multiple Views.** In the recent literature on outlier detection, Cribeiro-Ramallo et al. attempted to mathematically describe the MV effect for a given population. Previous attempts aim to study the distribution of data lying in subspaces by characterizing parts of the fullspace where the probability measure concentrates (Pal et al., 2023). In contrast, Cribeiro-Ramallo et al. focused on providing specific probability measures for the subspaces, allowing both their characterization and measurement. In particular, the authors defined a family of distributions called *myopic distributions*, and show how the MV effect occurs when the data is generated as such. For example, a distribution of a population $\mathbf{x}$ is *myopic* when its density $P_{\mathbf{x}}$ is invariant under the transformations of a random orthogonal projection matrix $\mathbf{U}$. That is, whenever $P_{\mathbf{x}} = P_{\mathbf{U}\mathbf{x}}$, with each realization of $\mathbf{U}$ being an orthogonal projection matrix (Cribeiro-Ramallo et al., 2024). As an example, if one considers again the population from Example 1, one can easily verify that it is myopic under the effects of:

$$\mathbf{U} = \begin{cases} U_1 = diag(1, 1, 0) & \text{with probability } F, \\ U_2 = diag(0, 0, 1) & \text{else.} \end{cases}$$

While the theory can predict certain behavior on paper, Cribeiro-Ramallo et al. do not provide any way to obtain such $\mathbf{U}'s$ in practice. Additionally, it lacks sufficient generality to do so trivially, as density functions

are difficult to estimate in practice (Guo et al., 2022). This complicates the task of obtaining the random projection matrix by directly using the definition in Cribeiro-Ramallo et al. (2024).

**Subspace Generation.** Our work centers around a generalization of the definition of myopic distribution that allows us to frame it using components one can easily estimate in practice. Thanks to this, we can fit a generative method capable of approximating the distribution of a **U** verifying the definition. This way, we can sample projections into these subspaces This novel approach to subspace selection, which we dubbed **Subspace Generation**, avoids searching in $\mathcal{P}(\{1, ..., d\})$, and provides a suitable notion of "important" subspace. This solves the *representation* problem of subspace search, while not sacrificing the ability to *project* the data into the subspaces.

## 3 Myopic Subspace Theory

In this section, we will discuss the preliminaries for introducing our Subspace Generation method. We will frame our theoretical background in a generalization of the theory of myopic distributions introduced by Cribeiro-Ramallo et al.. In particular, we will introduce their definition first and discuss the main drawbacks that motivate a more general framework (Section 3.1). After that, we will propose such a generalization and use it to write subspace selection as an optimization problem (Section 3.2). Lastly, we will show optimality guarantees under general conditions (Section 3.3).

### 3.1 Original Definition

A large collection of authors observed that high-dimensional tabular data seems to behave differently in certain feature subspaces than in others. In particular, a significant body of research empirically examines the occurrence of data variability concentrating on a specific collection of subspaces (Keller et al., 2012; Nguyen et al., 2014; Trittenbach and Böhm, 2019). This effect is called *Multiple Views* of the data (MV) (Müller et al., 2012). Cribeiro-Ramallo et al. tried to mathematically describe MV, to then propose a way to train parametric methods under it. Their definition goes as follows.

First, consider a metric space $(E^d, \mathcal{T})$. Let $\mathbf{x}^\nu : (\Omega, \mathcal{B}_\sigma, \mathbb{P}) \longrightarrow E^d$ be a random vector and $(\Omega, \mathcal{B}_\sigma, \mathbb{P})$ a measurable probability space with Borel's sigma algebra. Lastly, consider $Diag(\{0,1\})_{d \times d}$ the space of $d \times d$ diagonal binary matrices[2], and $\mathbb{P}_{\mathbf{x}^\nu} = \mathbf{x}^\nu_* \mathbb{P}$ the distribution of $\mathbf{x}^\nu$ with $P_{\mathbf{x}^\nu} = \frac{d\mathbb{P}_{\mathbf{x}^\nu}}{d\mu}$ its density in the Radon-Nikodym sense[3].

**Definition 1** (Cribeiro-Ramallo et al.)**.** *Consider $E = \mathbb{R}$ and $\mathbf{U} : (\Omega, \mathcal{B}_\sigma, \mathbb{P}) \longrightarrow Diag(\{0,1\})_{d \times d}$ a random binary matrix. We will say that $\mathbf{x}^\nu$ is* myopic under the views of $\mathbf{U}$ *iff:*

$$P_{\mathbf{x}^\nu} = P_{\mathbf{U}\mathbf{x}^\nu}, \text{ point-wise in their support.}$$

*In this case, we call $\mathbf{x}^\nu$* myopic under $\mathbf{U}$ *or simply,* myopic*, if there is no risk of confusion.*

By Definition 1, a population is myopic under the views of a random binary matrix iff the random vector $\mathbf{U}\mathbf{x}^\nu$ has the same density as $\mathbf{x}^\nu$ for all points in its support. As diagonal matrices are orthogonal projections, it is the same as saying that observing $\mathbf{x}^\nu$ and a randomly projected version of $\mathbf{x}^\nu$ lead to the same density for any point in its support. The authors then prove that one can calculate $P_{\mathbf{U}\mathbf{x}^\nu}$ under myopicity by

$$P_{\mathbf{U}\mathbf{x}^\nu} = \sum_{i=1}^{N} \mathbb{P}_{\mathbf{U}_i}(U_i) P_{U_i \mathbf{x}^\nu},$$

This result is very important for the particular use case of outlier detection (Cribeiro-Ramallo et al., 2024; Trittenbach and Böhm, 2019; Aggarwal, 2017). However, how to find such **U** is not properly described. In particular, we identify the following problems with Definition 1.

---

[2]Without the identity.
[3]With $\mu \gg \mathbb{P}_{\mathbf{x}^\nu}$.

1. **The point-wise equality of densities**. In order to provide an estimate for the validity of the definition, one would have to estimate first both densities. Not only is density estimation hard in high-dimensional data, but the existence of densities is not guaranteed for a general distribution, limiting the applicability.
2. **Limited to $E = \mathbb{R}$ and $U \in Diag(\{0, 1\})_{d \times d}$**. The limitation of the metric space $E$ to the real line and the realizations of $\mathbf{U}$ to diagonal binary matrices further restricts the use of this theory to more general data types.
3. **Estimation in Practice**. Even with all the limitations to the definition of $\mathbf{x}^\nu$ and $\mathbf{U}$, it is unclear how to properly find a $\mathbf{U}^*$ that verifies Definition 1 for a given $\mathbf{x}^\nu$. Even assuming that we can perfectly estimate the densities, how to find such a random matrix that $|P_{\mathbf{x}^\nu}(p) - P_{\mathbf{U}\mathbf{x}^\nu}(p)| = 0$ for almost all $p$ is unclear for the finite sample setting.

In the following section, we will propose a general definition that addresses all previous weaknesses, while also giving certain generality conditions for it.

### 3.2 Myopicity via its Representation in $\mathcal{H}$

We will first introduce a collection of notations and necessary conditions for our generalized definition. After that, we will explain how our generalization solves all of the previously raised problems.

#### 3.2.1 Tackling the Point-wise Equality of Densities and the Space Limitations

Consider $(E, \mathcal{T})$ a separable metric space, $\mathcal{H}$ the associated Reproducing Kernel Hilbert Space (RKHS) of real-valued functions on $E$ with kernel $\kappa$ and $\mathcal{M}_1^+(E) \subset \mathcal{M}^+(E)$ the space of positive signed Borel measures with value 1 (i.e., probability measures) on $E$. Further consider $\mathbf{x} : (\Omega, \mathcal{B}_\sigma, \mathbb{P}) \longrightarrow E$ a random variable with $(\Omega, \mathcal{B}_\sigma, \mathbb{P})$ a measurable probability space with Borel's sigma-algebra, and $\mathfrak{X}$ the space of such random variables. In order to avoid problems 1 and 2, one can consider a richer definition as follows:

**Definition 2** (Myopicity of a distribution). *Consider $\mathcal{C}(\mathfrak{X})$ the class of continuous operators from and to the space of random variables on $E$, $\mathfrak{X}$, and a subset $\Theta(\mathfrak{X}) \subset \mathcal{C}(\mathfrak{X})$. Further consider*

$$\mathbf{U} : (\Omega, \mathcal{B}_\sigma, \mathbb{P}) \longrightarrow \Theta(\mathfrak{X}) \subset \mathcal{C}(\mathfrak{X}),$$

*a random operator taking values on $\Theta(\mathfrak{X})$. We say that $\mathbf{x}$ is myopic to the views of $\mathbf{U}$ iff*

$$\mathbb{P}_{\mathbf{x}} = \mathbb{P}_{\mathbf{U}\mathbf{x}}. \tag{1}$$

*In this case, we say that $\mathbf{x}$ is $\Theta(\mathfrak{X})$-myopic and $\mathbf{U}$ is a lens operator for $\mathbf{x}$.*

It is clear that Definition 2 generalizes Definition 1, by taking $E = \mathbb{R}^d$, $\Theta(\mathfrak{X}) = Diag(\{0, 1\})_{d \times d}$ and invoking the uniqueness of Radon-Nikodym's derivative (Simonnet, 1996, Chapter 10). Furthermore, $\mathbf{U}\mathbf{x}$ is correctly defined as the mapping

$$\mathbf{U}\mathbf{x} : \omega \in (\Omega, \mathcal{B}_\sigma, \mathbb{P}) \longmapsto Ux \in E,$$

with both $U$ and $x$ being realizations of $\mathbf{U}$ and $\mathbf{x}$ respectively. Coming next, we will introduce a collection of examples of lens operators for a different set of cases — not necessarily having $\Theta(\mathfrak{X})$ defined on fibers of $E$.

**Example 2.** *i) **Normal Projected Population from Example 1.** Consider the same population as Example 1, with $F = 0.5$. Clearly, by the law of total probabilities*

$$\mathbb{P}_{\mathbf{x}} = \frac{1}{2}\mathbb{P}_{\mathbf{x}|\mathbf{x} \in S_1} + \frac{1}{2}\mathbb{P}_{\mathbf{x}|\mathbf{x} \in S_2}.$$

*In this case, any non-zero measurable set on $S_1$ will get projected into a zero-measure set in $S_2$, and vice versa. Thus, we can write $\mathbb{P}_{\mathbf{x}|\mathbf{x} \in S_1} = \mathbb{P}_{U_1\mathbf{x}}$ and $\mathbb{P}_{\mathbf{x}|\mathbf{x} \in S_2} = \mathbb{P}_{U_2\mathbf{x}}$, with*

$$U_1 = \begin{pmatrix} 1 & 0 & 0 \\ 0 & 1 & 0 \\ 0 & 0 & 0 \end{pmatrix}, \ U_2 = \begin{pmatrix} 0 & 0 & 0 \\ 0 & 0 & 0 \\ 0 & 0 & 1 \end{pmatrix},$$

*and observing that both $\mathbb{P}_{U_1\mathbf{x}}$ and $\mathbb{P}_{U_2\mathbf{x}}$ are probability measures by invoking the Disintegration Theorem (Faden, 1985).*

*ii)* **General non-parallel hyperplanes.** *Consider now a population lying in $H_1$ and $H_2$, two different non-parallel hyperplanes of $E = \mathbb{R}^n$ of (non-necessarily equal) finite dimensionality, with probability $F$ and $1 - F$. Further consider $E$ to be equipped with the usual topology. As an illustrative example, one can consider the simplified version of Figure 2a. Consider $E$ to be a normed topological vector space and both $H_1$ and $H_2$ subspaces. Now, consider the projections $U_1$ and $U_2$ to $H_1$ and $H_2$, parallel to $H_2$ and $H_1$ respectively. In other words, $U_1$ and $U_2$ are normal to the orthogonal planes $H_2^{\perp}$ and $H_1^{\perp}$, respectively. Clearly, $U_2(H_1) \subsetneq H_2$, otherwise any point of $H_2$ could be written as a linear combination of $H_1$'s basis. This would, in turn, lead to a contradiction as $H_1$ and $H_2$ are non-parallel, and thus $H_2 \not\subset H_1$. Therefore, $U_2(H_1)$ is a subspace of $H_2$ with lower dimension. The equivalent can be derived for $U_1(H_2)$.*

*Consider $\mathbf{U}$ as the random operator such that it takes values $U_1$ and $U_2$ with probabilities $F$ and $1 - F$ respectively. Clearly, $\mathbb{P}_{\mathbf{x}} = F\,\mathbb{P}_{\mathbf{x}|\mathbf{x}\in H_1} + (1 - F)\,\mathbb{P}_{\mathbf{x}|\mathbf{x}\in H_2}$ as in the previous example. Furthermore, as we just proved, any open set from $H_1$ will get projected into a subspace in $H_2$, and vice versa. As $\mathcal{F}$ is a Borel's sigma algebra and $E = \mathbb{R}^n$, $U_2(H_1)$ is of measure 0 in $H_2$ (Rudin, 1987, Theorem 2.20(e)). Thus, all measurable subsets of $H_1$ will get projected into zero-measure subsets in $H_2$, and vice versa. Then, as we did before, $\mathbb{P}_{\mathbf{x}|\mathbf{x}\in H_1} = \mathbb{P}_{U_1\mathbf{x}}$ and $\mathbb{P}_{\mathbf{x}|\mathbf{x}\in H_2} = \mathbb{P}_{U_2\mathbf{x}}$. Therefore,*

$$\mathbb{P}_{\mathbf{x}} = F\,\mathbb{P}_{U_1\mathbf{x}} + (1 - F)\,\mathbb{P}_{U_2\mathbf{x}},$$

*making $\mathbf{U}$ a lens operator for $\mathbf{x}$. This process can be repeated to obtain the lens operator to an arbitrary finite collection of non-parallel hyperplanes $\{H_i\}_{i\in I}$ with probabilities $\{p_i\}_{i\in I}$ with linearly independent bases.*

*iii)* **Transversal manifolds.** *We can take it one step further and consider the general case of two $(n-1)$-submanifolds $\mathcal{M}_1, \mathcal{M}_2$ of the ambient $n$-manifold $\mathcal{M}$. If the intersection of both submanifolds is non-empty, we can use the notion of transversality to prove that the intersection is a 0-measure set. For instance, we say that two manifolds intersect transversely iff the tangent spaces $T_p\mathcal{M}_1$, $T_p\mathcal{M}_2$ span $T_p\mathcal{M}$ at all points $p$ (Guillemin and Pollack, 2010). In this case, the intersection $\mathcal{M}_1 \cap \mathcal{M}_2$ is a submanifold with dimensionality $\dim\mathcal{M}_1 + \dim\mathcal{M}_2 - \dim\mathcal{M} = n - 2$. If one takes a look at Figure 2b we can see an example in 3 dimensions, where the intersection of these two 2-manifolds — 2-spheres — is of dimensionality 1. Consider $U_1$ the parallel transport to the closest point to $\mathcal{M}_1$ and $U_2$ the parallel transport to the closest point in $\mathcal{M}_2$. Thus, by taking $U_1$ and $U_2$, we can construct a similar lens operator for any population $\mathbf{x}$ lying in the transversal manifolds.*

*iv)* **Homoskedastic errors.** *Assume the following random variable $\mathbf{y} = \beta\mathbf{x} + \varepsilon$, with $\beta \in \mathbb{R}^{d\times d}$ and $\varepsilon$ another random variable acting as noise. Now, given an infinite set $D = \{\beta x_i + \varepsilon_i\}_{i\in I}$ of samples of $\mathbf{y}$, we can define*

$$U_i : \beta x_j + \varepsilon_j \in \mathbb{R}^d \longmapsto \beta x_j + \varepsilon_i \in \mathbb{R}^d,\ \forall i \in I.$$

*As such, defining a $\mathbf{U}$ selecting all $U_i$ with equal probability will trivially be a lens operator, if all $\varepsilon_i$ are equally distributed. The finite sample setting of this case is a common bootstrap technique known as* Resampling Residuals.

*v)* **Location Operator for the Variance.** *A trivial example in $\mathbb{R}$ can be obtained by considering the random variables $\mathbf{x} \sim \mathbb{P}_{\mathbf{x}}$ and $\mathbf{y} = Var(\mathbf{x})$. Consider $\mathbf{U}(\omega) \in \{U_1, U_2\}$, with $U_1\mathbf{y} = Var(\mathbf{x} + 3)$ and $U_2\mathbf{y} = Var(\mathbf{x} + 10)$. Trivially, as the variance is invariant to location changes, $\mathbf{U}\mathbf{y} = \mathbf{y}$.*

Certainly, both problems 1 and 2 are successfully addressed by Definition 2. However, equality between two measures in $\mathcal{M}_1^+$ is still too general to tackle problem 3. Generally, when searching for a way to determine when two elements $a, b$ of the same space $X$ are equal, one defaults to check whether $m(a, b)_X = 0$, if such a space $X$ is equipped with a metric $m(\cdot, \cdot)_X$. Our goal is to do the same for two probability measures $\mathfrak{p}, \mathfrak{q} \in \mathcal{M}_1^+(E)$. In what follows, we will introduce how to obtain such a metric, and in what conditions that metric exists.

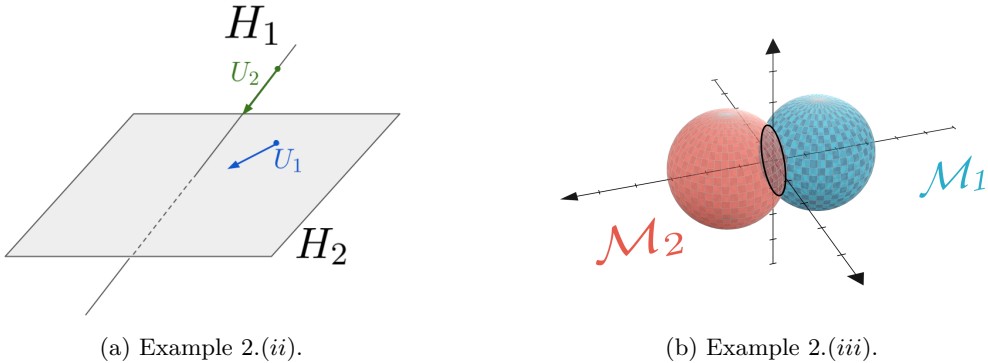

(a) Example 2.(*ii*).

(b) Example 2.(*iii*).

Figure 2: Simplified version of the subspaces in Example 2.(*ii*) and Example 2.(*iii*).

### 3.2.2   Tackling the Estimation

There exists a large body of literature focusing on embedding $\mathcal{M}_1^+(E) \subset \mathcal{M}^+(E)$ into an RKHS $\mathcal{H}$ of real-valued functions on $E$ — see (Berlinet and Thomas-Agnan, 2004, Chapter 4) for a survey. The particular embedding employed to represent a measure as a function in $\mathcal{H}$ will determine the metric that one obtains at the end. This is why it is important to carefully embed $\mathcal{M}^+$ in a way that the resulting metric can be easily estimated. For that, we will follow the existing body of work that aims to obtain a metric (Gretton et al., 2012; Schrab et al., 2023; Fukumizu et al., 2007) that is easy to estimate and has a known asymptotic distribution (Gretton et al., 2012).

First, consider the linear functional on $\mathcal{H}$:

$$\mathbb{E}_{\mathfrak{p}} : f \in \mathcal{H} \longmapsto \mathbb{E}_{\mathfrak{p}} f = \int f d\mathfrak{p} \in \mathbb{R}.$$

Given this mapping, one can define:

**Definition 3** (Definition 2 in Gretton et al. (2012)). *Let $\mathcal{F} \subset \mathcal{H}$ be a class of functionals on $E$. The Maximum Mean Discrepancy (*MMD*) is defined as:*

$$\mathrm{MMD}_\kappa(\mathfrak{p}, \mathfrak{q}) = \sup_{f \in \mathcal{F}} \left( \mathbb{E}_{\mathfrak{p}} f - \mathbb{E}_{\mathfrak{q}} f \right). \tag{2}$$

In $\mathcal{H}$ an RKHS with kernel $\kappa$ measurable and such that[4] $\int \sqrt{\kappa(\cdot, \cdot)} d\mathfrak{p} < \infty$, for all $\mathfrak{p} \in \mathcal{M}_1^+(E)$, and $\mathcal{F}$ the unit ball in $\mathcal{H}$, one can easily prove (Sriperumbudur et al., 2010) that:

$$\exists! \mu_{\mathfrak{p}} \in \mathcal{H} \text{ such that } \mathbb{E}_{\mathfrak{p}} f = < \mu_{\mathfrak{p}}, f >_{\mathcal{H}}, \ \forall \mathfrak{p} \in \mathcal{M}_1^+, \tag{3}$$

$$\mathrm{MMD}_\kappa(\mathfrak{p}, \mathfrak{q})^2 = \|\mu_{\mathfrak{p}} - \mu_{\mathfrak{q}}\|_{\mathcal{H}}^2 = \mathbb{E}_{\mathbf{x}, \mathbf{x}' \sim \mathfrak{p}} \left[ \kappa(\mathbf{x}, \mathbf{x}') \right] - 2\mathbb{E}_{\mathbf{x} \sim \mathfrak{p}, \mathbf{x}' \sim \mathfrak{q}} \left[ \kappa(\mathbf{x}, \mathbf{x}') \right] + \mathbb{E}_{\mathbf{x}, \mathbf{x}' \sim \mathfrak{q}} \left[ \kappa(\mathbf{x}, \mathbf{x}') \right]. \tag{4}$$

The unique representer of $\mathfrak{p}$ in $\mathcal{H}$, $\mu_{\mathfrak{p}}$, is known as the *mean embedding* of $\mathfrak{p}$ in $\mathcal{H}$. The use of the unit ball for $\mathcal{F}$ is not arbitrary, as different function classes lead to different metrics. We want to use the one in Equation 4 as it has a consistent[5] $U$-estimator with better rate of convergence than other popular metrics in $\mathcal{M}_1^+$ — see Sriperumbudur et al. (2010). An interesting consequence of working in an RKHS is that one can characterize specific properties of the MMD and the estimator by properties of the kernel $\kappa$. The most important one for us is that the MMD defined as in equation 4, on an RKHS $\mathcal{H}$ with a characteristic[6] kernel is a metric —see Fukumizu et al. (2007). Thus, $\mathrm{MMD}_\kappa(\mathfrak{p}, \mathfrak{q}) = 0 \iff \mathfrak{p} = \mathfrak{q}$.

Therefore, one can state the following:

---

[4]here the kernel is integrated with respect to both variables at the same time. This is, as:

$$\kappa(\cdot, \cdot) : x \in E \longmapsto \kappa(x, x)$$

[5]More precisely, $\sqrt{mn/(m+n)}-$consistent

[6]$\kappa$ is characteristic iff $\mathbb{E}_{\mathfrak{p}} f = \mathbb{E}_{\mathfrak{q}} f, \ \forall f \implies \mathfrak{p} = \mathfrak{q}$.

**Lemma 1.** *Consider $\mathcal{H}$ a RKHS with a characteristic kernel $\kappa$; and $\mathbf{x}$, $\mathbf{U}$ and MMD as previously defined. Further, consider $\mathbf{V}$ to be a lens operator for $\mathbf{x}$. Then,*

$$\arg\min_{\mathbf{U}} \mathrm{MMD}_\kappa(\mathbb{P}_\mathbf{x}, \mathbb{P}_{\mathbf{Ux}}) \ni \mathbf{V}$$

Thus, by Lemma 1, for a $\Theta(\mathfrak{X})$-myopic $\mathbf{x}$, one could find a lens operator $\mathbf{V}$ by solving the stochastic optimization problem

$$\arg\min_{\mathbf{U}} \mathrm{MMD}_\kappa(\mathbb{P}_\mathbf{x}, \mathbb{P}_{\mathbf{Ux}}) \ni \mathbf{V}.$$

As we only have access to the sample estimate of the MMD in the finite sample setting, $\widehat{\mathrm{MMD}}_\kappa$ (Gretton et al., 2012), we need to work with the problem

$$\arg\min_{\mathbf{U}} \widehat{\mathrm{MMD}}_\kappa^2(\mathbb{P}_\mathbf{x}, \mathbb{P}_{\mathbf{Ux}}) \ni \mathbf{V}. \tag{5}$$

The question now is whether the optimization problem 5 is optimizable, and under which conditions we can obtain a lens operator. We will answer these questions in what follows.

### 3.3 Convergence to a Lens Operator

Consider now a random operator $\mathbf{U}$ as before, and the space $\mathcal{M}_\mathbf{x}^{\Theta(\mathfrak{X})} \subset \mathcal{M}_1^+$ of probability measures on $E$ generated by $\Theta(\mathfrak{X})$ and $\mathbf{x}$. This is,

$$\mathfrak{p} \in \mathcal{M}_\mathbf{x}^{\Theta(\mathfrak{X})} \iff \exists \mathbf{U} \text{ such that } \mathbb{P}_{\mathbf{Ux}} = \mathfrak{p}.$$

The following theorem and corollary establish the conditions for the optimization problem 5 to have a global minimum for $\mathcal{M}_1^+$ — i.e., a lens operator. We first will write it in terms of probabilities in $\mathcal{M}_\mathbf{x}^{\Theta(\mathfrak{X})} \subset \mathcal{M}_1^+$, and then we will show that one can rewrite it in terms of the random operators under certain conditions.

**Theorem 2.** *Consider $\mathbf{x}$ a random variable on $(E, \mathcal{T})$ — a separable metric space — and $\mathbf{U}$ a random operator taking values on $\Theta(\mathfrak{X}) \subset \mathcal{C}(\mathfrak{X})$. Further consider the associated RKHS $\mathcal{H}$ of functions on $E$ with characteristic kernel $\kappa$, and the induced MMD metric on $\mathcal{M}_1^+$. Under these conditions, if $\mathcal{M}_\mathbf{x}^{\Theta(\mathfrak{X})}$ is compact and $\mathbf{x}$ $\Theta(\mathfrak{X})$-myopic, we have that:*

*Given an iterative convergence strategy $\mathfrak{F}$ such that $\mathfrak{F}(\mathfrak{p}_{n-1}) = \mathfrak{p}_n \in \mathcal{N} \subset \mathcal{M}_1^+$ and $\{\mathfrak{p}_n\}_{n\in\mathbb{N}} \longrightarrow \mathfrak{p}' \in \arg\inf_{\mathfrak{p}\in\mathcal{N}} \mathrm{MMD}_\kappa(\mathbb{P}_\mathbf{x}, \mathfrak{p})$, it follows that:*

$$\mathfrak{F}(\mathfrak{p}_{n-1}) = \mathfrak{p}_n \in \mathcal{M}_\mathbf{x}^{\Theta(\mathfrak{X})} \implies \{\mathfrak{p}_n\}_{n\in\mathbb{N}} \longrightarrow \mathfrak{p}' \in \arg\min_{\mathfrak{p}\in\mathcal{M}_1^+} \mathrm{MMD}_\kappa(\mathbb{P}_\mathbf{x}, \mathfrak{p}) \text{ and } \mathfrak{p}' \in \mathcal{M}_\mathbf{x}^{\Theta(\mathfrak{X})}.$$

Logically, any way of obtaining a sequence in a subset $\mathcal{N} \subset \mathcal{M}_1^+$ whose limit optimizes $\min \mathrm{MMD}_\kappa(\mathbb{P}_\mathbf{x}, \mathfrak{p})$ in $\mathcal{N}$ — like (Arbel et al., 2019; Mroueh and Nguyen, 2021) —, can be used to obtain such a sequence in $\mathcal{M}_\mathbf{x}^{\Theta(\mathfrak{X})}$. Under Theorem 2, we know that such a sequence has a limit in $\mathcal{M}_\mathbf{x}^{\Theta(\mathfrak{X})}$, and that the limit will also be a global optimum in $\mathcal{M}_1^+$. The usefulness of this is made clear in the following corollary, which also gives us the conditions to write Theorem 2 in terms of operators on $\Theta(\mathfrak{X})$ — see Figure 3 for a visual summary. This corollary will allow us to solve equation 5 given a large enough sample size for $\widehat{\mathrm{MMD}}_\kappa$, and a proper way of sampling realizations of random operators.

**Corollary 3** (Convergence to a lens operator)**.** *Consider $\mathbf{x}$ a random variable on $(E, \mathcal{T})$ — a separable metric space — and $\mathbf{U}$ a continuous random operator taking values on $\Theta(\mathfrak{X}) \subset \mathcal{C}(\mathfrak{X})$. Further consider the associated RKHS $\mathcal{H}$ of functions on $E$ with characteristic kernel $\kappa$ and the induced MMD metric on $\mathcal{M}_1^+$. Under this conditions, if $\Theta(\mathfrak{X})$ is compact and $\mathbf{x}$ is $\Theta(\mathfrak{X})$-myopic, we have that*

*Given an iterative convergence strategy $\mathfrak{F}$ such that $\mathfrak{F}(\mathfrak{p}_{n-1}) = \mathfrak{p}_n \in \mathcal{N} \subset \mathcal{M}_1^+$ and $\{\mathfrak{p}_n\}_{n\in\mathbb{N}} \longrightarrow \mathfrak{p}' \in \arg\inf_{\mathfrak{p}\in\mathcal{N}} \mathrm{MMD}_\kappa(\mathbb{P}_\mathbf{x}, \mathfrak{p})$, it follows that:*

$$\{\mathbf{U}_n\}_{n\in\mathbb{N}} \text{ such that } \mathfrak{F}(\mathbb{P}_{\mathbf{U}_{n-1}\mathbf{x}}) = \mathbb{P}_{\mathbf{U}_n\mathbf{x}} \implies \mathrm{MMD}_\kappa(\mathbb{P}_\mathbf{x}, \mathbb{P}_{\mathbf{U}_n\mathbf{x}}) \longrightarrow 0, \text{ and } \{\mathbf{U}_n\}_{n\in\mathbb{N}} \xrightarrow{a.s} \mathbf{V} \in \Theta(\mathfrak{X}).$$

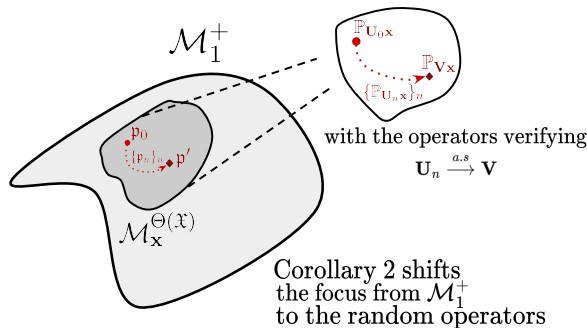

Figure 3: Diagram of the relation between Theorem 2 and Corollary 3.

In other words, Corollary 3 shows that Theorem 2 also imply that a sequence of operators $\{\mathbf{U}_n\}_{n\in\mathbb{N}}$ obtained via $\mathfrak{F}$, will converge almost surely to a lens operator in $\Theta(\mathfrak{X})$ — as long as $\Theta(\mathfrak{X})$ is compact, and $\mathbf{x}$ myopic. Thus, by Corollary 3, Equation 5 will have a solution that is a lens operator for $\mathbf{x}$.

Now that we know that we can solve Equation 5, there are only two questions left

1. In practice, how can we sample random operators to solve Equation 5 in a differentiable manner?
2. If we find a lens operator $\mathbf{V}$, can we still characterize the density $P_{\mathbf{V}\mathbf{x}}$ by the marginals $P_{V\mathbf{x}}$? That is, is there an equivalent to (Cribeiro-Ramallo et al., 2024, Proposition 1) in this general theory?

Solving Question 1 will give us a way to obtain lens operators in $\Theta(\mathfrak{X})$-myopic populations in practice. Section 4 will introduce such a method. Solving Question 2 is important to the downstream task of outlier detection. It is immediate under the assumptions of Corollary 3 by invoking the Disintegration and Radon-Nikodym Theorems (Faden, 1985; Simonnet, 1996). We included a more general result akin to (Cribeiro-Ramallo et al., 2024, Proposition 1) in the Appendix, as such generality is not necessary in our setting.

## 4 Adversarial Subspace Generation: $\mathbf{V}$-GAN

In this section, we will employ our previous theoretical findings to propose a method for sampling a lens operator. We will describe our setting and propose our method, and then propose a way to identify whether $\mathbf{V}$ is a lens operator or not. A pseudo-code of the training is included in the Appendix

### 4.1 Subspace Generation with MMD-GANs

Our goal is to find a way to sample a lens operator $\mathbf{V}$, i.e., we want to approximate the sampling function of $\mathbf{V}$ using a parametric model. In particular, we aim to learn a parametric function $G_\theta$, from an arbitrary latent space $\mathcal{Z}$ to the space of operators $\Theta(\mathfrak{X})$. The goal is that, when $G_\theta$ is composed with a uniform random variable $\mathbf{z}$ in $\mathcal{Z}$, $\mathbb{P}_{G_\theta(\mathbf{z})} = \mathbb{P}_{\mathbf{V}}$. We do so by minimizing the loss function:

$$\mathcal{L}(\theta) = \widehat{\mathrm{MMD}}_\kappa^2(\mathbb{P}_{\mathbf{x}}, \mathbb{P}_{G_\theta(\mathbf{z})\mathbf{x}}). \tag{6}$$

Approximating sampling functions is a common problem in the machine learning literature, being the main use case of generative models (Goodfellow et al., 2016, Chapter 20). In particular, Generative Moment Matching Networks (MMD-GANs) use the squared sample MMD as their loss function, written as

$$\mathcal{L}(\theta) = \widehat{\mathrm{MMD}}_\kappa^2(\mathbb{P}_{\mathbf{x}}, \mathbb{P}_{F_\theta(\mathbf{z})}),$$

with $F_\theta : z \in \mathcal{Z} \longmapsto F_\theta(z) = \hat{x} \in E$ a generative network. These networks guarantee convergence in distribution of $F_\theta(\mathbf{z})$ to $\mathbf{x}$ when minimizing the loss in terms of the parameters (Bińkowski et al., 2021; Arbel

et al., 2019; Li et al., 2017). However, none of them guarantee convergence to a solution in $\mathcal{M}_{\mathbf{x}}^{\Theta(\mathfrak{X})}$ that is a global optimum also in $\mathcal{M}_1^+$ — which we need for myopicity. Theorem 2 gives sufficient conditions that guarantee convergence within $\mathcal{M}_{\mathbf{x}}^{\Theta(\mathfrak{X})}$, and Corollary 3 writes it in terms of the space of operators $\Theta(\mathfrak{X})$.

As such, we will consider a neural network $G_\theta$ such that:

$$G_\theta : z \in \mathcal{Z} \longmapsto G_\theta(z) = U \in \Theta(\mathfrak{X}),$$

with $\Theta(\mathfrak{X})$ compact. In practice, the architecture of $G_\theta$, the metric space $E$, and the space of operators $\Theta(\mathfrak{X})$ have to be defined case-by-case. We will study the case of axis-parallel subspace selection, as it is the most common setting in the literature of subspace search and subspace outlier detection (Aggarwal, 2017). We call this strategy of searching subspaces by generating them *Subspace Generation*, and our proposed method, **V**-GAN. Section B.2 in the Appendix contains examples of how one can apply the Myopic Subspace Theory and **V**-GAN to different datatypes using different operators.

### 4.1.1 Axis-parallel Subspace Generation

Let $E = \mathbb{R}^d$ and $\Theta(\mathfrak{X}) = Diag(\{0,1\})_{d \times d}$, separable and compact, respectively. As matrix-vector multiplication with diagonal matrices is the same as the element-wise product of the vector and the diagonal, we will build $G_\theta$ such that $G_\theta(z) \in \{0,1\}^d$. Thus, the loss function of our network, given a set of samples $\{x_i\}_{i=1}^n$ and noise $\{z_j\}_{j=1}^n$, can be written as:

$$
\begin{aligned}
\mathcal{L}_\kappa\left(\{x_i\}_{i=1}^n, \{z_j\}_{j=1}^n; \theta\right) = \widehat{\mathrm{MMD}}_\kappa^2(\mathbb{P}_{\mathbf{x}}, \mathbb{P}_{G_\theta(\mathbf{z})\mathbf{x}}) = & \frac{1}{n(n-1)} \sum_{i=1}^n \sum_{\substack{j=1 \\ i \neq j}}^n \kappa(x_i, x_j) \\
& + \frac{1}{n(n-1)} \sum_{i=1}^n \sum_{\substack{j=1 \\ i \neq j}}^n \kappa(G_\theta(z_i) \odot x_i, G_\theta(z_j) \odot x_j) \\
& - \frac{2}{n^2} \sum_{i=1}^n \sum_{\substack{j=1 \\ i \neq j}}^n \kappa(x_i, G_\theta(z_j) \odot x_j),
\end{aligned}
\tag{7}
$$

with a characteristic kernel $\kappa$ (Gretton et al., 2012). To obtain a $G_\theta(z_j) \in \{0,1\}^d$ we will use the upper-softmax activation function, defined as:

$$\sigma_{\mathrm{us}}(x) = u\left(\sigma_{\mathrm{sm}}(x) - \frac{\vec{\mathbf{1}}}{d}\right),$$

with $\sigma_{\mathrm{sm}}$ the softmax activation, $u$ the element-wise unit-step function and $\frac{\vec{1}}{d}$ a vector of size $d$ with $\frac{1}{d}$ in each entry. As the unit-step function is not differentiable, we will use the softmax directly during backpropagation, similar to other binary-NN (Goodfellow et al., 2016). We described the particular layers employed in the experiments in the Experimental Details in Section 5.1.2.

### 4.1.2 Kernel Learning for **V**-GAN

The literature of MMD-GANs also studies the case of using kernel learning, where $\kappa$ is now a trainable function $\kappa_\phi$. Particularly, Li et al. provide a way to train such kernels while also maintaining the convergence guarantees. The resulting loss function can be written as:

$$\mathcal{L}_{\mathrm{kl}}\left(\{x_i\}_{i=1}^n, \{z_j\}_{j=1}^n; \theta, \phi)\right) = \mathcal{L}_{\kappa_\phi}\left(\{x_i\}_{i=1}^n, \{z_j\}_{j=1}^n; \theta\right) - \sum_{i=1}^n \|x_i - \mathcal{E}_\phi^{-1}(\mathcal{E}_\phi(x_i))\|_2, \tag{8}$$

with $\mathcal{E}_\phi$ and $\mathcal{E}_\phi^{-1}$ being an encoder and decoder network, and $\kappa_\phi = \kappa \circ \mathcal{E}_\phi = \kappa(\mathcal{E}_\phi(\cdot), \mathcal{E}_\phi(\cdot))$. $\kappa_\phi$ will be a characteristic kernel as long as $\kappa$ is characteristic and $\mathcal{E}_\phi$ is injective (Berlinet and Thomas-Agnan, 2004).

The second addend of Equation 8 guarantees the injectivity (Bińkowski et al., 2021). Thus, the optimization problem becomes (Li et al., 2017):

$$\min_{\theta} \max_{\phi} \mathcal{L}_{\text{kl}} \left( \{x_i\}_{i=1}^n, \{z_j\}_{j=1}^n; \theta, \phi \right). \tag{9}$$

## 4.2 A Test for Myopicity

In practice, we only have access to a sample of i.i.d. realizations of $\mathbf{x}$. Thus, if one limits itself to simply minimizing Loss (6), one could end with a random operator *approximately* lens. That is why, to assess whether the two random variables $\mathbf{x}$ and $\mathbf{Ux}$ have the same distribution, we need to use the following hypothesis test:

$$\begin{cases} H_0 : \mathbb{P}_{\mathbf{x}} = \mathbb{P}_{\mathbf{Ux}}, \\ H_a : \mathbb{P}_{\mathbf{x}} \neq \mathbb{P}_{\mathbf{Ux}}. \end{cases} \tag{10}$$

As the sample MMD's asymptotic distribution is tabulated, one can use it for such a statistical test.

In other words, we can test whether a given operator $\mathbf{U}$ is a lens operator for $\mathbf{x}$ by using the MMD test statistic (Gretton et al., 2012) for the Test 10. This is ideal, as Gretton et al. proved that the resulting test is asymptotically consistent[7]. Therefore, for a sufficiently large sample, we could study whether $\mathbf{U}$ is a lens operator for $\mathbf{x}$ with a probability of a false negative $\approx 0$. Thus, for any particular population, we can study whether a given $\mathbf{U}$ is a lens operator, allowing us to obtain random operators that are exactly lens in practice.

# 5 Experiments

We evaluate different aspects of $\mathbf{V}$-GAN as follows. First, we examine its ability to recover a derived lens operator. Second, we compare its effectiveness in building one-class classification ensembles across 42 real-world datasets to nine competitors. Finally, we analyze its scalability in comparison to other subspace selection methods. We will start by describing the experimental setup.

## 5.1 Experimental Details

This section has three parts. First, we describe the synthetic and real datasets for our experiments. Then, we describe $\mathbf{V}$-GAN's configuration. Finally, we introduce our competitors.

### 5.1.1 Datasets

**Real**  We used 42 normalized datasets from the benchmark study by Han et al., listed in Tables 11-15 in the appendix. For those datasets with multiple versions, we chose the first in alphanumeric order. Details about each dataset are available in (Han et al., 2022).

**Synthetic**  Consider the random variables $\mathbf{x}_1, \mathbf{x}_2, \mathbf{x}_3 \sim N(0, 1)$. As data for our experiments in section 5.2, we will consider a 3-dimensional population $\mathbf{x}$ generated by randomly drawing points from $S_1 = \langle \mathbf{x}_1, \mathbf{x}_2, 0 \rangle$ and $S_2 = \langle 0, 0, \mathbf{x}_3 \rangle$ with probabilities $F_1 = F$ and $F_2 = 1 - F$ respectively. In section 5.4 we generate $n$ points from a $d$-dimensional Uniform distribution and vary $n$ and $d$ to study the scalability of various methods.

### 5.1.2 Network Settings

**Generator**  Figure 4 contains a diagram of the architecture and the training of the generator. It features four hidden linear layers with an increasing number of neurons: $h_{l_1} = \frac{d}{8}$, $h_{l_2} = \frac{d}{4}$, $h_{l_3} = \frac{d}{2}$, and $h_{l_4} = d$, where $d$ represents the data dimensionality. The input layer from the latent space has $\frac{d}{16}$ neurons, while the output layer employs the upper softmax activation $\sigma_{\text{us}}$ — see Section 4.

---

[7]A test $\Delta$ is called consistent iff, given any level $\alpha$, the Type II error is $\beta = 0$.

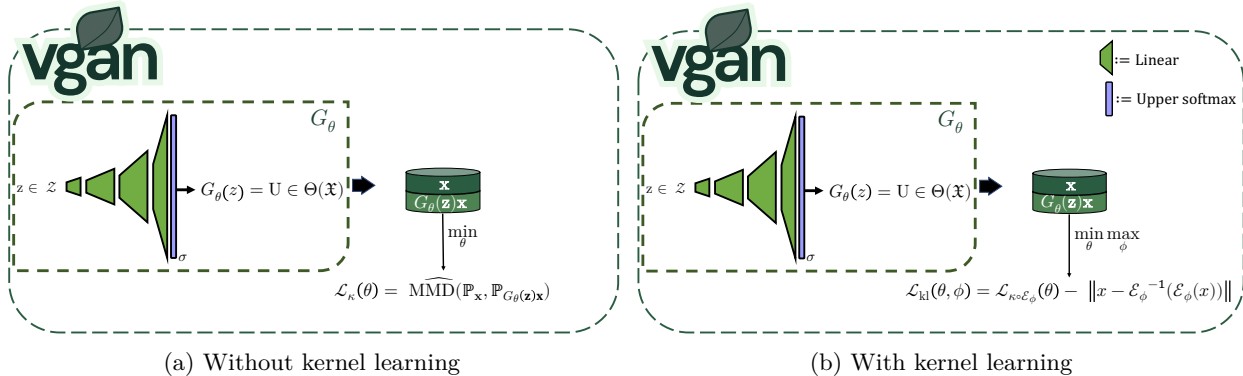

(a) Without kernel learning        (b) With kernel learning

Figure 4: Diagram of the network and the training without and with kernel learning, from left to right, respectively. (a) The network $G_\theta$ is trained to minimize the loss $\mathcal{L}_\kappa(\theta)$ — the empirical estimator of the MMD (Gretton et al., 2012) between samples of $\mathbf{x}$ and $G_\theta(\mathbf{z})\mathbf{x}$ using kernel $\kappa$. (b) The network $G_\theta$ is trained to minimize the same loss as before, but with $\kappa$ composed with $\mathcal{E}_\phi$, the encoder part of an autoencoder. At the same time, $\mathcal{E}_\phi$ is trained to maximize $\mathcal{L}_{\kappa \circ \mathcal{E}_\phi}$, while minimizing the reconstruction loss.

Table 2: Table of the different competitors in our experiments grouped by method type.

| Type | Competitors |
|---|---|
| Subspace Selection | CLIQUE 2005, HiCS 2012, GMD 2019 |
| Feature Selection | CAE 2019 |
| Embedding Method | PCA 1901, UMAP 2024, ELM 2023 |

**Kernel**    Unless stated otherwise, following the advice from (Li et al., 2017), we will use the kernel $\kappa_\phi = \varsigma \circ E_\phi$. Here, $\varsigma$ is a Gaussian kernel with the median heuristic bandwidth parameter (Garreau et al., 2018) and $E_\phi$ an encoder trained by kernel learning — see Section 4.1.2. Particularly, we use an upside-down version of the generator's hidden layers for $E_\phi$, with the identity function as the output layer.

**Training**    We trained the network for 2000 epochs, with minibatch gradient descent using the Adadelta optimizer (Zeiler, 2012) following preliminary results. In particular, we use batches of size 500, a learning rate of $lr_G = lr_E = 0.007$ for the generator and the encoder, respectively. We set momentum (0.99) and weight-decay (0.04) (Goodfellow et al., 2016). Additionally, we updated $E_\phi$ once every 5 epochs.

**Number of Subspaces**    We generate 500 samples of the lens operator $\mathbf{V}$ to approximate its distribution. Thus, the number of subspaces depends on the number of unique values of its distribution.

### 5.1.3 Competitors & Baselines

We selected popularly used in outlier detection and state-of-the-art (SotA) subspace search — CLIQUE (Agrawal et al., 2005), HiCS (Keller et al., 2012), GMD (Trittenbach and Böhm, 2019) — feature selection — CAE (Balın et al., 2019) —, and embedding methods — PCA (F.R.S., 1901), UMAP (Healy and McInnes, 2024), ELM (Xu et al., 2023) — with openly available implementations, as competitors; see Table 2 for a quick summary. As subspace discovery methods cannot project newly incoming data into the selected subspaces, they can not be used in our setting. For all methods included, we used the recommended parameters and training regimes. Specific details for each competitor are in the appendix, Section B.1. Additionally, we included regular Feature Bagging (FB) (Lazarevic and Kumar, 2005) as a baseline in Section 5.3.1. We built homogeneous feature ensembles using off-the-shelf outlier detectors in the outlier detection experiments. For the embedding methods, we used the embedded version of the dataset to fit a singular off-the-shelf detector. Specifically, we utilized the most popular and best-performing detectors from (Han et al., 2022): LOF, kNN,

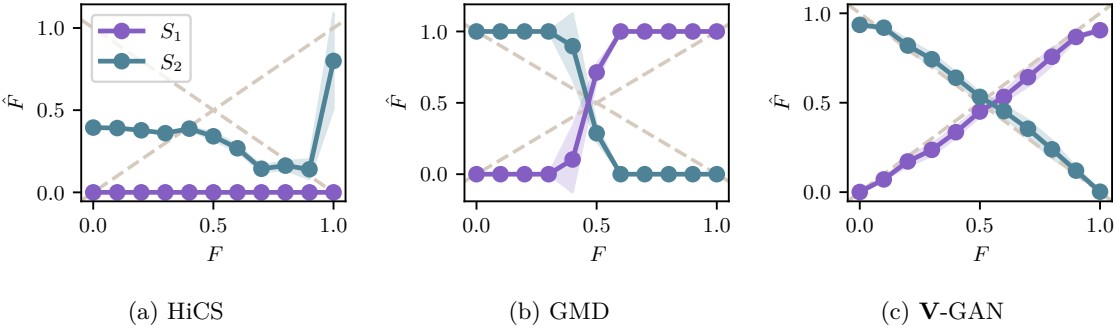

Figure 5: Comparison of the relative scores for each subspace across different values of $F$.

CBLOF, ECOD, and COPOD (Breunig et al., 2000; Aggarwal, 2017; He et al., 2003; Li et al., 2023; 2020), with their respective recommended or default parameters.

All experiments were implemented in Python. We used popular implementations for all competitors and baselines and implemented **V**-GAN in `PyTorch`. We used the Python package `pyod` for all outlier detectors. All subspace selection methods where implemented in `Python`, except HiCS, which is implemented in `Nim`. Experiments ran on a Ryzen 9 7900X CPU and an Nvidia RTX 4090 GPU.

## 5.2 Obtaining the Theoretical Lens Operator

To study the properties of the operator **V** obtained by **V**-GAN, we use synthetic data. Specifically, we consider a population similar to Example 1, where a lens operator can be directly calculated. Using the synthetic population described in Section 5.1.1, we define a random operator **U** with values $U_1 = \text{diag}(1, 1, 0)$ and $U_2 = \text{diag}(0, 0, 1)$, occurring with probabilities $F_1 = F$ and $F_2 = 1 - F$, respectively. This operator is trivially a lens operator. The experiment aims to extract subspaces $S_1$ and $S_2$ with scores $\hat{F}_1$ and $\hat{F}_2$ as close as possible to $F_1$ and $F_2$, using a subspace selection method. The steps are as follows:

1. Generate a dataset $D$ by sampling 10000 points from **x**.
2. Use $D$ to train a given subspace selection method.
3. Obtain the subspace qualities $\hat{F}_i$ of all selected subspaces $\{S_i\}$ and map them into $[0, 1]$ probabilities by $\hat{F}'_i = \frac{\hat{F}_i}{\sum_j \hat{F}_j}$.
4. Report the probabilities $\hat{F}'_1, \hat{F}'_2$ of subspaces $S_1$ and $S_2$.
5. Repeat steps $1-4$ 10 times.

In step 3 we considered the subspace selection methods HiCS and GMD (Keller et al., 2012; Trittenbach and Böhm, 2019) apart from **V**-GAN. We could not include the subspace selection method CLIQUE (Agrawal et al., 2005) as it does not report a quality metric for the subspaces. As we are using a 3-dimensional dataset, it will be enough to employ a regular Gaussian kernel with the recommended bandwidth parameter for **V**-GAN's training. We reported the results in Figure 5. As we can see, **V**-GAN is the only method capable of properly extracting the true weight of each subspace.

## 5.3 One-class Classification

This section presents outlier detection experiments using **V**-GAN to build ensembles. The goal is to detect outliers in a test set $D^{\text{test}}$ after training on an inlier train set $D^{\text{train}}$, a problem known as one-class classification (Perera et al., 2021). The experimental process is as follows:

1. Split the dataset $D$ into a training set $D^{\text{train}}$ containing 80% of the inliers from $D$, and a test set $D^{\text{test}}$ containing the remaining 20% and the outliers.
2. Obtain a collection of $K$ subspaces $\{S_i\}_{i=1}^K$ using a subspace selection method.

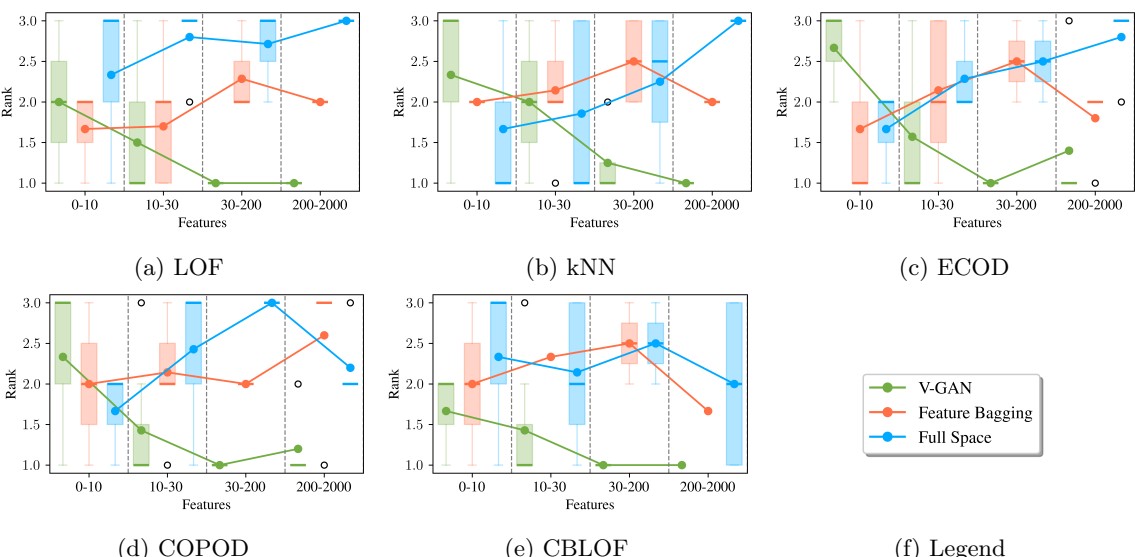

(a) LOF    (b) kNN    (c) ECOD

(d) COPOD    (e) CBLOF    (f) Legend

Figure 6: Boxplots of ranks of the comparison with baselines using myopic datasets with different numbers of features. The bins contained 3, 7, 5, and 6 datasets, respectively.

Table 3: Results of the Conover-Iman test for the rankings against baselines on myopic datasets

|  | LOF | | | kNN | | | ECOD | | | COPOD | | | CBLOF | | |
|---|---|---|---|---|---|---|---|---|---|---|---|---|---|---|---|
|  | FB | None | **V-GAN** | FB | None | **V-GAN** | FB | None | **V-GAN** | FB | None | **V-GAN** | FB | None | **V-GAN** |
| FB |  | + + | - - |  |  | - - |  |  | - - |  |  | - - |  |  | - - |
| None | - - |  | - - |  |  | - - |  |  | - - |  |  | - - |  |  |  |
| **V-GAN** | + + | + + |  | + + | + + |  | + + | + + |  | + + | + + |  | + + | + + |  |

3. Given an outlier detector $\mathcal{M}$, obtain $\{\mathcal{M}_i\}_{i=1}^K$ by fitting $\mathcal{M}$ on each of the $K$ selected subspaces. As a dataset, use $D^{\mathrm{train}}|_{S_i}$, the projection of $D^{\mathrm{train}}$ into the subspace.

4. Evaluate the performance of each detector by reporting the AUC of the aggregated scores across all $D^{\mathrm{test}}|_{S_i}$. If $K = 1$ (like in feature selection), use the score in $D^{\mathrm{test}}|_S$.

5. Repeat steps 2 to 4 10 times.

We aim to address two key questions about the performance of **V**-GAN's lens operator: (Q1) *How does it compare to baselines for outlier detection, such as the full-space method and a randomly selected collection of subspaces (feature bagging)?* (Q2) *How does it perform relative to other subspace selection methods and dimensionality reduction techniques?* Furthermore, we will evaluate its performance on datasets with and without a myopic distribution, providing insights into both the best-case scenario (where **V** acts as a lens operator) and the worst-case scenario (where **V** does not).

### 5.3.1 Comparison with Baselines (Q1)

In this section, we compare **V**-GAN to two classical baselines in the subspace selection literature: the full-space method and Feature Bagging (FB) (Lazarevic and Kumar, 2005). For FB, we chose the number of subspaces $K$ from a set of five equidistant values from 50 to 500. For each dataset, we selected the $K$ yielding the highest average AUC across 10 repetitions. To aggregate scores, we used a weighted average based on the probability assigned to each subspace, following (Cribeiro-Ramallo et al., 2024, Proposition 1). For FB, this reduces to a simple average.

Furthermore, we will evaluate its performance on datasets with and without a myopic distribution, providing insights into both the best-case scenario (where **V** is a lens operator, and hence **x** is myopic) and the worst-case scenario (where **V** is not a lens operator, and hence **x** is not myopic). To study whether this is the case for each dataset, we will study the hypothesis test presented in Section 4.2. We collected the

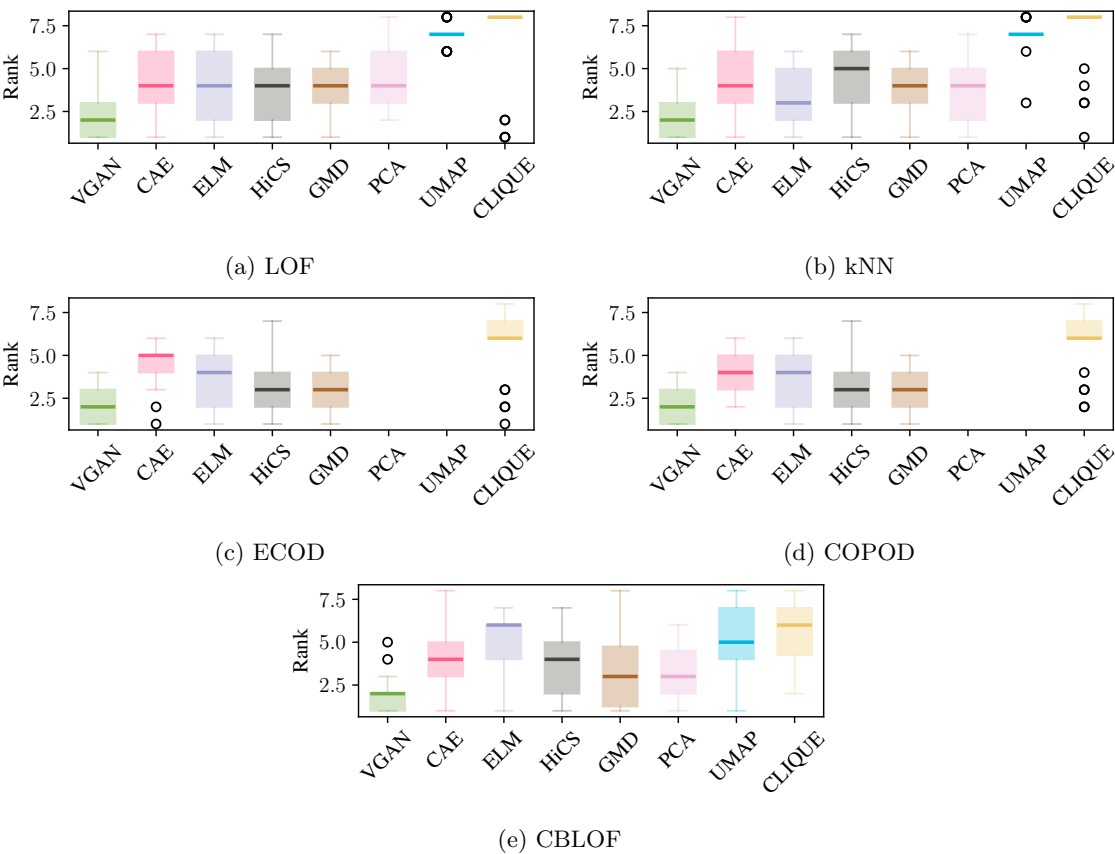

Figure 7: Boxplots of ranks of the comparison with our competitors using myopic datasets.

test results in Tables 11-15. If the test returns in favor of the null hypothesis, then the operator is lens (i.e., $\text{MMD}(\mathbb{P}_{\mathbf{x}}, \mathbb{P}_{\mathbf{Ux}})$) for the studied population. Further details can be found in Section B.1 in the Appendix.

**Myopic Datasets.** Figure 6 shows rankings contingent on dataset dimensionality group and average rankings. **V**-GAN demonstrates consistent performance improvements as dimensionality increases, often outperforming baselines for all outlier detectors. To assess statistical significance, we apply the Conover-Iman post-hoc test (Conover and Iman, 1979), commonly used in outlier detection (Campos et al., 2016), following a preliminary positive result from the Kruskal-Wallis test (Kruskal and Wallis, 1952). Table 3 contains the results, where '+' indicates the row method has a significantly lower median rank than the column method, and '−' indicates a significantly higher rank. One symbol marks p-values $\leq 0.1$, two symbols mark p-values $\leq 0.05$, and blanks indicate no significant difference. Entirely grayed-out subtables denote cases where the Kruskal-Wallis test, a prerequisite for using the Conover-Iman post-hoc test, was not passed. **V**-GAN outperformed all baselines across detectors. The appendix summarizes the complete AUC results in Tables 11-15.

**Non-myopic Datasets.** Non-myopicity represents the worst-case scenario for **V**-GAN, as its guarantees rely on this property. Figure 9 in the Appendix B.1 shows rank boxplots for non-myopic datasets, similar to those for myopic datasets. It is evident that **V**-GAN's performance is not worse than any baseline, and the Conover-Iman test results (Table 5) support this. **V**-GAN's performance in its worst-case scenario is no worse than that of a tuned feature bagging (FB) and outperforms the full-space approach for some outlier detectors.

Table 4: Results of the Conover-Iman test for the rankings against competitors on myopic datasets

| OD method | SS method | CAE | HiCS | CLIQUE | ELM | GMD | PCA | UMAP | **V-GAN** |
|---|---|---|---|---|---|---|---|---|---|
| LOF | CAE | | | ++ | | | | ++ | −− |
| | HiCS | | | ++ | | | | ++ | −− |
| | CLIQUE | −− | −− | | −− | −− | −− | | −− |
| | ELM | | | ++ | | | | ++ | −− |
| | GMD | | | ++ | | | | ++ | −− |
| | PCA | | | ++ | | | | ++ | −− |
| | UMAP | −− | −− | | −− | −− | −− | | −− |
| | **V-GAN** | ++ | ++ | ++ | ++ | ++ | ++ | ++ | |
| kNN | CAE | | | ++ | −− | | + | ++ | −− |
| | HiCS | | | ++ | + | | | ++ | −− |
| | CLIQUE | −− | −− | | −− | −− | −− | | −− |
| | ELM | ++ | − | ++ | | | | ++ | −− |
| | GMD | | | ++ | | | | ++ | −− |
| | PCA | − | | ++ | | | | ++ | −− |
| | UMAP | −− | −− | | −− | −− | −− | | −− |
| | **V-GAN** | ++ | ++ | ++ | ++ | ++ | ++ | ++ | |
| ECOD | CAE | | −− | ++ | + | −− | | | −− |
| | HiCS | ++ | | ++ | | | | | −− |
| | CLIQUE | −− | −− | | −− | −− | | | −− |
| | ELM | − | | ++ | | | | | −− |
| | GMD | ++ | | ++ | | | | | −− |
| | PCA | | | | | | | | |
| | UMAP | | | | | | | | |
| | **V-GAN** | ++ | ++ | ++ | ++ | ++ | | | |
| COPOD | CAE | | −− | ++ | | −− | | | −− |
| | HiCS | ++ | | ++ | | | | | −− |
| | CLIQUE | −− | −− | | −− | −− | | | −− |
| | ELM | | | ++ | | | | | −− |
| | GMD | ++ | | ++ | | | | | − |
| | PCA | | | | | | | | |
| | UMAP | | | | | | | | |
| | **V-GAN** | ++ | ++ | ++ | ++ | + | | | |
| CBLOF | CAE | | | ++ | | | | | −− |
| | HiCS | | | ++ | ++ | | | ++ | −− |
| | CLIQUE | −− | −− | | | −− | −− | | −− |
| | ELM | | −− | | | −− | −− | | −− |
| | GMD | | | ++ | ++ | | | ++ | − |
| | PCA | | | ++ | ++ | | | ++ | −− |
| | UMAP | | −− | | | −− | −− | | −− |
| | **V-GAN** | ++ | ++ | ++ | ++ | + | ++ | ++ | |

### 5.3.2 Comparison with Competitors (Q2)

We now compare **V**-GAN to the competitors introduced in Section 5.1.3 (see Table 1). As before, we analyze performance separately for myopic and non-myopic datasets.

**Myopic Datasets.** Figure 7 plots the ranks of all competitors for myopic datasets. **V**-GAN consistently achieves the lowest median rank, with GMD typically being the closest competitor. Table 4 contains the results of the Conover-Iman test. **V**-GAN significantly outperforms all methods and is the best option for enhancing outlier detection performance under myopicity.

**Non-myopic Datasets.** Figure 10 plots ranks for the non-myopic case, and Table 6 contains the Conover-Iman test results. **V**-GAN demonstrates a closer performance to its competitors on non-myopic datasets, as expected, but it is never statistically worse than any competitor. Therefore, we can recommend **V**-GAN as a default approach for ensemble outlier detection using subspaces, which brings significant advantages in the myopic case while having no apparent disadvantage in the absence of myopicity.

### 5.4 Scalability

We now compare the scalability of **V**-GAN with other subspace search methods. For these experiments, all methods were tested with their parameters set as in Section 5.1.2. The dataset consists of uniformly

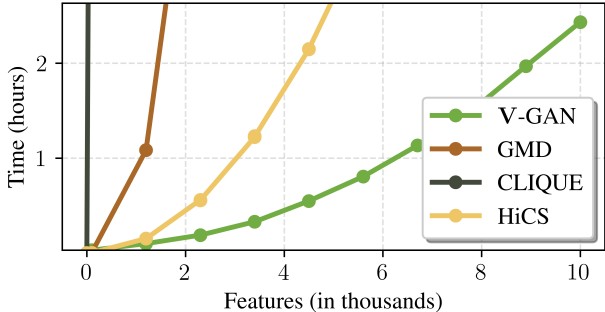

Figure 8: Time taken for performing subspace search, per feature count.

generated noise with $d \in \{0.1, 1.2, 2.3, 3.4, 4.5, 5.6, 6.7, 7.8, 8.9, 10\} \cdot 10^3$ features. All experiments are run using a single CPU thread to ensure a fair comparison.

Figure 8 presents the results of scalability experiments, showing the runtime in hours required to obtain a collection of subspaces as a function of the number of features. **V**-GAN is more scalable than all subspace search competitors: It is over 4, 30, and 8000 times faster than HiCS, GMD, and CLIQUE, respectively.

## 6 Limitations & Future Work

The experiments in this article center around obtaining lens operators in high-dimensional tabular data. While our specific setting work with simple axis-parallel projections, MST potentially allows arbitrary levels of complexity for $\Theta(\mathfrak{X})$. This, especially for high-dimensional tabular data, can lead to a limitation of the interpretability of the derived operators. This limitation is a common characteristic of tabular data, leading to its own field of *interpretable models for tabular data* aiming to alleviate this (Yan et al., 2025; Si et al., 2023). For instance, it is clear that in domains such as image processing or NLP, the results of **V**-GAN are interpretable — see Figures 12 and 14. We believe that further research into these interpretable datatypes can help to better interpret lens operators for tabular data.

While MST is a general theory, it does not consider general stochastic structures of the form:

$$\mathbf{x}. : \Omega \times T \longrightarrow E$$
$$\omega, t \mapsto \mathbf{x}_t(\omega) = x_t,$$

i.e., it does not consider time-correlated datatypes like time series, functional data, or Markov processes. Analyzing such datatypes requires a suitable extension of MST, and thus, we consider it as future work. Furthermore, we focus our experimental evaluation on outlier detection, but there exist other downstream tasks interested in obtaining projections of the data, like clustering (Cheng et al., 1999; Agrawal et al., 2005; Qu et al., 2023). In this setting, a classical approach is to employ some clustering aggregation method, like consensus clustering (Lipor et al., 2021), or hierarchical clustering (Baumgartner et al., 2004). Another interesting application, that might not necessarily consider projections, is contrastive learning. In contrastive learning, the goal is to augment the data by utilizing transformations $x^+$ of samples of **x** with high similarity (Chen et al., 2020). This augmented data, together with other samples $x^-$ not following the same distribution as **x**, are used to train representation learning methods for a wide variety of downstream tasks(Chen et al., 2020; Van Gansbeke et al., 2021). MST could provide a set of particular transformations given by **U** that have guarantees to be optimal in terms of similarity — since $\mathbb{P}_{\mathbf{x}} = \mathbb{P}_{\mathbf{Ux}}$. This is appealing, as we know that there exists literature on obtaining better transformations by heuristically searching from a predefined collection of transformations (Cubuk et al., 2019). Studying whether MST can derive useful transformations in practice, and how it can affect the rich theoretical background of contrastive learning (Wang et al., 2021) is treated as future work.

Next, in Section 5 we performed experiments for outlier detection using **V**-GAN. Therefore, we set up a default choice of hyperparameters for the training. This was to study how easily one can converge to a

significant solution in real, practical scenarios. As we have Test (10), we can study whether the network has converged to a solution or not. While in our case one could derive lens operators effortlessly — see Table 11's last column — we believe that an analysis of the effect of hyperparameters in training can help with harder datatypes. Firstly, harder, higher-dimensional datatypes like high-resolution images may result in a more complicated optimization surface for Equation (8). Secondly, hyperparameters might not only affect the optimization surface, but also Test (10) itself. Particularly, Test (10) depends on the selected kernel for its performance, as mentioned in (Gretton et al., 2012; Bińkowski et al., 2021; Schrab et al., 2023). Lastly, while Test (10) is consistent — see Section 4.2 —, its consistency is only asymptotic. This means that the performance of Test (10) depends heavily on the batch size selected — the higher the better. This is not a problem for our particular setting, but it can be one for datasets of much higher dimensionality — like text with a large token count, or high-resolution images — or when trying to use models with a high memory complexity — like large transformers. These four limitations are shared between all MMD-based generative models and kernel-based inference methods (Gretton et al., 2012; Li et al., 2015). Thus, an exhaustive analysis like the one performed in (Bińkowski et al., 2021) is considered an important future work.

## 7 Conclusions

Subspace search can improve outlier detection for an off-the-shelf detector in tabular data (Müller et al., 2012; Keller et al., 2012; Nguyen et al., 2014; Trittenbach and Böhm, 2019). In our experiments, however, we did not observe this improvement in all datasets, with the methods sometimes failing to beat näive baselines. Besides, existing subspace search methods can hardly be applied to non-tabular data due to poor scaling. The above-mentioned factors hindered the use of such methods in practice. This paper proposes a new theoretical framework that explains **when** subspace selection is helpful and, more importantly, **how** we can exploit it to our advantage. Using this theory, we introduced a new way of performing subspace selection, akin to subspace search methods, that is theoretically sound, scalable, and usable in general scenarios — a strategy that we called *subspace generation*. Our first attempt in subspace generation, called **V**-GAN, demonstrates a significant performance increase against other baselines and competitors in the downstream task of outlier detection — one of the main use cases for subspace search. In addition, our experiments suggest that the performance increase is conditioned on the data's distribution being myopic, a property we can infer from data without any prior knowledge. Furthermore, even when the data is not myopic, **V**-GAN is still not outperformed by its competitors. Our findings not only validate the superior performance of **V**-GAN for subspace selection, but also show the potential of our Myopic Subspace Theory (MST) beyond the use case of outlier detection on tabular data.

### Aknowledgements

This work was supported by the Ministry of Science, Research and the Arts Baden-Württemberg, project Algorithm Engineering for the Scalability Challenge (AESC).

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

# A   Theoretical Appendix

This appendix contains the proofs for all the statements in Section 3, extra general statements, and a collection of examples of lens operators on different spaces.

### A.1 Myopic Subspace Theory (Extension)

We will first introduce all of the proofs of the statements from Section 3, and then introduce all of the additional statements and proofs. To maintain the clarity of this section, we will reintroduce all of the statements before their proofs.

**Lemma 1.** *Consider $\mathcal{H}$ a RKHS with a characteristic kernel $\kappa$; and $\mathbf{x}$, $\mathbf{U}$ and MMD as previously defined. Further, consider $\mathbf{V}$ to be a lens operator for $\mathbf{x}$. Then,*

$$\underset{\mathbf{U}}{\arg\min} \mathrm{MMD}_\kappa(\mathbb{P}_\mathbf{x}, \mathbb{P}_{\mathbf{Ux}}) \ni \mathbf{V}$$

*Proof.* $\mathbf{V}$ lens for $\mathbf{x} \implies \mathbb{P}_{\mathbf{Vx}} = \mathbb{P}_\mathbf{x} \implies \mathrm{MMD}_\kappa(\mathbb{P}_{\mathbf{Vx}}, \mathbb{P}_\mathbf{x}) = 0 \implies \mathbf{V} \in \underset{\mathbf{U}}{\arg\min} \mathrm{MMD}(\mathbb{P}_\mathbf{x}, \mathbb{P}_{\mathbf{Vx}})$. The first implication comes from the definition of a lens operator, the second for $\kappa$ being characteristic, and the last one is trivial when considering $\mathrm{MMD}_\kappa(\mathfrak{p}, \mathfrak{q}) \geq 0, \ \forall \mathfrak{p}, \mathfrak{q} \in \mathcal{M}_1^+$. $\square$

**Theorem 2.** *Consider $\mathbf{x}$ a random variable on $(E, \mathcal{T})$ — a separable metric space — and $\mathbf{U}$ a random operator taking values on $\Theta(\mathfrak{X}) \subset \mathcal{C}(\mathfrak{X})$. Further consider the associated RKHS $\mathcal{H}$ of functions on $E$ with characteristic kernel $\kappa$, and the induced MMD metric on $\mathcal{M}_1^+$. Under these conditions, if $\mathcal{M}_\mathbf{x}^{\Theta(\mathfrak{X})}$ is compact and $\mathbf{x}$ $\Theta(\mathfrak{X})$-myopic, we have that:*

> *Given an iterative convergence strategy $\mathfrak{F}$ such that $\mathfrak{F}(\mathfrak{p}_{n-1}) = \mathfrak{p}_n \in \mathcal{N} \subset \mathcal{M}_1^+$ and $\{\mathfrak{p}_n\}_{n\in\mathbb{N}} \longrightarrow \mathfrak{p}' \in \underset{\mathfrak{p}\in\mathcal{N}}{\arg\inf} \mathrm{MMD}_\kappa(\mathbb{P}_\mathbf{x}, \mathfrak{p})$, it follows that:*
>
> $$\mathfrak{F}(\mathfrak{p}_{n-1}) = \mathfrak{p}_n \in \mathcal{M}_\mathbf{x}^{\Theta(\mathfrak{X})} \implies \{\mathfrak{p_n}\}_{n\in\mathbb{N}} \longrightarrow \mathfrak{p}' \in \underset{\mathfrak{p}\in\mathcal{M}_1^+}{\arg\min} \mathrm{MMD}_\kappa(\mathbb{P}_\mathbf{x}, \mathfrak{p}) \ and \ \mathfrak{p}' \in \mathcal{M}_\mathbf{x}^{\Theta(\mathfrak{X})}.$$

*Proof.* By the definition of $\mathfrak{F}$, we can construct a sequence $\{\mathfrak{p}_n\}_{n\in\mathbb{N}} \in \mathcal{M}_\mathbf{x}^{\Theta(\mathfrak{X})}$ such that

$$\{\mathfrak{p}_n\}_{n\in\mathbb{N}} \longrightarrow \mathfrak{p}' \in \underset{\mathfrak{p}\in\mathcal{M}_\mathbf{x}^{\Theta(\mathfrak{X})}}{\arg\min} \mathrm{MMD}_\kappa(\mathbb{P}_\mathbf{x}, \mathfrak{p}) \tag{11}$$

Since $\mathbf{x}$ is $\Theta(\mathfrak{X})-$myopic, $\exists \mathbf{V} : \Omega \longrightarrow \Theta(\mathfrak{X})$ that is a lens operator for $\mathbf{x}$. By Lemma 1, and the definion of $\mathcal{M}_\mathbf{x}^{\Theta(\mathfrak{X})}$, is clear that:

$$\mathbb{P}_{\mathbf{Vx}} \in \underset{\mathfrak{p}\in\mathcal{M}_\mathbf{x}^{\Theta(\mathfrak{X})}}{\arg\min} \mathrm{MMD}_\kappa(\mathbb{P}_\mathbf{x}, \mathfrak{p}). \tag{12}$$

Additionally, by the definition of a lens operator,

$$\mathbb{P}_{\mathbf{Vx}} \in \underset{\mathfrak{p}\in\mathcal{M}_1^+}{\arg\min} \mathrm{MMD}_\kappa(\mathbb{P}_\mathbf{x}, \mathfrak{p}). \tag{13}$$

Thus, by (11), (12), and (13), is clear that

$$\mathfrak{p}' \in \underset{\mathfrak{p}\in\mathcal{M}_1^+}{\arg\min} \mathrm{MMD}_\kappa(\mathbb{P}_\mathbf{x}, \mathfrak{p}).$$

Additionally, as $\{\mathfrak{p}_n\}_{n\in\mathbb{N}}$ is a sequence in a compact space,

$$\{\mathfrak{p}_n\}_{n\in\mathbb{N}} \longrightarrow \mathfrak{p}' \in \mathcal{M}_\mathbf{x}^{\Theta(\mathfrak{X})}.$$

$\square$

**Corollary 3** (Convergence to a myopic operator)**.** *Consider $\mathbf{x}$ a random variable on $(E, \mathcal{T})$ — a separable metric space — and $\mathbf{U}$ a continous random operator taking values on $\Theta(\mathfrak{X}) \subset \mathcal{C}(\mathfrak{X})$. Further consider the associated RKHS $\mathcal{H}$ of functions on $E$ with characteristic kernel $\kappa$ and the induced MMD metric on $\mathcal{M}_1^+$. Under this conditions, if $\Theta(\mathfrak{X})$ is compact and $\mathbf{x}$ is $\Theta(\mathfrak{X})$-myopic, we have that*

*Given an iterative convergence strategy $\mathfrak{F}$ such that $\mathfrak{F}(\mathfrak{p}_{n-1}) = \mathfrak{p}_n \in \mathcal{N} \subset \mathcal{M}_1^+$ and $\{\mathfrak{p}_n\}_{n\in\mathbb{N}} \longrightarrow \mathfrak{p}' \in \underset{\mathfrak{p}\in\mathcal{N}}{\arg\inf}\, \mathrm{MMD}_\kappa(\mathbb{P}_\mathbf{x}, \mathfrak{p})$, it follows that:*

$$\{\mathbf{U}_n\}_{n\in\mathbb{N}}\ \textit{such that}\ \mathfrak{F}(\mathbb{P}_{\mathbf{U}_{n-1}\mathbf{x}}) = \mathbb{P}_{\mathbf{U}_n\mathbf{x}} \Longrightarrow \mathrm{MMD}_\kappa(\mathbb{P}_\mathbf{x}, \mathbb{P}_{\mathbf{U}_n\mathbf{x}}) \longrightarrow 0,\ \textit{and}\ \{\mathbf{U}_n\}_{n\in\mathbb{N}} \longrightarrow \mathbf{V} \in \Theta(\mathfrak{X}).$$

*Proof.* Consider $\{\mathfrak{p}_n\}_{n\in\mathbb{N}} \in \mathcal{M}_\mathbf{x}^{\Theta(\mathfrak{X})}$, such that $\mathfrak{F}(\mathfrak{p}_{n-1}) = \mathfrak{p}_n$. By Zorn's Lemma, one can construct a parallel sequence $\{\mathbf{U}_n\}_{n\in\mathbb{N}}$ such that $\mathfrak{F}(\mathfrak{p}_{n-1}) = \mathfrak{p}_n = \mathbb{P}_{\mathbf{U}_n\mathbf{x}}$. As such,

$$\{\mathbb{P}_{\mathbf{U}_n\mathbf{x}}\} \longrightarrow \mathfrak{p}' \in \underset{\mathfrak{p}\in\mathcal{M}_\mathbf{x}^{\Theta(\mathfrak{X})}}{\arg\inf}\, \mathrm{MMD}(\mathbb{P}_\mathbf{x}, \mathfrak{p}).$$

If $\mathcal{M}_\mathbf{x}^{\Theta(\mathfrak{X})}$ is compact, we can solve the remainder of the proof equivalently as done for Theorem 2. Thus, we will focus on proving such a statement.

$\Theta(\mathfrak{X})$ compact $\Longrightarrow \{U_n\}_{n\in\mathbb{N}} \longrightarrow V$ for all sequences in $\Theta(\mathfrak{X})$. If we now consider $\{U_n\}_{n\in\mathbb{N}}$ and $\mathbf{V}$ such that $\mathbf{U}_n(\omega) = U_n \longrightarrow V = \mathbf{V}(\omega)$ for almost all $\omega \in \Omega$, it is clear that:

$$\mathbb{P}(\lim_n \mathbf{U}_n(\omega) = \mathbf{V}(\omega)) = 1.$$

In other words, $\mathbf{U}_n \xrightarrow{a.s} \mathbf{V}$. Therefore, since $\Theta(\mathfrak{X}) \subset \mathcal{C}(\mathfrak{X})$ and $E$ is a separable metric space, by the definition of almost sure convergence it is clear that:

$$\mathbf{U}_n \xrightarrow{a.s} \mathbf{V} \Longrightarrow \mathbf{U}_n\mathbf{x} \xrightarrow{a.s} \mathbf{V}\mathbf{x}.$$

And lastly, by the convergence laws of random variables, we know that:

$$\mathbf{U}_n\mathbf{x} \xrightarrow{a.s} \mathbf{V}\mathbf{x} \Longrightarrow \mathbf{U}_n\mathbf{x} \xrightarrow{d} \mathbf{V}\mathbf{x} \Longrightarrow \mathbb{P}_{\mathbf{U}_n\mathbf{x}} \longrightarrow \mathbb{P}_{\mathbf{V}\mathbf{x}}.$$

$\square$

Now, we introduce the result mentioned at the end of Section 3.3. This result motivates the way we aggregate in our outlier detection experiments.

**Proposition 4.** *Consider $E$ a Radon space, $(\Omega, \mathcal{F}, \mathbb{P})$ a probability space, $\mathfrak{X}$ the space of random variables on $E$, and $\Theta(\mathfrak{X})$ the space of operators from $\mathfrak{X}$ to $\mathfrak{X}$. Further consider all $U \in \Theta(\mathfrak{X})$ to be defined on fibers of $E$, and $\mathbf{U} : \Omega \longrightarrow \Theta(\mathfrak{X}) \subset \mathcal{C}(\mathfrak{X})$ a lens operator for $\mathbf{x} \in \mathfrak{X}$. Lastly, consider the following conditions*

    *i)* $\mathbf{U}$ *is such that, given any two realizations $U_1$ and $U_2$ $(U_1 \neq U_2)$, if $\mathbb{P}_{U_1\mathbf{x}}(A) \neq 0 \Longrightarrow \mathbb{P}_{U_2\mathbf{x}}(A) = 0$, for $A \in \mathcal{F}(\mathbf{U}\mathbf{x})$ — i.e., all realizations are mutually exclusive.*

    *ii)* *The set of all realizations of $\mathbf{U}$ is countable.*

    *iii)* *There exists a measure $\mu$ such that $\mu >> \mathbb{P}_{\mathbf{U}\mathbf{x}}$ and $\mu >> \mathbb{P}_{\mathbf{U}(\omega)\mathbf{x}}$, $\forall \omega \in \Omega$.*

*In this case, $\mathbb{P}_\mathbf{x} = \sum_{\omega\in\Omega} \mathbb{P}_\mathbf{U}(\mathbf{U}(\omega))\mathbb{P}_{\mathbf{U}(\omega)\mathbf{x}}$ and $P_\mathbf{x} = \sum_{\omega\in\Omega} \mathbb{P}_\mathbf{U}(\mathbf{U}(\omega))P_{\mathbf{U}(\omega)\mathbf{x}}$.*

*Proof.* Consider all $U \in \Theta(\mathfrak{X})$ to be defined on fibers of $E$. By the disintegration theorem, we know that the pushforward functions $U_*\mathbb{P}_\mathbf{x}$ are (probability) measures. Then, by $(i) - (ii)$, we can apply the law of total probabilities to derive:

$$\mathbb{P}_{\mathbf{U}\mathbf{x}} = \sum_{\omega\in\Omega} \mathbb{P}_\mathbf{U}(\mathbf{U}(\omega))\mathbb{P}_{\mathbf{U}(\omega)\mathbf{x}}.$$

The result for the densities (in the Radon-Nikodym sense) is immediate by the Radon-Nikodym Theorem. $\square$

These conditions are trivially fulfilled by the axis-parallel case for outlier detection — akin to the one in (Cribeiro-Ramallo et al., 2024, Proposition 1). Consider that this result focuses on the case when the operators $U \in \Theta(\mathfrak{X})$ each reside on different fibers of the space $E$, which is enough for our downstream setting.

### A.2 Subspace Generation with MMD-GANs (Extension)

In this Section, we will extend the results from Section 4.1 by including a pseudo-code of **V**-GAN in Algorithm 1. Here, we included the training for **V**-GAN with kernel learning. Using an identity matrix $I$ as the encoder is sufficient to derive the pseudo-code for training without kernel learning. In practice, the simultaneous training of the generator $G$ and the autoencoder $(\mathcal{E}_\phi, \mathcal{E}_\phi^{-1})$ has to be done sequentially. In other words, we will train the autoencoder for a given number of epochs first, then the generator, and after that, we restart the loop until we reach the maximum number of epochs.

---

**Algorithm 1 V-GAN training**

---

**Require:** Dataset $D$, the RKHS kernel $\kappa$, *epochs*, *batches*, number of epochs training the autoencoder $iternum_{\mathcal{E}_\phi}$, number of epochs training the generator $iternum_{G_\theta}$

1: Initialize Generator $G_\theta$
2: Initialize the Encoder $\mathcal{E}_\phi$ and Decoder $\mathcal{E}_\phi^{-1}$
3: **for** $epoch \in \{1, ..., epochs\}$ **do**
4:    **for** $batch \in \{1, ..., batches\}$ **do**
5:       $noise \leftarrow$ Random noise $z^{(1)}, ..., z^{(m)}$ from $Z$
6:       $data \leftarrow$ Draw current batch $x^{(1)}, ..., x^{(m)}$
7:       $trained\_epochs_{\mathcal{E}_\phi} = 0$, $trained\_epochs_{G_\theta} = 0$
8:       **if** $trained\_epochs_{\mathcal{E}_\phi} < iternum_{\mathcal{E}_\phi}$ **then**
9:          Update $\mathcal{E}_\phi$ and $\mathcal{E}_\phi^{-1}$ by ascending the stochastic gradient: $\nabla_\phi \mathcal{L}_{\mathrm{kl}}(data, noise; \theta, \phi)$
10:         $trained\_epochs_{\mathcal{E}_\phi}\mathrel{+}= 1$
11:       **else if** $trained\_epochs_{G_\theta} < iternum_{G_\theta}$ **then**
12:          Update $G$ by descending the stochastic gradient: $\nabla_\theta \mathcal{L}_{\mathrm{kl}}(data, noise; \theta, \phi)$
13:          $trained\_epochs_{G_\theta}\mathrel{+}= 1$
14:          **if** $trained\_epochs_{G_\theta} \geq iternum_{G_\theta}$ & $trained\_epochs_{\mathcal{E}_\phi} \geq iternum_{\mathcal{E}_\phi}$ **then**
15:             $trained\_epochs_{\mathcal{E}_\phi} = 0$, $trained\_epochs_{G_\theta} = 0$
16:          **end if**
17:       **end if**
18:    **end for**
19: **end for**

---

Algorithm 1 takes as input the dataset $D$, the kernel $\kappa$, the number of epochs and batches, and the iteration count for the autoencoder and generator — $iternum_{\mathcal{E}_\phi}$ and $iternum_{G_\theta}$, respectively. During training (lines 3-19), a batch of data points is drawn from $D$, and an equal amount of random noise is sampled (lines 5-6). The update loop (lines 8-17) alternates between updating the encoder and the generator. The autoencoder is updated (lines 8-10) as long as its epoch counter $trained\_epochs_{\mathcal{E}_\phi}$ is less than $iternum_{\mathcal{E}_\phi}$. After each update, the counter is incremented by 1. Once the autoencoder's counter reaches $iternum_{\mathcal{E}_\phi}$, the generator is updated (lines 11-16) until its counter $trained\_epochs_{G_\theta}$ reaches $iternum_{G_\theta}$. When both counters reach their limits, they are reset to 0 (lines 14-16), and the process repeats.

## B Experimental Appendix

In this Appendix, we extend Section 5 by including further information about our experimental settings, extra images and tables from Section 5.3, and further experiments with extra data types.

### B.1 One-class Classification (Extended)

In Section 5.3, we compared **V**-GAN with other subspace selection, embedding, and feature selection methods. We now introduce the exact default values employed for each of them, as well as specific information regarding their implementation.

**CAE (Balın et al., 2019).** We followed the original CAE implementation by selecting $K = 20$ features, fixing a start and minimum temperature of 10 and 0.1, respectively, and 300 epochs with a tryout limit of

Table 5: Results of the Conover-Iman test for the rankings against baselines on non-myopic datasets

| | LOF | | | kNN | | | ECOD | | | COPOD | | | CBLOF | | |
|---|---|---|---|---|---|---|---|---|---|---|---|---|---|---|---|
| | FB | None | **V-GAN** | FB | None | **V-GAN** | FB | None | **V-GAN** | FB | None | **V-GAN** | FB | None | **V-GAN** |
| FB | | + | + | | | | | | | | | | | + | |
| None | - - | | - - | | | | | | | | | | - | | - - |
| **V-GAN** | + | + | | | | | | | | | | | | ++ | |

5. The architecture of the network, optimizer, and default learning rate were taken as-is from their official implementation.

**HiCS (Keller et al., 2012).** We used the only official implementation of HiCS available, together with their recommended parameters. In particular, we used 100 runs with 500 subspace candidates and kept a critical value for the test statistic $\alpha = 0.10$. We did use a different amount of output subspaces, 500, to keep it consistent with what **V**-GAN uses. Additionally, we added direct Python support to their source code — originally in `Nim`. The compiled binary is available for download in our code repository.

**CLIQUE (Agrawal et al., 2005).** We used the only readily available implementation of the algorithm in Python.[8] In our experiments we employed the default values of $\xi = 3$ and $\tau = 0.1$.

**ELM (Xu et al., 2023).** We used the Extreme Learning Machines that perform the dimensionality reduction for Deep Isolation Forests as another competitor, due to the popularity and similarity of the method. In particular, we used the default architecture and parameters from their implementation in `pyod`. This is 50 ensemble members, a hyperbolic tangent activation layer, and a representation space of dimensionality 20.

**GMD (Trittenbach and Böhm, 2019).** We employed the only readily available implementation of GMD online[9]. We employed the default parameters of $\alpha = 0.1$ and 100 runs.

**PCA (Maćkiewicz and Ratajczak, 1993).** We used the implementation of PCA available in `sklearn` (Pedregosa et al., 2011). For reducing the dimensionality of the data, we selected the components with the largest share of variability, until reaching 90%.

**UMAP (Healy and McInnes, 2024).** We employed the implementation provided in their official package. We chose 15 neighbors as recommended by the authors. As for the dimensionality of the underlying manifold, the authors recommend using between 10 and 100 for downstream machine learning tasks. As the dimensionality of our datasets $\mathcal{D}$ varies, we opted to use $\min\left\{\frac{\dim(\mathcal{D})}{2}, 100\right\}$.

Additionally, we included the figures and tables from the non-myopic case from our experiments. Figure 9 contains the boxplots, and Table 6 the Conover-Iman test for the baselines. Figure 10 and Table 5 contain both the boxplots and the Conover-Iman test for the competitors, respectively. We also included the $p$-values of all the Conover-Iman tests, in Tables 1-1. To finalize, we included the raw AUC results in Tables 11-15. We fixed a 5-hour time-out per repetition of the subspace search experiment, denoted by **OT**. Additionally, if the Outlier Detection Method employed failed to report any results due to an implementation error, we reported **ERR**. We excluded results with errors from the ODM during the Conover-Iman test analysis, but treated time-outs as 0 AUC values when calculating the ranks. The last column of the tables contains the results of the Myopicity test with the derived operator. A 1 signifies that the $p$-value of the test statistic was larger than 0.10 — i.e., that the distribution is myopic — and 0 signifies the contrary.

## B.2 Other Data types

This section aims to exemplify the flexibility of the Myopic Subspace Theory (MST) to adapt to different data types. In particular, we will present preliminary experiments for both Images and Natural Language in

---

[8]https://github.com/georgekatona/Clique
[9]https://github.com/andersonvaf/gmd

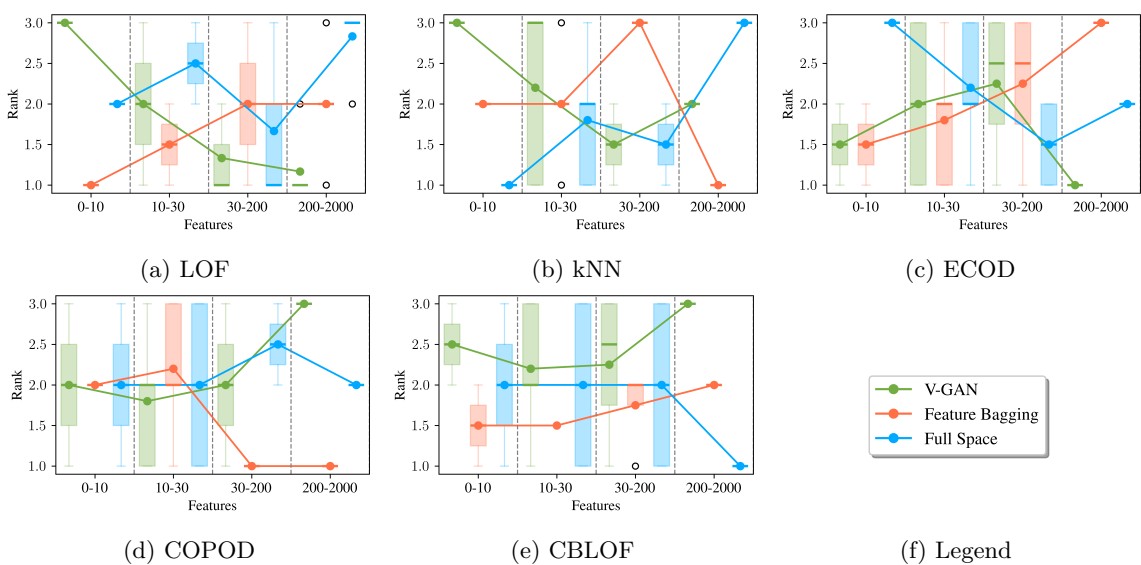

Figure 9: Boxplots of ranks of the comparison with baselines using non-myopic datasets across features

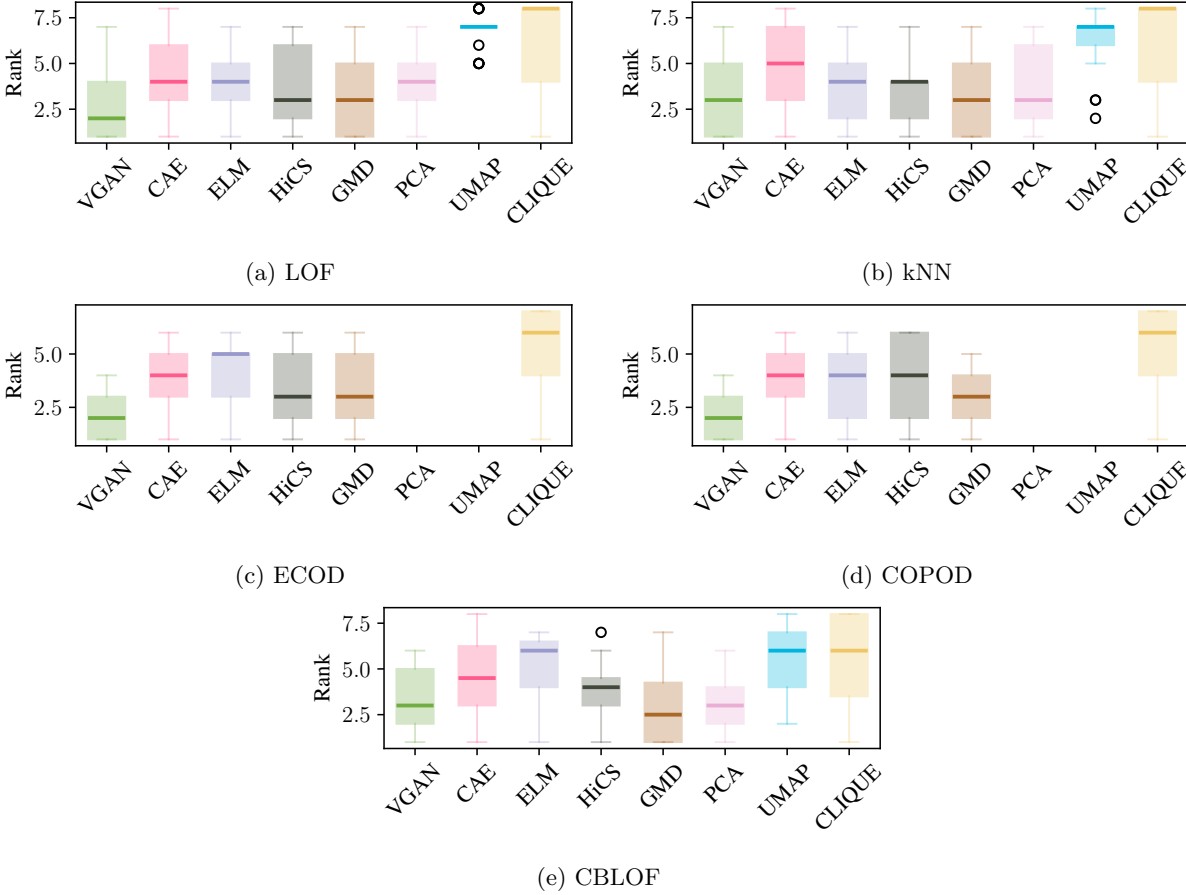

Figure 10: Boxplots of ranks of the comparison with our competitors using non-myopic datasets.

Table 6: Results of the Conover-Iman Test for the rankings against competitors on non-myopic Datasets

| OD method | SS method | CAE | HiCS | CLIQUE | ELM | GMD | PCA | UMAP | V-GAN |
|---|---|---|---|---|---|---|---|---|---|
| LOF | CAE | | | ++ | | - | | ++ | -- |
| | HiCS | | | ++ | | | | ++ | |
| | CLIQUE | -- | -- | | | -- | -- | -- | -- |
| | ELM | | | ++ | | | | ++ | - |
| | GMD | + | | ++ | | | | ++ | |
| | PCA | | | ++ | | | | ++ | - |
| | UMAP | -- | -- | | -- | -- | -- | | -- |
| | **V-GAN** | ++ | | ++ | + | | + | ++ | |
| kNN | CAE | | -- | ++ | - | -- | -- | + | -- |
| | HiCS | ++ | | ++ | | | | ++ | |
| | CLIQUE | -- | -- | | -- | -- | -- | | -- |
| | ELM | + | | ++ | | | | ++ | -- |
| | GMD | ++ | | ++ | | | | ++ | |
| | PCA | ++ | | ++ | | | | ++ | |
| | UMAP | - | -- | | -- | -- | -- | | -- |
| | **V-GAN** | ++ | | ++ | | | | ++ | |
| ECOD | CAE | | | ++ | | - | | | -- |
| | HiCS | | | ++ | | | | | -- |
| | CLIQUE | -- | -- | | | -- | | | -- |
| | ELM | | | | | -- | | | -- |
| | GMD | + | | ++ | ++ | | | | |
| | PCA | | | | | | | | |
| | UMAP | | | | | | | | |
| | **V-GAN** | ++ | ++ | ++ | ++ | | | | |
| COPOD | CAE | | | ++ | | - | | | -- |
| | HiCS | | | ++ | | - | | | -- |
| | CLIQUE | -- | -- | | -- | -- | | | -- |
| | ELM | | | ++ | | | | | -- |
| | GMD | ++ | + | ++ | | | | | |
| | PCA | | | | | | | | |
| | UMAP | | | | | | | | |
| | **V-GAN** | ++ | ++ | ++ | ++ | | | | |
| CBLOF | CAE | | | | | -- | -- | | - |
| | HiCS | | | ++ | ++ | | | ++ | |
| | CLIQUE | | | | -- | -- | -- | | -- |
| | ELM | | | | -- | -- | -- | | -- |
| | GMD | ++ | | ++ | ++ | | | ++ | |
| | PCA | ++ | | ++ | ++ | | | ++ | |
| | UMAP | | | | -- | -- | -- | | -- |
| | **V-GAN** | + | | ++ | ++ | | | ++ | |

Table 7: *p*-values of the Conover-Iman Test for Rankings against baselines on myopic Datasets

| | LOF | | | kNN | | | ECOD | | | COPOD | | | CBLOF | | |
|---|---|---|---|---|---|---|---|---|---|---|---|---|---|---|---|
| | FB | None | **V-GAN** | FB | None | **V-GAN** | FB | None | **V-GAN** | FB | None | **V-GAN** | FB | None | **V-GAN** |
| FB | | 0.00 | 0.00 | | 0.84 | 0.01 | | 0.41 | 0.04 | | 0.83 | 0.00 | | 0.42 | 0.00 |
| None | 0.00 | | 0.00 | 0.84 | | 0.01 | 0.41 | | 0.01 | 0.83 | | 0.00 | 0.42 | | 0.00 |
| **V-GAN** | 0.00 | 0.00 | | 0.01 | 0.01 | | 0.04 | 0.01 | | 0.00 | 0.00 | | 0.00 | 0.00 | |

Table 8: *p*-values of the Conover-Iman Test for Rankings against baselines on non-myopic Datasets

| | LOF | | | kNN | | | ECOD | | | COPOD | | | CBLOF | | |
|---|---|---|---|---|---|---|---|---|---|---|---|---|---|---|---|
| | FB | None | **V-GAN** | FB | None | **V-GAN** | FB | None | **V-GAN** | FB | None | **V-GAN** | FB | None | **V-GAN** |
| FB | | 0.06 | 0.80 | | | | | | | | | | | 0.06 | 0.80 |
| None | 0.06 | | 0.05 | | | | | | | | | | 0.06 | | 0.05 |
| **V-GAN** | 0.80 | 0.05 | | | | | | | | | | | 0.80 | 0.05 | |

Table 9: $p$-values of the Conover-Iman Test for Rankings against competitors on myopic Datasets

| OD Method | SS method | CAE | HiCS | CLIQUE | ELM | GMD | PCA | UMAP | **V-GAN** |
|---|---|---|---|---|---|---|---|---|---|
| LOF | CAE | | 0.25 | 0.00 | 0.68 | 0.46 | 0.87 | 0.00 | 0.00 |
| | HiCS | 0.25 | | 0.00 | 0.46 | 0.68 | 0.19 | 0.00 | 0.04 |
| | CLIQUE | 0.00 | 0.00 | | 0.00 | 0.00 | 0.00 | 0.36 | 0.00 |
| | ELM | 0.68 | 0.46 | 0.00 | | 0.74 | 0.56 | 0.00 | 0.01 |
| | GMD | 0.46 | 0.68 | 0.00 | 0.74 | | 0.36 | 0.00 | 0.01 |
| | PCA | 0.87 | 0.19 | 0.00 | 0.56 | 0.36 | | 0.00 | 0.00 |
| | UMAP | 0.00 | 0.00 | 0.36 | 0.00 | 0.00 | 0.00 | | 0.00 |
| | **V-GAN** | 0.00 | 0.04 | 0.00 | 0.01 | 0.01 | 0.00 | 0.00 | |
| kNN | CAE | | 0.59 | 0.00 | 0.02 | 0.18 | 0.06 | 0.00 | 0.00 |
| | HiCS | 0.59 | | 0.00 | 0.06 | 0.42 | 0.18 | 0.00 | 0.00 |
| | CLIQUE | 0.00 | 0.00 | | 0.00 | 0.00 | 0.00 | 0.86 | 0.00 |
| | ELM | 0.02 | 0.06 | 0.00 | | 0.28 | 0.59 | 0.00 | 0.02 |
| | GMD | 0.18 | 0.42 | 0.00 | 0.28 | | 0.59 | 0.00 | 0.00 |
| | PCA | 0.06 | 0.18 | 0.00 | 0.59 | 0.59 | | 0.00 | 0.00 |
| | UMAP | 0.00 | 0.00 | 0.86 | 0.00 | 0.00 | 0.00 | | 0.00 |
| | **V-GAN** | 0.00 | 0.00 | 0.00 | 0.02 | 0.00 | 0.00 | 0.00 | |
| ECOD | CAE | | 0.01 | 0.04 | 0.06 | 0.00 | | | 0.00 |
| | HiCS | 0.01 | | 0.00 | 0.38 | 0.86 | | | 0.02 |
| | CLIQUE | 0.04 | 0.00 | | 0.00 | 0.00 | | | 0.00 |
| | ELM | 0.06 | 0.38 | 0.00 | | 0.29 | | | 0.00 |
| | GMD | 0.00 | 0.86 | 0.00 | 0.29 | | | | 0.03 |
| | PCA | | | | | | | | |
| | UMAP | | | | | | | | |
| | **V-GAN** | 0.00 | 0.02 | 0.00 | 0.00 | 0.03 | | | |
| COPOD | CAE | | 0.03 | 0.00 | 0.21 | 0.01 | | | 0.00 |
| | HiCS | 0.03 | | 0.00 | 0.38 | 0.70 | | | 0.03 |
| | CLIQUE | 0.00 | 0.00 | | 0.00 | 0.00 | | | 0.00 |
| | ELM | 0.21 | 0.38 | 0.00 | | 0.21 | | | 0.00 |
| | GMD | 0.01 | 0.70 | 0.00 | 0.21 | | | | 0.07 |
| | PCA | | | | | | | | |
| | UMAP | | | | | | | | |
| | **V-GAN** | 0.00 | 0.03 | 0.00 | 0.00 | 0.07 | | | |
| CBLOF | CAE | | 0.43 | 0.02 | 0.11 | 0.12 | 0.14 | 0.10 | 0.00 |
| | HiCS | 0.43 | | 0.00 | 0.02 | 0.49 | 0.57 | 0.02 | 0.02 |
| | CLIQUE | 0.02 | 0.00 | | 0.47 | 0.00 | 0.00 | 0.41 | 0.00 |
| | ELM | 0.11 | 0.02 | 0.47 | | 0.00 | 0.00 | 0.95 | 0.00 |
| | GMD | 0.12 | 0.49 | 0.00 | 0.00 | | 0.90 | 0.00 | 0.06 |
| | PCA | 0.14 | 0.57 | 0.00 | 0.00 | 0.90 | | 0.00 | 0.05 |
| | UMAP | 0.10 | 0.02 | 0.41 | 0.95 | 0.00 | 0.00 | | 0.00 |
| | **V-GAN** | 0.00 | 0.02 | 0.00 | 0.00 | 0.06 | 0.05 | 0.00 | |

Table 10: $p$-values of the Conover-Iman Test for Rankings against competitors on non-myopic Datasets

| OD Method | SS method | CAE | HiCS | CLIQUE | ELM | GMD | PCA | UMAP | **V-GAN** |
|---|---|---|---|---|---|---|---|---|---|
| LOF | CAE | | 0.19 | 0.01 | 0.40 | 0.06 | 0.44 | 0.00 | 0.01 |
| | HiCS | 0.19 | | 0.00 | 0.64 | 0.54 | 0.59 | 0.00 | 0.18 |
| | CLIQUE | 0.01 | 0.00 | | 0.00 | 0.00 | 0.00 | 0.27 | 0.00 |
| | ELM | 0.40 | 0.64 | 0.00 | | 0.28 | 0.94 | 0.00 | 0.07 |
| | GMD | 0.06 | 0.54 | 0.00 | 0.28 | | 0.25 | 0.00 | 0.46 |
| | PCA | 0.44 | 0.59 | 0.00 | 0.94 | 0.25 | | 0.00 | 0.06 |
| | UMAP | 0.00 | 0.00 | 0.27 | 0.00 | 0.00 | 0.00 | | 0.00 |
| | **V-GAN** | 0.01 | 0.18 | 0.00 | 0.07 | 0.46 | 0.06 | 0.00 | |
| kNN | CAE | | 0.03 | 0.05 | 0.06 | 0.00 | 0.05 | 0.08 | 0.00 |
| | HiCS | 0.03 | | 0.00 | 0.83 | 0.40 | 0.88 | 0.00 | 0.48 |
| | CLIQUE | 0.05 | 0.00 | | 0.00 | 0.00 | 0.00 | 0.81 | 0.00 |
| | ELM | 0.06 | 0.83 | 0.00 | | 0.29 | 0.94 | 0.00 | 0.36 |
| | GMD | 0.00 | 0.40 | 0.00 | 0.29 | | 0.32 | 0.00 | 0.88 |
| | PCA | 0.05 | 0.88 | 0.00 | 0.94 | 0.32 | | 0.00 | 0.40 |
| | UMAP | 0.08 | 0.00 | 0.81 | 0.00 | 0.00 | 0.00 | | 0.00 |
| | **V-GAN** | 0.00 | 0.48 | 0.00 | 0.36 | 0.88 | 0.40 | 0.00 | |
| ECOD | CAE | | 0.29 | 0.04 | 0.57 | 0.07 | | | 0.00 |
| | HiCS | 0.29 | | 0.00 | 0.10 | 0.45 | | | 0.02 |
| | CLIQUE | 0.04 | 0.00 | | 0.12 | 0.00 | | | 0.00 |
| | ELM | 0.57 | 0.10 | 0.12 | | 0.02 | | | 0.00 |
| | GMD | 0.07 | 0.45 | 0.00 | 0.02 | | | | 0.13 |
| | PCA | | | | | | | | |
| | UMAP | | | | | | | | |
| | **V-GAN** | 0.00 | 0.02 | 0.00 | 0.00 | 0.13 | | | |
| COPOD | CAE | | 0.85 | 0.02 | 0.57 | 0.05 | | | 0.00 |
| | HiCS | 0.85 | | 0.01 | 0.71 | 0.07 | | | 0.00 |
| | CLIQUE | 0.02 | 0.01 | | 0.00 | 0.00 | | | 0.00 |
| | ELM | 0.57 | 0.71 | 0.00 | | 0.16 | | | 0.01 |
| | GMD | 0.05 | 0.07 | 0.00 | 0.16 | | | | 0.22 |
| | PCA | | | | | | | | |
| | UMAP | | | | | | | | |
| | **V-GAN** | 0.00 | 0.00 | 0.00 | 0.01 | 0.22 | | | |
| CBLOF | CAE | | 0.27 | 0.13 | 0.22 | 0.03 | 0.04 | 0.11 | 0.06 |
| | HiCS | 0.27 | | 0.01 | 0.02 | 0.27 | 0.38 | 0.01 | 0.41 |
| | CLIQUE | 0.13 | 0.01 | | 0.75 | 0.00 | 0.00 | 0.98 | 0.00 |
| | ELM | 0.22 | 0.02 | 0.75 | | 0.00 | 0.00 | 0.73 | 0.00 |
| | GMD | 0.03 | 0.27 | 0.00 | 0.00 | | 0.81 | 0.00 | 0.84 |
| | PCA | 0.04 | 0.38 | 0.00 | 0.00 | 0.81 | | 0.00 | 0.99 |
| | UMAP | 0.11 | 0.01 | 0.98 | 0.73 | 0.00 | 0.00 | | 0.00 |
| | **V-GAN** | 0.06 | 0.41 | 0.00 | 0.00 | 0.84 | 0.99 | 0.00 | |

Table 11: Full table of raw AUCs from Section 5.3 using LOF. We denoted time-outs by **OT**, and ODM errors by **ERR**.

| Dataset | Features | Feature Bagging | Full Space | CAE | CLIQUE | HiCS | GMD | PCA | UMAP | ELM | V-GAN | Myopicity |
|---|---|---|---|---|---|---|---|---|---|---|---|---|
| InternetAds | 1555 | 0.8613 | 0.7757 | 0.5154 | OT | 0.8300 | 0.8127 | 0.4632 | 0.4546 | 0.7642 | 0.8211 | 0 |
| 20news | 768 | 0.7867 | 0.7861 | 0.6613 | OT | 0.5463 | 0.6609 | 0.6382 | 0.5219 | 0.7647 | 0.7902 | 1 |
| Agnews | 768 | 0.6920 | 0.6900 | 0.6336 | OT | 0.4863 | 0.5239 | 0.5489 | 0.5280 | 0.6841 | 0.6971 | 0 |
| Amazon | 768 | 0.5975 | 0.5975 | 0.5880 | OT | 0.5071 | 0.6302 | 0.5952 | 0.5226 | 0.5777 | 0.5998 | 0 |
| Imdb | 768 | 0.5433 | 0.5433 | 0.5555 | OT | 0.5086 | 0.5905 | 0.5720 | 0.5058 | 0.5345 | 0.5443 | 0 |
| Yelp | 768 | 0.6792 | 0.6782 | 0.6151 | OT | 0.5198 | 0.6704 | 0.6443 | 0.5196 | 0.6584 | 0.6825 | 0 |
| CIFAR10 | 512 | 0.7389 | 0.7328 | 0.6122 | OT | 0.6624 | 0.6492 | 0.7449 | 0.5942 | 0.7415 | 0.7491 | 1 |
| FashionMNIST | 512 | 0.8891 | 0.8869 | 0.8636 | OT | 0.7863 | 0.9227 | 0.9048 | 0.8570 | 0.8923 | 0.8978 | 1 |
| MNIST-C | 512 | 0.9757 | 0.9752 | 0.8736 | OT | 0.6405 | 0.6187 | 0.7066 | 0.5099 | 0.9288 | 0.9762 | 1 |
| MVTec-AD | 512 | 0.9704 | 0.9700 | 0.9624 | OT | 0.9275 | 0.7539 | 0.8635 | 0.5969 | 0.9543 | 0.9713 | 0 |
| SVHN | 512 | 0.7168 | 0.7151 | 0.6474 | OT | 0.5891 | 0.5795 | 0.6981 | 0.5184 | 0.6766 | 0.7175 | 1 |
| speech | 400 | 0.5513 | 0.5243 | 0.5995 | OT | 0.5390 | 0.5812 | 0.3882 | 0.5325 | 0.5704 | 0.5895 | 1 |
| musk | 166 | 1.0000 | 1.0000 | 0.9988 | OT | 1.0000 | 1.0000 | 1.0000 | 0.6175 | 1.0000 | 1.0000 | 0 |
| mnist | 100 | 0.9681 | 0.9664 | 0.8394 | OT | 0.6373 | 0.7369 | 0.9126 | 0.5523 | 0.9638 | 0.9733 | 1 |
| optdigits | 64 | 0.9984 | 0.9988 | 0.9964 | OT | 0.5057 | 0.6032 | 0.9996 | 0.6970 | 0.9802 | 0.9986 | 0 |
| SpamBase | 57 | 0.4305 | 0.3608 | 0.3619 | OT | 0.8251 | 0.7897 | 0.3315 | 0.4732 | 0.5017 | 0.4901 | 0 |
| landsat | 36 | 0.7785 | 0.7744 | 0.7810 | OT | 0.8208 | 0.7447 | 0.7880 | 0.5257 | 0.6925 | 0.7792 | 1 |
| satellite | 36 | 0.8664 | 0.8686 | 0.8591 | OT | 0.8075 | 0.8148 | 0.8489 | 0.6293 | 0.5741 | 0.8702 | 1 |
| satimage-2 | 36 | 0.9972 | 0.9968 | 0.9926 | OT | 0.9972 | 0.9977 | 0.9970 | 0.5501 | 0.9969 | 0.9974 | 1 |
| Ionosphere | 33 | 0.9311 | 0.9287 | 0.9190 | OT | 0.9376 | 0.5723 | 0.5227 | 0.4699 | 0.8865 | 0.9327 | 0 |
| WPBC | 33 | 0.5304 | 0.5326 | 0.4433 | OT | 0.5950 | 0.9384 | 0.8379 | 0.5290 | 0.5011 | 0.5370 | 0 |
| letter | 32 | 0.9221 | 0.9093 | 0.8523 | OT | 0.7813 | 0.7609 | 0.8387 | 0.6419 | 0.9207 | 0.9286 | 1 |
| WDBC | 30 | 0.9958 | 0.9944 | 0.9845 | OT | 1.0000 | 0.9915 | 0.9944 | 0.8313 | 0.9799 | 0.9956 | 0 |
| fault | 27 | 0.6647 | 0.6516 | 0.6378 | OT | 0.6107 | 0.6011 | 0.6312 | 0.5320 | 0.6316 | 0.6578 | 1 |
| annthyroid | 21 | 0.9357 | 0.8752 | 0.6605 | 0.9593 | 0.9629 | 0.7895 | 0.8107 | 0.5842 | 0.7229 | 0.9273 | 1 |
| cardio | 21 | 0.6727 | 0.6135 | 0.7542 | OT | 0.8836 | 0.8517 | 0.6721 | 0.5587 | 0.6425 | 0.6827 | 1 |
| Cardiotocography | 21 | 0.8004 | 0.7818 | 0.7275 | OT | 0.7638 | 0.8136 | 0.3545 | 0.6144 | 0.7033 | 0.8107 | 1 |
| Waveform | 21 | 0.8157 | 0.8131 | 0.6935 | OT | 0.8173 | 0.7599 | 0.7317 | 0.6614 | 0.8292 | 0.8173 | 1 |
| Hepatitis | 19 | 0.6432 | 0.6272 | 0.6923 | OT | 0.8107 | 0.8757 | 0.6331 | 0.4615 | 0.5083 | 0.6473 | 0 |
| Lymphography | 18 | 0.9756 | 0.9821 | 0.9018 | OT | 0.9524 | 0.9524 | 0.9464 | 0.7226 | 0.9643 | 0.9994 | 0 |
| pendigits | 16 | 0.9962 | 0.9916 | 0.8698 | OT | 0.9825 | 0.9698 | 0.9871 | 0.6882 | 0.9945 | 0.9699 | 1 |
| wine | 13 | 0.9737 | 0.9708 | 0.9792 | 0.9792 | 0.9625 | 0.9708 | 0.9708 | 0.7038 | 0.8008 | 0.9771 | 0 |
| vowels | 12 | 0.9653 | 0.9610 | 0.6538 | 0.9285 | 0.9196 | 0.9446 | 0.8443 | 0.6826 | 0.9639 | 0.9447 | 0 |
| PageBlocks | 10 | 0.9645 | 0.9411 | 0.9432 | 0.9789 | 0.9767 | 0.9708 | 0.9339 | 0.5048 | 0.7374 | 0.9662 | 1 |
| breastw | 9 | 0.8019 | 0.8059 | 0.7698 | 0.8391 | 0.8619 | 0.6721 | 0.7349 | 0.4886 | 0.7599 | 0.8726 | 0 |
| Stamps | 9 | 0.9802 | 0.9662 | 0.9136 | 0.9886 | 0.9579 | 0.9818 | 0.9287 | 0.5517 | 0.8256 | 0.9760 | 0 |
| WBC | 9 | 0.7370 | 0.7047 | 0.7372 | 0.7326 | 0.6884 | 0.8934 | 0.8453 | 0.5833 | 0.6912 | 0.5921 | 0 |
| Pima | 8 | 0.6422 | 0.6343 | 0.4597 | 0.6530 | 0.6365 | 0.6224 | 0.6253 | 0.5863 | 0.6225 | 0.6103 | 0 |
| yeast | 8 | 0.5097 | 0.4734 | 0.5299 | 0.5810 | 0.5464 | 0.5779 | 0.4715 | 0.5231 | 0.5186 | 0.5604 | 1 |
| thyroid | 6 | 0.9357 | 0.8752 | 0.6613 | 0.9593 | 0.9629 | 0.8493 | 0.6721 | 0.5587 | 0.7229 | 0.9173 | 1 |
| vertebral | 6 | 0.4958 | 0.4873 | 0.4915 | 0.4952 | 0.4944 | 0.4786 | 0.4841 | 0.4293 | 0.5696 | 0.4612 | 0 |
| Wilt | 5 | 0.8907 | 0.9054 | 0.7215 | 0.8728 | 0.8565 | 0.8398 | 0.5836 | 0.5540 | 0.7306 | 0.6122 | 1 |

Table 12: Full table of raw AUCs from Section 5.3 using kNN. We denoted time-outs by **OT**, and ODM errors by **ERR**.

| Dataset | Features | Feature Bagging | Full Space | CAE | CLIQUE | HiCS | GMD | PCA | UMAP | ELM | V-GAN | Myopicity |
|---|---|---|---|---|---|---|---|---|---|---|---|---|
| InternetAds | 1555 | 0.8914 | 0.8801 | 0.6037 | OT | 0.8716 | 0.8814 | 0.5578 | 0.6910 | 0.8403 | 0.7420 | 0 |
| 20news | 768 | 0.6852 | 0.6847 | 0.6659 | OT | 0.5213 | 0.6793 | 0.6120 | 0.4779 | 0.6819 | 0.6860 | 1 |
| Agnews | 768 | 0.5689 | 0.5696 | 0.5092 | OT | 0.4663 | 0.5304 | 0.5310 | 0.5286 | 0.5569 | 0.5650 | 0 |
| Amazon | 768 | 0.6019 | 0.6019 | 0.5717 | OT | 0.5328 | 0.5533 | 0.5303 | 0.5735 | 0.5860 | 0.6024 | 0 |
| Imdb | 768 | 0.5288 | 0.5285 | 0.5526 | OT | 0.5177 | 0.5898 | 0.5818 | 0.4757 | 0.5309 | 0.5294 | 0 |
| Yelp | 768 | 0.6668 | 0.6668 | 0.6019 | OT | 0.5526 | 0.6466 | 0.6398 | 0.4890 | 0.6448 | 0.6676 | 0 |
| CIFAR10 | 512 | 0.7898 | 0.7882 | 0.6933 | OT | 0.7037 | 0.7854 | 0.8621 | 0.5670 | 0.7671 | 0.7913 | 1 |
| FashionMNIST | 512 | 0.9169 | 0.9161 | 0.8400 | OT | 0.8477 | 0.9031 | 0.9717 | 0.3872 | 0.9035 | 0.9171 | 1 |
| MNIST-C | 512 | 0.8960 | 0.8958 | 0.7859 | OT | 0.6464 | 0.7432 | 0.7812 | 0.5003 | 0.8568 | 0.8993 | 1 |
| MVTec-AD | 512 | 0.9748 | 0.9745 | 0.9658 | OT | 0.9279 | 0.8649 | 0.9083 | 0.5747 | 0.9666 | 0.9745 | 0 |
| SVHN | 512 | 0.7219 | 0.7205 | 0.6407 | OT | 0.6034 | 0.5880 | 0.7149 | 0.5700 | 0.6984 | 0.7220 | 1 |
| speech | 400 | 0.6570 | 0.6503 | 0.5486 | OT | 0.5346 | 0.6664 | 0.5636 | 0.6075 | 0.5857 | 0.6745 | 1 |
| musk | 166 | 1.0000 | 1.0000 | 1.0000 | OT | 1.0000 | 1.0000 | 1.0000 | 0.6791 | 1.0000 | 1.0000 | 0 |
| mnist | 100 | 0.9771 | 0.9779 | 0.8867 | OT | 0.8349 | 0.8685 | 0.9489 | 0.4804 | 0.9618 | 0.9780 | 1 |
| optdigits | 64 | 0.9979 | 0.9981 | 0.9975 | OT | 0.5289 | 0.9077 | 0.9994 | 0.7617 | 0.9844 | 0.9973 | 0 |
| SpamBase | 57 | 0.3579 | 0.3584 | 0.3758 | OT | 0.7329 | 0.4295 | 0.3511 | 0.4495 | 0.3057 | 0.3601 | 0 |
| landsat | 36 | 0.7850 | 0.7842 | 0.7815 | OT | 0.7824 | 0.7624 | 0.8215 | 0.5057 | 0.7715 | 0.7857 | 1 |
| satellite | 36 | 0.8290 | 0.8295 | 0.8304 | OT | 0.7941 | 0.8153 | 0.8243 | 0.5704 | 0.7882 | 0.8291 | 1 |
| satimage-2 | 36 | 0.9993 | 0.9993 | 0.9992 | OT | 0.9983 | 0.9981 | 0.9990 | 0.1325 | 0.9993 | 0.9993 | 1 |
| Ionosphere | 33 | 0.9767 | 0.9771 | 0.9690 | OT | 0.9646 | 0.5702 | 0.5298 | 0.4509 | 0.9786 | 0.9777 | 0 |
| WPBC | 33 | 0.5091 | 0.5191 | 0.4801 | OT | 0.5709 | 0.9621 | 0.8760 | 0.6175 | 0.4955 | 0.5128 | 0 |
| letter | 32 | 0.9350 | 0.9305 | 0.9261 | OT | 0.8602 | 0.8608 | 0.8989 | 0.6701 | 0.9344 | 0.9364 | 1 |
| WDBC | 30 | 0.9742 | 0.9718 | 0.9338 | OT | 0.9958 | 0.9930 | 0.9803 | 0.4587 | 0.9168 | 0.9756 | 0 |
| fault | 27 | 0.8105 | 0.8093 | 0.7886 | OT | 0.7169 | 0.7058 | 0.7854 | 0.6394 | 0.8034 | 0.8098 | 1 |
| annthyroid | 21 | 0.7092 | 0.7681 | 0.5823 | 0.7954 | 0.8065 | 0.8263 | 0.8064 | 0.4749 | 0.6669 | 0.7167 | 1 |
| cardio | 21 | 0.8153 | 0.6976 | 0.5419 | OT | 0.8378 | 0.6687 | 0.6091 | 0.4294 | 0.6734 | 0.8166 | 1 |
| Cardiotocography | 21 | 0.7816 | 0.7187 | 0.7598 | OT | 0.7502 | 0.7903 | 0.4990 | 0.5295 | 0.6553 | 0.8087 | 1 |
| Waveform | 21 | 0.7190 | 0.8040 | 0.7220 | OT | 0.8592 | 0.7457 | 0.6822 | 0.4876 | 0.6996 | 0.7264 | 1 |
| Hepatitis | 19 | 0.7284 | 0.7337 | 0.6331 | OT | 0.7811 | 0.7929 | 0.5799 | 0.5959 | 0.6314 | 0.6580 | 0 |
| Lymphography | 18 | 0.9577 | 0.9583 | 0.7321 | OT | 0.9583 | 0.9524 | 0.9583 | 0.7399 | 0.9548 | 0.9804 | 0 |
| pendigits | 16 | 0.9987 | 0.9988 | 0.9805 | OT | 0.9943 | 0.9903 | 0.9991 | 0.6486 | 0.9969 | 0.9905 | 1 |
| wine | 13 | 0.9396 | 0.9417 | 0.9542 | 0.9417 | 0.9417 | 0.9708 | 0.9625 | 0.2663 | 0.4542 | 0.9337 | 0 |
| vowels | 12 | 0.9836 | 0.9806 | 0.7430 | 0.9572 | 0.9679 | 0.9717 | 0.9163 | 0.5841 | 0.9803 | 0.9572 | 0 |
| PageBlocks | 10 | 0.9720 | 0.9805 | 0.9646 | 0.9738 | 0.9547 | 0.9729 | 0.9665 | 0.2513 | 0.9549 | 0.9592 | 1 |
| breastw | 9 | 0.9876 | 0.9245 | 0.8457 | 0.9571 | 0.9644 | 0.9140 | 0.7860 | 0.6814 | 0.9066 | 0.9830 | 0 |
| Stamps | 9 | 0.8270 | 0.9880 | 0.9240 | 0.9886 | 0.9631 | 0.9657 | 0.9693 | 0.6981 | 0.9627 | 0.8640 | 0 |
| WBC | 9 | 0.9415 | 0.7814 | 0.6814 | 0.8767 | 0.9186 | 0.9735 | 0.9245 | 0.7544 | 0.7995 | 0.9686 | 0 |
| Pima | 8 | 0.5518 | 0.5593 | 0.4660 | 0.5431 | 0.5249 | 0.4781 | 0.5340 | 0.4849 | 0.5213 | 0.5151 | 0 |
| yeast | 8 | 0.5834 | 0.5871 | 0.5192 | 0.5710 | 0.5606 | 0.5787 | 0.5883 | 0.5355 | 0.5955 | 0.5833 | 1 |
| thyroid | 6 | 0.7816 | 0.7681 | 0.5823 | 0.7954 | 0.8065 | 0.7883 | 0.6091 | 0.4294 | 0.6669 | 0.8087 | 1 |
| vertebral | 6 | 0.5260 | 0.5333 | 0.4770 | 0.5278 | 0.5254 | 0.4913 | 0.5611 | 0.4828 | 0.5293 | 0.5214 | 0 |
| Wilt | 5 | 0.8434 | 0.8620 | 0.6910 | 0.8257 | 0.8493 | 0.7971 | 0.6548 | 0.5643 | 0.8578 | 0.7216 | 1 |

Table 13: Full table of raw AUCs from Section 5.3 using ECOD. We denoted time-outs by **OT**, and ODM errors by **ERR**.

| Dataset | Features | Feature Bagging | Full Space | CAE | CLIQUE | HiCS | GMD | PCA | UMAP | ELM | V-GAN | Myopicity |
|---|---|---|---|---|---|---|---|---|---|---|---|---|
| InternetAds | 1555 | 0.6994 | 0.6990 | 0.3337 | OT | 0.7746 | 0.7114 | ERR | 0.2977 | 0.3670 | 0.6350 | 0 |
| 20news | 768 | 0.6547 | 0.6547 | 0.6032 | OT | 0.4776 | 0.6198 | 0.6552 | 0.5953 | 0.6510 | 0.6556 | 1 |
| Agnews | 768 | 0.5089 | 0.5090 | 0.4934 | OT | 0.4646 | 0.5093 | ERR | 0.5182 | 0.4916 | 0.5094 | 0 |
| Amazon | 768 | 0.5635 | 0.5635 | 0.5604 | OT | 0.5015 | 0.4939 | ERR | 0.5065 | 0.5675 | 0.5640 | 0 |
| Imdb | 768 | 0.5052 | 0.5052 | 0.4834 | OT | 0.5130 | 0.5509 | ERR | 0.5334 | 0.4941 | 0.5043 | 0 |
| Yelp | 768 | 0.6023 | 0.6024 | 0.5605 | OT | 0.5023 | 0.5849 | ERR | 0.5488 | 0.5930 | 0.6020 | 0 |
| CIFAR10 | 512 | 0.7228 | 0.7222 | 0.6382 | OT | 0.7027 | 0.6562 | ERR | 0.4936 | 0.7218 | 0.7229 | 1 |
| FashionMNIST | 512 | 0.8215 | 0.8213 | 0.6934 | OT | 0.8216 | 0.9393 | ERR | 0.5920 | 0.8118 | 0.8216 | 1 |
| MNIST-C | 512 | 0.6473 | 0.6470 | 0.5242 | OT | 0.6288 | 0.7073 | ERR | 0.4760 | 0.6364 | 0.6490 | 1 |
| MVTec-AD | 512 | 0.9399 | 0.9403 | 0.9331 | OT | 0.9234 | 0.8127 | ERR | 0.5296 | 0.8566 | 0.9415 | 0 |
| SVHN | 512 | 0.5609 | 0.5608 | 0.4566 | OT | 0.6007 | 0.5634 | ERR | 0.3949 | 0.5232 | 0.5653 | 1 |
| speech | 400 | 0.5451 | 0.5421 | 0.4884 | OT | 0.5325 | 0.5814 | ERR | 0.4251 | 0.5420 | 0.5420 | 1 |
| musk | 166 | 0.8685 | 0.8681 | 0.1663 | OT | 0.0430 | 0.6342 | ERR | 0.4911 | 0.1949 | 0.8683 | 0 |
| mnist | 100 | 0.7476 | 0.7478 | 0.7719 | OT | 0.6536 | 0.7024 | ERR | 0.5676 | 0.8846 | 0.7822 | 1 |
| optdigits | 64 | 0.5611 | 0.5608 | 0.4640 | OT | 0.4910 | 0.6191 | ERR | ERR | 0.3226 | 0.5615 | 0 |
| SpamBase | 57 | 0.5957 | 0.5962 | 0.2578 | OT | 0.8083 | 0.6351 | ERR | ERR | 0.3359 | 0.5943 | 0 |
| landsat | 36 | 0.4304 | 0.4310 | 0.4292 | OT | 0.5084 | 0.4487 | ERR | ERR | 0.3224 | 0.4326 | 1 |
| satellite | 36 | 0.6979 | 0.6983 | 0.6802 | OT | 0.6199 | 0.6801 | ERR | ERR | 0.6694 | 0.6985 | 1 |
| satimage-2 | 36 | 0.9932 | 0.9931 | 0.9949 | OT | 0.9859 | 0.9916 | ERR | ERR | 0.9917 | 0.9932 | 1 |
| Ionosphere | 33 | 0.8406 | 0.8409 | 0.8335 | OT | 0.7944 | 0.4972 | ERR | ERR | 0.8720 | 0.8405 | 0 |
| WPBC | 33 | 0.5090 | 0.5085 | 0.4638 | OT | 0.4801 | 0.8097 | ERR | ERR | 0.4228 | 0.5077 | 0 |
| letter | 32 | 0.5106 | 0.5112 | 0.4830 | OT | 0.4861 | 0.4862 | ERR | ERR | 0.4993 | 0.5107 | 1 |
| WDBC | 30 | 0.9486 | 0.9479 | 0.9324 | OT | 0.9648 | 0.9620 | ERR | ERR | 0.9565 | 0.9477 | 0 |
| fault | 27 | 0.4431 | 0.4430 | 0.4110 | OT | 0.4056 | 0.4269 | ERR | ERR | 0.3800 | 0.4442 | 1 |
| annthyroid | 21 | 0.7514 | 0.7513 | 0.6448 | 0.7527 | 0.7513 | 0.6585 | ERR | ERR | 0.5977 | 0.7521 | 1 |
| cardio | 21 | 0.5101 | 0.5107 | 0.4407 | OT | 0.7397 | 0.6742 | ERR | ERR | 0.1092 | 0.5129 | 1 |
| Cardiotocography | 21 | 0.7369 | 0.7375 | 0.7176 | OT | 0.7322 | 0.7808 | ERR | ERR | 0.6140 | 0.7386 | 1 |
| Waveform | 21 | 0.7132 | 0.7128 | 0.6209 | OT | 0.6922 | 0.7235 | ERR | ERR | 0.7609 | 0.7118 | 1 |
| Hepatitis | 19 | 0.6361 | 0.6391 | 0.6391 | OT | 0.6095 | 0.6272 | ERR | ERR | 0.4450 | 0.6402 | 0 |
| Lymphography | 18 | 0.9327 | 0.9286 | 0.7857 | OT | 0.9643 | 0.9464 | ERR | ERR | 0.9452 | 0.9304 | 0 |
| pendigits | 16 | 0.8690 | 0.8709 | 0.8053 | OT | 0.8996 | 0.8998 | ERR | ERR | 0.8313 | 0.8725 | 1 |
| wine | 13 | 0.8913 | 0.8917 | 0.9292 | 0.8917 | 0.8708 | 0.8417 | ERR | ERR | 0.7492 | 0.8900 | 0 |
| vowels | 12 | 0.4608 | 0.4596 | 0.4044 | 0.4752 | 0.2231 | 0.3682 | ERR | ERR | 0.4261 | 0.4929 | 0 |
| PageBlocks | 10 | 0.9475 | 0.9472 | 0.9342 | 0.9472 | 0.9482 | 0.9356 | ERR | ERR | 0.9633 | 0.9468 | 1 |
| breastw | 9 | 0.7857 | 0.7884 | 0.2853 | 0.7836 | 0.8384 | 0.8953 | ERR | ERR | 0.3482 | 0.7856 | 0 |
| Stamps | 9 | 0.9085 | 0.9095 | 0.8387 | 0.9095 | 0.8715 | 0.8933 | ERR | ERR | 0.6948 | 0.9113 | 0 |
| WBC | 9 | 0.7944 | 0.7977 | 0.6442 | 0.7930 | 0.8349 | 0.8742 | ERR | ERR | 0.4526 | 0.7935 | 0 |
| Pima | 8 | 0.5485 | 0.5474 | 0.5256 | 0.5494 | 0.5422 | 0.5159 | ERR | ERR | 0.4825 | 0.5481 | 0 |
| yeast | 8 | 0.6006 | 0.6016 | 0.4625 | 0.5989 | 0.5843 | 0.5595 | ERR | ERR | 0.3393 | 0.6014 | 1 |
| thyroid | 6 | 0.7514 | 0.7513 | 0.6333 | 0.7527 | 0.7543 | 0.6732 | ERR | ERR | 0.5977 | 0.7501 | 1 |
| vertebral | 6 | 0.5171 | 0.5159 | 0.5774 | 0.5278 | 0.5143 | 0.5167 | ERR | ERR | 0.2838 | 0.5194 | 0 |
| Wilt | 5 | 0.3977 | 0.3970 | 0.3821 | 0.3970 | 0.4068 | 0.4109 | ERR | ERR | 0.4220 | 0.3968 | 1 |

Table 14: Full table of raw AUCs from Section 5.3 using COPOD. We denoted time-outs by **OT**, and ODM errors by **ERR**.

| Dataset | Features | Feature Bagging | Full Space | CAE | CLIQUE | HiCS | GMD | PCA | UMAP | ELM | V-GAN | Myopicity |
|---|---|---|---|---|---|---|---|---|---|---|---|---|
| InternetAds | 1555 | 0.7000 | 0.6994 | 0.4041 | OT | 0.7748 | 0.7117 | ERR | 0.2415 | 0.4269 | 0.6356 | 0 |
| 20news | 768 | 0.6480 | 0.6482 | 0.6107 | OT | 0.4755 | 0.6185 | 0.6030 | 0.5922 | 0.6347 | 0.6492 | 1 |
| Agnews | 768 | 0.5235 | 0.5235 | 0.4441 | OT | 0.4661 | 0.4985 | ERR | 0.5110 | 0.4818 | 0.5242 | 0 |
| Amazon | 768 | 0.5715 | 0.5715 | 0.5608 | OT | 0.4758 | 0.4984 | ERR | 0.5220 | 0.5719 | 0.5731 | 0 |
| Imdb | 768 | 0.5119 | 0.5118 | 0.5178 | OT | 0.4716 | 0.5488 | ERR | 0.5374 | 0.5029 | 0.5121 | 0 |
| Yelp | 768 | 0.6186 | 0.6186 | 0.5638 | OT | 0.4704 | 0.5913 | ERR | 0.5219 | 0.5945 | 0.6190 | 0 |
| CIFAR10 | 512 | 0.7126 | 0.7128 | 0.6685 | OT | 0.7027 | 0.6481 | ERR | 0.4877 | 0.7454 | 0.7131 | 1 |
| FashionMNIST | 512 | 0.7996 | 0.7999 | 0.6227 | OT | 0.8216 | 0.9703 | ERR | 0.5923 | 0.7817 | 0.8002 | 1 |
| MNIST-C | 512 | 0.6441 | 0.6436 | 0.4702 | OT | 0.6288 | 0.6387 | ERR | 0.4320 | 0.6409 | 0.6440 | 1 |
| MVTec-AD | 512 | 0.9722 | 0.9720 | 0.9638 | OT | 0.9234 | 0.7939 | ERR | 0.5365 | 0.9460 | 0.9717 | 0 |
| SVHN | 512 | 0.5357 | 0.5362 | 0.4512 | OT | 0.6008 | 0.5407 | ERR | 0.4338 | 0.5969 | 0.5410 | 1 |
| speech | 400 | 0.5885 | 0.5891 | 0.5854 | OT | 0.5562 | 0.6189 | ERR | 0.3711 | 0.5453 | 0.5950 | 1 |
| musk | 166 | 0.7623 | 0.7597 | 0.1850 | OT | 0.1501 | 0.5693 | ERR | 0.4921 | 0.3524 | 0.7637 | 0 |
| mnist | 100 | 0.7418 | 0.7422 | 0.7218 | OT | 0.6248 | 0.6977 | ERR | 0.5500 | 0.8736 | 0.7430 | 1 |
| optdigits | 64 | 0.6509 | 0.6511 | 0.6553 | OT | 0.4930 | 0.6764 | ERR | ERR | 0.3467 | 0.6522 | 0 |
| SpamBase | 57 | 0.6223 | 0.6233 | 0.2214 | OT | 0.8106 | 0.6552 | ERR | ERR | 0.3679 | 0.6213 | 0 |
| landsat | 36 | 0.6420 | 0.6416 | 0.6065 | OT | 0.6256 | 0.6257 | ERR | ERR | 0.3353 | 0.6429 | 1 |
| satellite | 36 | 0.7318 | 0.7318 | 0.7100 | OT | 0.6295 | 0.7087 | ERR | ERR | 0.7075 | 0.7321 | 1 |
| satimage-2 | 36 | 0.9944 | 0.9944 | 0.9952 | OT | 0.9861 | 0.9923 | ERR | ERR | 0.9978 | 0.9944 | 1 |
| Ionosphere | 33 | 0.8499 | 0.8499 | 0.8148 | OT | 0.8127 | 0.4468 | ERR | ERR | 0.8718 | 0.8497 | 0 |
| WPBC | 33 | 0.4465 | 0.4454 | 0.3652 | OT | 0.4376 | 0.8360 | ERR | ERR | 0.4087 | 0.4465 | 0 |
| letter | 32 | 0.5559 | 0.5558 | 0.5152 | OT | 0.5524 | 0.5355 | ERR | ERR | 0.5205 | 0.5564 | 1 |
| WDBC | 30 | 0.7954 | 0.7944 | 0.7915 | OT | 0.8324 | 0.8211 | ERR | ERR | 0.9445 | 0.7939 | 0 |
| fault | 27 | 0.4025 | 0.4020 | 0.3901 | OT | 0.3885 | 0.3710 | ERR | ERR | 0.3900 | 0.4034 | 1 |
| annthyroid | 21 | 0.7539 | 0.7522 | 0.5694 | 0.7528 | 0.7534 | 0.7704 | ERR | ERR | 0.5686 | 0.7599 | 1 |
| cardio | 21 | 0.7797 | 0.7808 | 0.3937 | OT | 0.7832 | 0.6177 | ERR | ERR | 0.1989 | 0.7837 | 1 |
| Cardiotocography | 21 | 0.6114 | 0.6103 | 0.7371 | OT | 0.6082 | 0.8992 | ERR | ERR | 0.6057 | 0.6097 | 1 |
| Waveform | 21 | 0.8284 | 0.8280 | 0.7212 | OT | 0.8003 | 0.5962 | ERR | ERR | 0.8110 | 0.8330 | 1 |
| Hepatitis | 19 | 0.6568 | 0.6627 | 0.4852 | OT | 0.6627 | 0.6391 | ERR | ERR | 0.4716 | 0.6621 | 0 |
| Lymphography | 18 | 0.9417 | 0.9405 | 0.6964 | OT | 0.9464 | 0.9167 | ERR | ERR | 0.9536 | 0.9423 | 0 |
| pendigits | 16 | 0.8993 | 0.8988 | 0.8096 | OT | 0.9187 | 0.9079 | ERR | ERR | 0.8613 | 0.8999 | 1 |
| wine | 13 | 0.5942 | 0.6083 | 0.7333 | 0.6083 | 0.5458 | 0.5833 | ERR | ERR | 0.6925 | 0.6042 | 0 |
| vowels | 12 | 0.4099 | 0.4005 | 0.3800 | 0.4285 | 0.2643 | 0.3070 | ERR | ERR | 0.4760 | 0.4176 | 0 |
| PageBlocks | 10 | 0.9534 | 0.9537 | 0.9270 | 0.9537 | 0.9559 | 0.9410 | ERR | ERR | 0.9659 | 0.9534 | 1 |
| breastw | 9 | 0.8441 | 0.8452 | 0.1872 | 0.8397 | 0.8876 | 0.8977 | ERR | ERR | 0.4579 | 0.8445 | 0 |
| Stamps | 9 | 0.9580 | 0.9579 | 0.9386 | 0.9579 | 0.9209 | 0.9542 | ERR | ERR | 0.9019 | 0.9579 | 0 |
| WBC | 9 | 0.8058 | 0.8047 | 0.7558 | 0.8023 | 0.8651 | 0.9179 | ERR | ERR | 0.5363 | 0.8044 | 0 |
| Pima | 8 | 0.5686 | 0.5681 | 0.5171 | 0.5710 | 0.5415 | 0.5441 | ERR | ERR | 0.4992 | 0.5701 | 0 |
| yeast | 8 | 0.4574 | 0.4576 | 0.4297 | 0.4631 | 0.4503 | 0.4833 | ERR | ERR | 0.3536 | 0.4589 | 1 |
| thyroid | 6 | 0.7539 | 0.7522 | 0.5694 | 0.7528 | 0.7534 | 0.6180 | ERR | ERR | 0.5686 | 0.7499 | 1 |
| vertebral | 6 | 0.4110 | 0.4111 | 0.4198 | 0.4087 | 0.3746 | 0.3952 | ERR | ERR | 0.3143 | 0.4105 | 0 |
| Wilt | 5 | 0.5192 | 0.5196 | 0.4749 | 0.5196 | 0.5302 | 0.5282 | ERR | ERR | 0.5057 | 0.5191 | 1 |

Table 15: Full table of raw AUCs from Section 5.3 using CBLOF. We denoted time-outs by **OT**, and ODM errors by **ERR**.

| Dataset | Features | Feature Bagging | Full Space | CAE | CLIQUE | HiCS | GMD | PCA | UMAP | ELM | V-GAN | Myopicity |
|---|---|---|---|---|---|---|---|---|---|---|---|---|
| InternetAds | 1555 | 0.7347 | 0.7211 | 0.4141 | OT | ERR | ERR | 0.1804 | 0.3566 | ERR | ERR | 0 |
| 20news | 768 | 0.6779 | 0.6680 | 0.5954 | OT | 0.4792 | 0.6906 | 0.6154 | 0.5232 | 0.6650 | 0.6788 | 1 |
| Agnews | 768 | ERR | 0.7543 | 0.4909 | OT | 0.5193 | 0.5323 | 0.5357 | 0.5312 | ERR | ERR | 0 |
| Amazon | 768 | 0.8843 | 0.8805 | 0.5550 | OT | 0.4940 | 0.5449 | 0.5138 | 0.5161 | ERR | 0.8864 | 0 |
| Imdb | 768 | 0.7984 | 0.7945 | 0.5489 | OT | 0.5211 | 0.5865 | 0.5667 | 0.5151 | ERR | 0.8015 | 0 |
| Yelp | 768 | 0.9767 | 0.9790 | 0.5595 | OT | 0.5265 | 0.6398 | 0.6092 | 0.5349 | ERR | 0.9757 | 0 |
| CIFAR10 | 512 | 0.6997 | 0.6966 | 0.6747 | OT | 0.7061 | ERR | 0.7314 | 0.5709 | ERR | ERR | 1 |
| FashionMNIST | 512 | ERR | 0.5382 | 0.7742 | OT | ERR | ERR | 0.9738 | 0.6374 | ERR | ERR | 1 |
| MNIST-C | 512 | 1.0000 | 1.0000 | 0.7079 | OT | ERR | 0.7391 | 0.7547 | 0.5110 | ERR | 1.0000 | 1 |
| MVTec-AD | 512 | ERR | 0.9491 | 0.9679 | OT | ERR | 0.8407 | 0.8571 | 0.5750 | 0.9588 | ERR | 0 |
| SVHN | 512 | ERR | 0.9230 | ERR | OT | ERR | 0.5935 | 0.6729 | 0.5181 | ERR | ERR | 1 |
| speech | 400 | 0.3935 | 0.4077 | ERR | OT | ERR | ERR | 0.3581 | 0.5406 | ERR | 0.3924 | 1 |
| musk | 166 | 0.7634 | 0.7414 | 0.9234 | OT | 0.9981 | 1.0000 | 0.9998 | 0.3957 | ERR | 0.7628 | 0 |
| mnist | 100 | 0.8194 | 0.8170 | ERR | OT | ERR | 0.9247 | ERR | 0.5780 | ERR | 0.8196 | 1 |
| optdigits | 64 | 0.9989 | 0.9991 | ERR | OT | ERR | 0.8250 | 0.8529 | 0.7542 | ERR | 0.9995 | 0 |
| SpamBase | 57 | 0.9501 | 0.9631 | 0.4160 | OT | 0.6151 | 0.3576 | 0.4038 | 0.5056 | 0.3331 | 0.9486 | 0 |
| landsat | 36 | 0.5294 | 0.5759 | 0.7566 | OT | 0.8156 | 0.7337 | 0.7972 | 0.6178 | 0.7317 | 0.5258 | 1 |
| satellite | 36 | 0.8236 | 0.8009 | 0.8244 | OT | 0.7985 | 0.8072 | 0.8030 | 0.6327 | 0.7855 | 0.8298 | 1 |
| satimage-2 | 36 | 0.9835 | 0.9915 | 0.9988 | OT | 0.9979 | 0.9979 | 0.9987 | 0.7356 | 0.9990 | 0.9797 | 1 |
| Ionosphere | 33 | 0.7743 | 0.7512 | 0.9155 | OT | 0.9434 | 0.5442 | 0.5255 | 0.4682 | 0.9647 | 0.7760 | 0 |
| WPBC | 33 | 0.6590 | 0.6615 | 0.5691 | OT | 0.5423 | 0.9523 | 0.8279 | 0.5247 | 0.4844 | 0.6631 | 0 |
| letter | 32 | 0.6793 | 0.6288 | 0.7445 | OT | 0.8223 | 0.8040 | 0.7145 | 0.5806 | ERR | 0.6990 | 1 |
| WDBC | 30 | 0.7224 | 0.8073 | 0.9266 | OT | 0.9901 | 0.9693 | 0.9859 | 0.6718 | 0.9513 | 0.7761 | 0 |
| fault | 27 | ERR | 0.7940 | 0.7229 | OT | 0.6983 | 0.6748 | 0.7142 | 0.5132 | ERR | 0.7979 | 1 |
| annthyroid | 21 | 0.7414 | 0.7219 | 0.6131 | 0.6682 | 0.6661 | 0.7971 | 0.7800 | 0.4919 | 0.6109 | 0.5769 | 1 |
| cardio | 21 | 0.9689 | 0.9643 | 0.5914 | OT | ERR | 0.6037 | 0.6129 | 0.6238 | ERR | 0.9804 | 1 |
| Cardiotocography | 21 | 0.9773 | 0.9798 | 0.6766 | OT | ERR | 0.8350 | ERR | 0.5623 | 0.6647 | 0.9506 | 1 |
| Waveform | 21 | 0.9317 | 0.8542 | 0.7152 | OT | ERR | 0.7812 | 0.7142 | 0.5700 | ERR | 0.9246 | 1 |
| Hepatitis | 19 | ERR | 0.9374 | 0.5811 | OT | 0.5030 | 0.7675 | 0.5503 | 0.4885 | 0.6166 | 0.8797 | 0 |
| Lymphography | 18 | 0.9710 | 0.9694 | 0.7905 | OT | 0.9583 | 0.9542 | 0.9762 | 0.7280 | 0.9548 | 0.9724 | 0 |
| pendigits | 16 | 0.9139 | 0.9011 | 0.9733 | OT | ERR | 0.9664 | 0.9795 | 0.4131 | ERR | 0.9291 | 1 |
| wine | 13 | 0.9886 | 0.9802 | 0.9342 | 0.9042 | 0.8639 | 0.8996 | 0.8375 | 0.1929 | 0.5398 | 0.9750 | 0 |
| vowels | 12 | 0.7842 | 0.6349 | 0.5752 | ERR | ERR | 0.9353 | 0.9288 | 0.6589 | ERR | 0.7914 | 0 |
| PageBlocks | 10 | 0.5891 | 0.5583 | 0.9484 | 0.9715 | 0.9698 | 0.9712 | 0.9664 | 0.8058 | 0.9725 | 0.5604 | 1 |
| breastw | 9 | 0.5777 | 0.5675 | 0.7112 | 0.9245 | 0.9364 | 0.8826 | 0.7395 | 0.5253 | 0.8548 | 0.5701 | 0 |
| Stamps | 9 | 0.6590 | 0.6615 | 0.9285 | 0.9888 | 0.9649 | 0.9728 | 0.9558 | 0.4891 | 0.9462 | 0.6591 | 0 |
| WBC | 9 | 0.5151 | 0.5254 | 0.5772 | 0.8281 | 0.8756 | 0.9587 | 0.8979 | 0.7153 | 0.7213 | 0.4733 | 0 |
| Pima | 8 | 0.7218 | 0.7151 | 0.4886 | 0.5906 | 0.5766 | 0.5696 | 0.5702 | 0.5443 | 0.5388 | 0.7403 | 0 |
| yeast | 8 | 0.5942 | 0.6083 | 0.5064 | 0.5590 | 0.5670 | 0.5786 | 0.5502 | 0.5429 | | 0.6042 | 1 |
| thyroid | 6 | 0.4465 | 0.4454 | 0.6131 | 0.6687 | 0.6731 | 0.5980 | 0.6129 | 0.6238 | 0.6109 | 0.4465 | 1 |
| vertebral | 6 | 0.4574 | 0.4576 | 0.4322 | 0.5042 | 0.4927 | 0.4702 | 0.5825 | 0.4916 | 0.4952 | 0.4589 | 0 |
| Wilt | 5 | 0.6186 | 0.6186 | 0.5734 | 0.7171 | 0.7651 | 0.6892 | 0.5999 | 0.5858 | 0.7365 | 0.6190 | 1 |

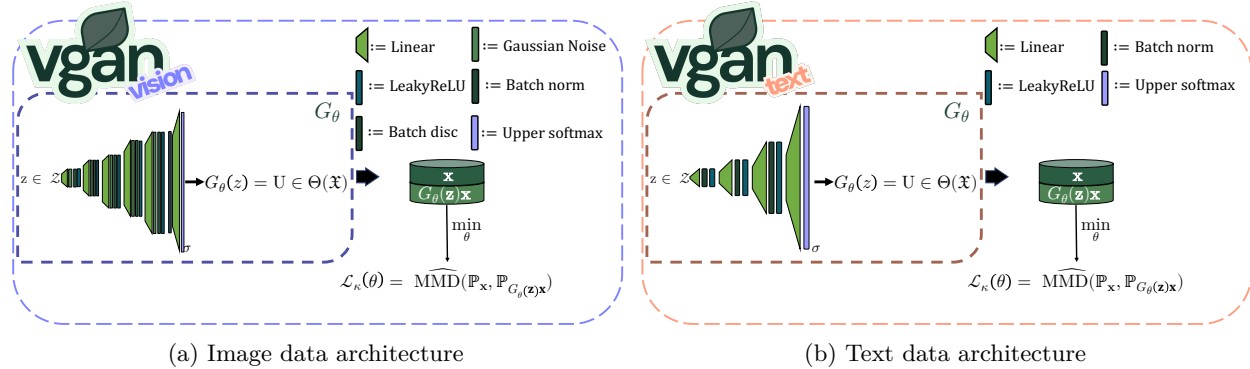

(a) Image data architecture          (b) Text data architecture

Figure 11: Diagram of the network for both image (left) and text (right) data.

what follows. Our goal will be to exploit the human-friendly nature of images and text to visualize what a lens operator looks like in these cases. We include more specific details regarding all implementations and the training for both data types in their respective repositories. To encourage further research on MST, we are releasing all original data, code, and even model weights for all experiments for both data types[10].

### B.2.1 V-GAN vision: Myopic Subspace Theory on Image Data

We will first explore the results of applying **V**-GAN to image data. For this experiment, we defined $E = \mathbb{R}^{3 \cdot h \cdot w}$ and $\Theta(\mathfrak{X}) = Diag_{(h \cdot w) \times (h \cdot w)}(\{0, 1\})$, with $h$ and $w$ being the height and width of an image. Therefore, we are considering $E$ to be the vectorized space of all 3 `RGB` channels of an image, and $\Theta(\mathfrak{X})$ to be the space of $(h \cdot w) \times (h \cdot w)$ binary diagonal matrices acting on random images as:

$$\Theta(\mathfrak{X}) \ni U : \; \mathfrak{X} \longrightarrow \mathfrak{X}$$
$$x \rightsquigarrow Ux = (Ux_{\mathrm{R}} \mid Ux_{\mathrm{G}} \mid Ux_{\mathrm{B}}). \tag{14}$$

In other words, we are applying the operators simultaneously in all color channels. To make **V**-GAN generate such subspaces, the only required change is to alter the head of the network to output a binary vector of the required size. In particular, we changed the layers of **V**-GAN in order to better handle image data. In particular, we employ 5 sequential layers, each consisting of a linear layer for the learning, a Gaussian noise layer and a Batch Normalization layer for regularization, and a Leaky ReLU as an activation. After this, we include a last Batch Discrimination layer and a final linear layer, before passing the output into an upper softmax. The kernel that we use is a Gaussian kernel composed with an ImageNet-pretrained Autoencoder, and we did not employ kernel learning during training. We included a summary in Figure 11. We will study **V**-GAN's lens operators for image data with both real and synthetic data.

**Real Data.** We chose two popular datasets, FashionMNIST and MVTec-AD . The first consists of $28 \times 28$ grey-scaled images of clothes, while the second consists of $900 \times 900$ colored images of different industrial materials. We used the classes `pants` (images of pants) and `bottle` (cross-sections of steel bottles), respectively. Figure 12 contains the results for both FashionMNIST (Figure 12a) and MVTec-AD (Figure 12b). As we can see, the lens operators managed to extract interesting similarity patterns for each class. While in FashionMNIST, operators are mostly pant-shaped, in MVTec-AD, a more complex pattern emerged. Here, the operators selected different ring-shaped parts of the cross-sections rather than chunks of the image. These results further validate our theoretical claims that lens operators can extract complex relations in the data.

**Synthetic Data.** To mirror the methodology in Section 5.2, we generated 10,000 synthetic images ($32 \times 32$ pixels) with half-white/half-black sections. In particular, 5000 images had a white top part and a black bottom part, while the remaining 5000 had the inverse. Logically, as what happened with Section's 5.2

---
[10]Forked from the original repository

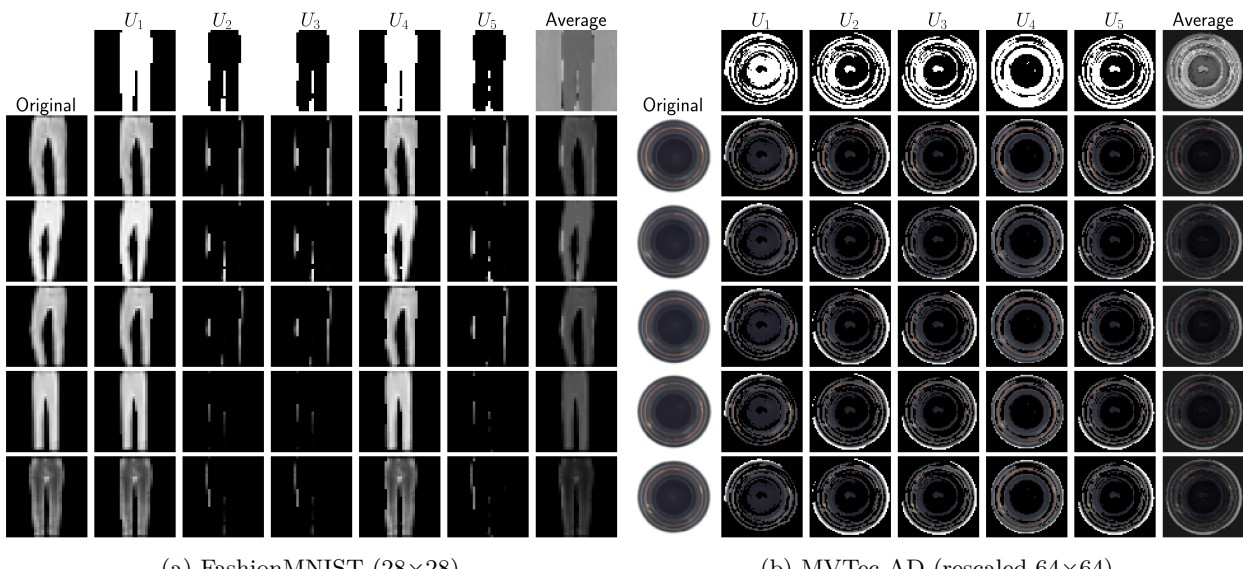

(a) FashionMNIST (28×28)  (b) MVTec-AD (rescaled 64×64)

Figure 12: Visualization of the lens operators obtained using **V**-GAN in FashionMNIST and MVTec-AD. In the top row of both figures, we plotted 5 realizations of **U**, and a final average between 500 samples. On the left-most column of both figures, we plotted 5 non-altered images. In the remaining columns and rows, we plotted each image after applying the corresponding operator in the column.

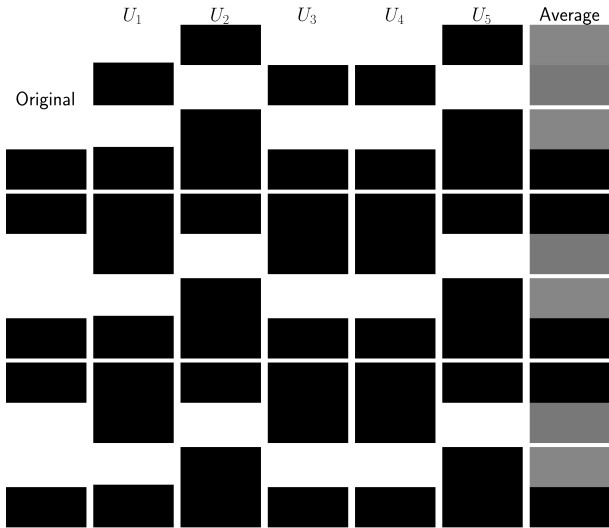

Figure 13: Visualization of the lens operators obtained using **V**-GAN in synthetic data. In the top row of the figure, we plotted 5 realizations of **U**, and a final average between 500 samples. On the left-most column, we plotted 5 non-altered images. In the remaining columns and rows, we plotted each image after applying the corresponding operator in the column.

population, one would expect that a lens operator consists of the two parts with equal probability. As we can see in Figure 13, the derived lens operator was exactly as we expected, further strengthening our derivations.

### B.2.2 V-GAN text: Myopic Subspace Theory on Natural Language

We will try to extract lens operators from the token space of Natural Language data directly. We could do this by considering $E = \mathbb{N}^s$ and $\Theta(\mathfrak{X}) = Diag_{s \times s}(\{0, 1\})$, where $s$ is the predifined sentence length for each

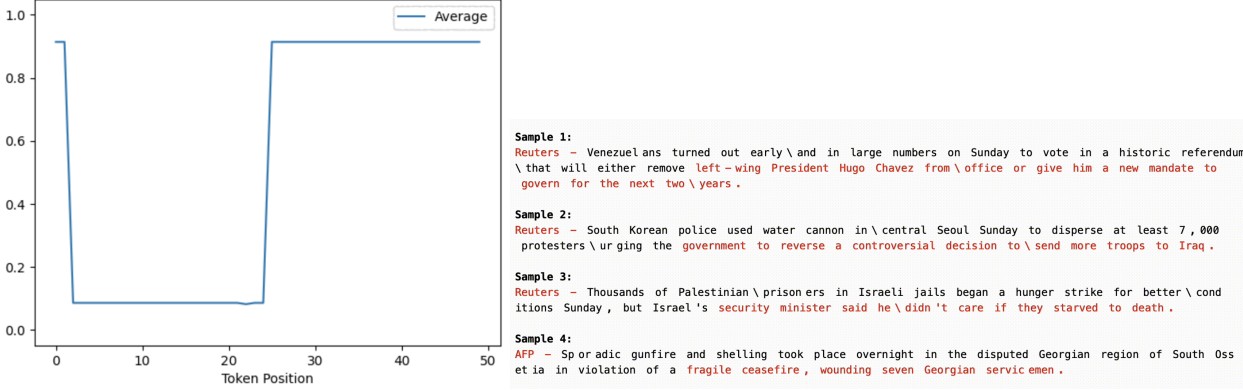

(a) Average selected token by **U** in AGnews    (b) Average selected token in AGnews' text

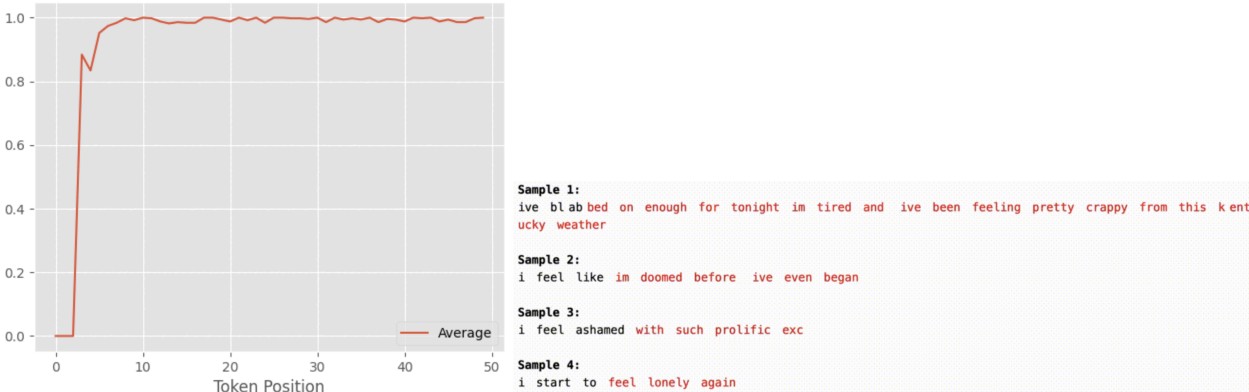

(c) Average selected token by **U** in Emotions    (d) Average selected token in Emotions' text

Figure 14: Visualization of the lens operators obtained using **V**-GAN in AGnews and Emotions. The first row corresponds to AGnews, and the second to Emotions. The left figures contain the average probability of selecting each token using **U**. The right figures contain a visualization of the previous average probability on top of 4 text samples.

dataset. We also utilize a different architecture than before, featuring 3 sequential layers, each consisting of a Linear layer, a leaky ReLU, and a Batch Normalization layer. After them, the outputs are passed into a final Linear layer with an upper softmax output for obtaining the operators. We use a simple Gaussian kernel during training with no kernel learning.

We planned similar experiments utilizing real data for Natural Language. In particular, we employed the AG news dataset and the Emotions dataset. The first one contains descriptions of news articles from different outlets, and the latter contains tweets about sentiments and feelings. Figure 14 contains the results for both.

In AGnews, Figure 14a shows how the derived lens operator **U** selects, on average, the beginning and the end of the sentence. By Figure 14b, this means that **U** on average selects the news outlet — the beginning of each sample — and the core event of the news — the end of each sample. A similar behaviour can be observed for Emotions, where we can see that the network focuses on selecting the end of each tweet, corresponding to the actual feeling — see Figures 14c and 14d. These results motivate the possible use of Myopic Subspace Theory also for Natural Language, further strengthening the applicability of this theory.

