# OpenReview forum: "Adversarial Subspace Generation for Outlier Detection in High-Dimensional Data"
_TMLR — Accepted by TMLR_

### Review · Reviewer_2sjW · 2025-04-30

**Summary Of Contributions:**

This paper proposes a subspace detection method applicable to outlier detection in high-dimensional data. The paper points out that existing subspace detection methods suffer from (i) inaccurate data representation due to dependence on heuristic quality metrics and/or (ii) inefficient computation time due to exponential search spaces. To address these issues, the paper extends existing multiple views theory to propose a generalized myopic subspace theory (MST). MST generalizes existing myopic distribution theory and makes it applicable to a wide range of data types and operators. Specifically, while existing theories require consistency at the probability density function level for returns under random projection operators, MST requires consistency at the probability measure level (Definition 2). This enables the theory to handle data types other than real-valued functions and operators other than projections. Furthermore, under the myopic distribution assumption, the paper formulates subspace detection as a probabilistic optimization problem that minimizes maximum mean discrepancy. Furthermore, the paper proposes V-GAN, a generative network based on MMD-GAN to solve this stochastic optimization problem, i.e., subspace generation. By training V-GAN, we can theoretically find an operator that minimizes MMD. In the experiments, the paper evaluates V-GAN mainly on toy data for sanity checks and real-world tabular datasets. The experiments demonstrate that V-GAN can recover the correct subspace and its weights for Example 1, which existing methods could not recover, and confirm that it achieves the best performance on real-world datasets under the MST assumption. Scalability and applicability to other data types such as images are also verified, supporting the practicality of V-GAN.

**Audience:**

Yes

**Broader Impact Concerns:**

Nothing to report.

**Claims And Evidence:**

Yes

**Requested Changes:**

- Please add the discussion on the interpretability of the proposed method (see W1).
- Please add the discussion on the stability of V-GAN training (see W2).
- Please add the discussion on the limitation of the proposed theory and method (see W3).

**Strengths And Weaknesses:**

### Strengths
+ **S1.**  The paper is well motivated. It carefully discusses the limitations of existing methods and theories, and the proposed theory and methodology are intuitive.
+ **S2.** The approach of extending existing theory (myopic distribution) to construct a theory suitable for real-world subspace detection and formulating subspace detection as a theoretical probabilistic optimization problem is novel and interesting.
+ **S3.** The paper proposes V-GAN, a generative approach for efficiently solving the proposed probabilistic optimization problem. It also proposes practical techniques for training kernels.
+ **S4.** The experiments on toy data and real-world datasets are extensive and directly validate the research questions. In particular, it is fair to provide results not only for datasets where the myopic assumption holds but also for those where it does not.
### Weaknesses
- **W1.** V-GAN generates lens operators, but in cases of high dimensions or complex operators, it may not be easy to interpret what specific data structures or relationships between features the generated operators capture. While axis-parallel projections are relatively easy to interpret, interpretability may become an issue when using more general operators.
- **W2.** Since V-GAN is a GAN-based method, its performance may depend on the settings of many hyperparameters, including network architecture, kernel function selection (whether to use kernel learning, and if so, the kernel type and parameters), optimizer, learning rate, batch size, and number of training epochs. There may be significant costs associated with tuning to find optimal parameter settings. Furthermore, the paper should discuss the instability resulting from min-max optimization in GANs.
- **W3.** MST has been proposed as a general framework, but further discussion about the limitation of this work is needed on specific application methods and effectiveness for tasks other than outlier detection (e.g., clustering, contrastive learning) and more complex data structures (e.g., time series data, graph data).

---

> ### Author Response · Authors · 2025-06-06
> **Authors' rebuttal**
>
> We want to thank the reviewer for their positive review of our paper. All changes made in our paper product of this review are colored $\textcolor{orange}{\text{orange}}$ for better identification. As they asked, we will discuss each of the mentioned weaknesses in order.
>
> ## Weakness 1
> We want to thank the reviewer for their insightful suggestion. We agree with the reviewer’s observation that MST extends existing myopic distribution theory, providing a richer set of tools for identifying subspaces. This indeed can result in space decompositions that are challenging to interpret, especially in high-dimensional settings. However, this is a characteristic of tabular data rather than a limitation of our approach.
>
> We would like to address this by highlighting the following points:
>
> 1. In domains such as image processing or NLP, the results of V-GAN are interpretable. For example, Figures 11 and 12 clearly show that these operators identify important “chunks” of an image. Additionally, the last column of each image illustrates how the average operator highlights the “important” or “relevant” parts.
>
> 2. If interpretability is a priority, one can simplify the operator space to orthogonal projections or potentially penalize the number of derived operators in $\mathbf{U}$’s distribution.
>
> 3. If decomposition quality is more important, $\mathbf{V}$-GAN offers an exceptional performance in practice, as demonstrated in our experiments.
>
>
> We will incorporate this discussion into a new Limitations section.
>
> ## Weakness 2
> Currently, our network is able to converge to (significant) solutions when using a default collection of training parameters across all datasets. While we share the concern with the reviewer, we believe that the issues listed here are not a problem for our setting. This is because our main concern lies in solving Equation (5), to which we have a statistically significant way of analyzing the results (see Section 4.2). As we can see by our experimental results, we have no major problems finding solutions verifying Test (10).
> We believe that an analysis like this could help to find (significant) solutions to Equation (5) in harder datasets or datatypes. Thus, for the reasons listed, we consider such an analysis an important future work that falls outside of the scope of this article. We have included this discussion as future work.
>
> However, we do believe that there exist certain training parameters that could limit our work. Not because of the network training, but because of Test (10). Particularly, Test (10) will heavily depend on the selected kernel for its performance, as mentioned in [20, 21, 22]. Furthermore, while we mentioned in the article that Test (10) is consistent (see section 4.2), its consistency is only asymptotic. This means that the performance of Test (10) depends heavily on the batch size selected — the higher the better. This is not a problem for our particular setting, but it can be one for datasets of much higher dimensionality — like text with a large token count, or high-resolution images — or when trying to use models with a high memory complexity — like large transformers. These two limitations are shared by all MMD-based generative models and will be included in the limitations section.
>
> ## Weakness 3
> We want to thank the reviewer for bringing this out, as it appears it was not sufficiently clear from our text that MST's goal was to be as general as possible in all regards. That is why our theoretical hypothesis for the spaces is broad like that. In essence, as long as there exists a pushforward probability measure $\mathbf{x_*}\mathbb{P}$ with $\mathbf{x}$ a random variable on a separable space $E$, our theory can be applicable.
> For instance, this includes datatypes like Images, text, and even graphs [23]. MST, however, does not consider more general structures like random elements — functional data, Markov processes, etc. — that are of the shape:
>
> $\\begin{align}
>         \\mathbf{x}\_{\\cdot}: \\Omega \\times T & \\longrightarrow E \\\\ \\omega, t & \\leadsto \\mathbf{x}\_t(\\omega)=x\_{t}.
>     \\end{align}$
>
> Analyzing such datatypes requires a suitable extension and thus, it is considered as future work. We have added this discussion to the Limitations section.
>
> Additionally, the reviewer also mentions the application of MST to other problems, like clustering and contrastive learning. We already briefly introduced the applicability of MST to other problems in our previous Section 6. We have expanded this discussion as requested in our new Limitations section.

---

> ### Author Response · Authors · 2025-06-06
> **Authors' rebutal**
>
> ## References
>
> [20] Mikołaj Bińkowski, Danica J. Sutherland, Michael Arbel, and Arthur Gretton. Demystifying MMD GANs. February 2018.
>
> [21] Michael Arbel, Anna Korba, Adil SALIM, and Arthur Gretton. Maximum Mean Discrepancy Gradient Flow. In _Advances in Neural Information Processing Systems_, volume 32. Curran Associates, Inc., 2019.
>
> [22] Arthur Gretton, Karsten M. Borgwardt, Malte J. Rasch, Bernhard Schölkopf, and Alexander Smola. A Kernel Two-Sample Test. _Journal of Machine Learning Research_, 13(25):723–773, 2012.
>
> [23] Mikhail Drobyshevskiy and Denis Turdakov. Random Graph Modeling: A survey of the concepts. _ACM Computing Surveys_, 52(6):1–36, November 2020. arXiv:2403.14415 [cs].

---

### Review · Reviewer_dFsV · 2025-05-17

**Summary Of Contributions:**

This paper proposes subspace generation as a new method to improve outlier detection performance on tabular data. The paper’s contribution is twofold: a theoretical framework and a practical design for subspace generation. The authors point out the problems of existing subspace selection methods, which use some heuristic quality metric, resulting in the loss of the data’s property in the extracted subspace.

To overcome this problem, this paper generalizes an existing theoretical work about the myopic distribution. The existing theory of myopic distribution is limited to some invariance under transformations of a random orthogonal matrix, and the authors generalize this to an invariance under a larger class of operators, which they call a lens operator. Then, the paper shows that a lens operator can be found by solving a stochastic optimization problem.

The paper proposes **V**-GAN to solve the optimization and find the desired lens operator. The authors present the realization of **V**-GAN for a specific case (axis-parallel subspace generation) and demonstrate its performance with a set of experiments. The experiments show that **V**-GAN can successfully extract the lens operator and outperform its competitors.

**Audience:**

Yes

**Broader Impact Concerns:**

I don’t see a particular broader impact concern regarding this paper.

**Claims And Evidence:**

Yes

**Requested Changes:**

1. There are many grammatical mistakes and typos. Please read through the paper once again and fix them.
2. When referring to the Appendix, please include which section is related to the point that refers to it. Also, when referring to tables or figures that are not in the main body of the paper, it is better to clarify the part of the Appendix they are in.
3. The theory part of this paper is quite hard to understand. Adding some illustrative examples for the definitions could make it easier to understand, improving the overall readability of the paper.
4. While the tables with ‘+’ and ‘-’ signs could demonstrate enough progress, it would be better to include the actual p-values (in the Appendix) because some p-values slightly bigger than 0.1 could have been marked as no significant difference.
5. The authors mention that the neural network architecture, the metric space, and the space of operators have to be defined case-by-case. If any guidance or design principles can be provided, this would be appreciated.

**Strengths And Weaknesses:**

### Strength
1. To the best of my knowledge, the proposed idea is novel.
2. The paper contains some theoretical progress, which is rare and valuable in the machine learning community.
3. The experiments contain experiments on a large number of real-world datasets and competitors.
4. The experiments demonstrate remarkable improvements from the baseline.

### Weaknesses
1. In general, the paper’s contents are good enough. Maybe some efforts for better writing could improve the paper’s quality.

---

> ### Author Response · Authors · 2025-06-06
> **Authors' rebuttal**
>
> We want to thank the reviewer for the very positive review and the suggestions to improve the quality of the article. We will comment on them in order of appearance. All changes made in our paper product of this review are colored \textcolor{red}{\text{red}} for better identification.
>
>
> ## R1
> We are very sorry to hear. We have checked again for typos in the revised version of the paper.
>
>
> ## R2
> We thank the reviewer for the suggestion, we added such clarification.
>
>
> ## R3
> We thank the reviewer for their suggestion. We added visualizations of some theoretical aspects that should clarify the theoretical part and the examples. In particular, we added an illustration in Section 3.3 that helps understand the goal of our main Theorem and Corollary. Furthermore, we also added additional illustrations for Example 2 in Appendix A.1 for better understanding how one can measure in MST.
>
>
> ## R4
> We want to thank the reviewer for the suggestion. We have added a table containing the p-values in Tables 7-10 in the appendix.
>
>
> ## R5
> We thank the reviewer for bringing this to our attention. In practice, when using V-GAN, $E$ is the target space to work with, and the operators $\Theta(\mathfrak{X})$ are whatever the network generates. Because of this, if one wants to adapt this theory to its specific use case, one would need to first rewrite their setting into these terms. For instance, we know we can generate different types of operators with generative networks, like rotations [17], swirls [18], and even complex embeddings [19]. An interesting example would be to extend $\mathbf{V}$-GAN to general vectorial subspaces (not only axis-parallel). For this, one could consider the parallel projections of each subspace as the operators, like we did in the newly added Example 2.(b).
> A discussion about this is already included in our remote code repository, and we will include their summary in the paper.
>
> ------------
> ## References
>
> [17] Viktória Pravdová, Lukáš Gajdošech, Hassan Ali, and Viktor Kocur. On Representation of 3D Rotation in the Context of Deep Learning, October 2024. arXiv:2410.10350 [cs].
>
> [18] Sangwook Park, David K. Han, and Hanseok Ko. Sinusoidal wave generating network based on adversarial learning and its application: synthesizing frog sounds for data augmentation, January 2019. arXiv:1901.02050 [cs].
>
> [19] Vinod Kumar Chauhan, Jiandong Zhou, Ping Lu, Soheila Molaei, and David A. Clifton. A Brief Review of Hypernetworks in Deep Learning. _Artificial Intelligence Review_, 57(9):250, August 2024. arXiv:2306.06955 [cs].

---

> > ### Comment · Reviewer_dFsV · 2025-06-14
> >
> > I appreciate the authors' changes.
> >
> > ### Regarding R3
> > One thing I want to clarify is that I'm not asking for an illustration in a Figure, but I'm asking for concrete examples that will help people to understand. I don't think that readers generally have hardship with drawing such an abstract figure in their mind, but if anyone asks to pick an example for the proposed theory, it could be a difficult task in my opinion.
> >
> > Honestly, I don't think that Figure 2 is a change relevant to my comment. If the authors can still update the version, I'd ask the authors to add some examples.

---

> > > ### Author Response · Authors · 2025-06-14
> > >
> > > We want to thank the reviewer for their response.
> > >
> > > We added 5 examples in Example 2 in Appendix A with distributions and their lens operators. We also added an indicator right after Definition 2 stating the existence of Example 2.
> > > Does these examples verifying Definition 2 accomplish the reviewer’s petitions?
> > >
> > > Additionally, we placed Example 2 in the appendix to not break the flow of the text. If deemed better, we could also move it to Section 3 directly.

---

### Review · Reviewer_DTvK · 2025-06-01

**Summary Of Contributions:**

This paper extends theory from prior work on identifying "myopic distributions", which, at a high-level, are probability distributions which remain the same when viewed under the operation of a "lens operator". As an example, if the lens is an orthonormal matrix $U$, then the myopic distribution is supported on a single subspace whose orthonormal projector is $U$. The paper identifies shortcomings of the prior work and outlines an improved theory for general myopic distributions supporting general random lens operators.

The paper then relaxes the general theory to propose practical schemes for identifying lens operators for "almost myopic" distributions, under whose action the distribution remains almost the same (in some notion of distributional distance, the authors use maximum mean discrepancy). Such operators are obtained on real data distributions, and compared to baselines for the task of outlier detection.

**Audience:**

Yes

**Claims And Evidence:**

No

**Requested Changes:**

1. The treatment of related work is incomplete. Specifically, methods in the sub-field of subspace clustering (which the authors group in Subspace Discovery methods in the related work) are not treated sufficiently. The problem of identifying multiple subspaces in data is central to subspace clustering, and several algorithms with theoretical guarantees for success of subspace-identification methods have proposed. Here is a (very small, incomplete) list which the authors can use to find other references:

[1]: Provable Self-Representation Based Outlier Detection in a Union of Subspaces: https://openaccess.thecvf.com/content_cvpr_2017/html/You_Provable_Self-Representation_Based_CVPR_2017_paper.html

[2]: Sparse methods for learning multiple subspaces from large-scale, corrupted and imbalanced data: https://jscholarship.library.jhu.edu/server/api/core/bitstreams/2dfff1da-9ef7-4ba1-9444-a2d7913fee56/content

[3]: Estimation of Subspace Arrangements with Applications in Modeling and Segmenting Mixed Data: https://people.eecs.berkeley.edu/~yima/psfile/Ma-SIREV07.pdf

[4]: Noisy Sparse Subspace Clustering: https://jmlr.org/papers/volume17/13-354/13-354.pdf

---

2. Please add K-subspaces to related work, which seems to be able to do all three ticks in Table 1. Also see:

[5]: k-means projective clustering. https://dl.acm.org/doi/10.1145/1055558.1055581

[6]: Nearest q-Flat to m Points. https://link.springer.com/article/10.1023/A:1004678431677

---

3. What PCA fall into? It does not seem to be subspace search, as the subspace need not be axis-aligned. It does not seem to be subspace discovery because it explicitly outputs a subspace and the projection matrix.

---

4. > no work in the subspace search literature offers a theoretical definition of what an "important" subspace is.

	This requires a rewording with references and a toning down of the contrast: subspace clustering literature, and more generally any clustering work always grapples with the problem that "accurately finding clusters" is an ill-defined problem. The same data distribution can be seen to be having 1 cluster at a coarse level, and 100 clusters at a finer level. Typically one assumes some ideal "good clustering" (e.g., dimension of the subspaces should be small, $\ell_2$ radius of the clusters should be small, etc.), and determines whether the subspace structure was captured according to that model. It is unclear what the authors mean by "accurately preserving data properties" here, and this makes the tick and crosses in the "representation" column of Fig 1 ambiguous.

---

5. > [...] they do not provide a way to project the data [...].

	The reviewer disagrees with this statement. Computing projections is an intermediate step in subspace clustering methods. In fact, spectral clustering methods work precisely because this low dimensional projection contains task-specific structures.

	In its most basic version, the projections of the points are obtained by computing the eigenvectors of the laplacian of a graph whose
	edge-weights are given by the affinity matrix. In spectral clustering, these eigenvectors are then clustered (say, using k-means) to obtain the subspace labels.

	The reviewer encourages the authors to read Section 4.3.2 of Chapter 4 in [Book 1]  (this tutorial [Tutorial 1] also contains similar material in Section 4)

[Book 1]: Generalized Principal Component Analysis: https://link.springer.com/book/10.1007/978-0-387-87811-9

[Tutorial 1]: Spectral Clustering. https://people.csail.mit.edu/dsontag/courses/ml14/notes/Luxburg07_tutorial_spectral_clustering.pdf

---

6. The notion of myopic distributions needs to be contrasted and compared to more general notions of distributions concentrated on lower dimensional subsets of the input space. In addition to references [1-4] above, the authors can find recent theory for such "localized distributions" in:

[7]: Adversarial Examples Might be Avoidable: The Role of Data Concentration in Adversarial Robustness: https://proceedings.neurips.cc/paper_files/paper/2023/hash/92d21245424f3898b7110f555a00e829-Abstract-Conference.html

---

7. Write a few examples of myopic distributions, and precisely specify their random lens operators and their distributions, and what real world settings would one expect to see such distributions.

---

8. Add a discussion on how distributions which only satisfy myopicity approximately differ from truly myopic distributions.

---

9. Once the related work weaknesses above are resolved, it would be important to have a discussion on why subspace clustering methods / subspace detection methods based on sparse representations were not compared to as baselines in Section 5.3.

**Strengths And Weaknesses:**

### Strengths

1. Studying theoretical properties of data distributions that concentrate on lower dimensional manifolds is an important exercise, and new approaches to modeling properties of such distributions is valuable to the community.

2. The present paper comprehensively identifies, and fixes holes in the prior work it extends from, and as such presents a well rounded theory of myopic distributions, and their computational properties.

### Weaknesses

1. The treatment of related work is incomplete, and should be greatly improved. In particular, this reviewer disagrees with some of the statements in the related work section, and some existing methods like K-subspaces are skipped, which seem to satisfy all criteria in Table 1. Details in requested changes (1, 2, 3, 4, 5).

2. There is a need for a lot more contrast of the notion of "myopic distributions" to several notions of "low dimensional distributions" or more generally "localized distributions" available in the literature. Details in requested changes (6).

3. Concrete, mathematically precise examples of a few different myopic distributions is missing. The content is spread out between Example 1, Page 4, and elsewhere. Beyond extremely degenerate examples, this makes it difficult for a reader to have a mental picture of a distribution which (A) admits a lens operator, and (B) is somewhat similar to how a real world distribution would look like. Details in requested changes (7)

4. The experiments, while insightful, seem to be a bit detached from the motivating theory: while myopicity is defined in Defn. 2 to have $P_{x}  = P_{Ux}$, the practical procedures only strive for $ {\rm} dist(P_x, P_{Ux}) $ to be small. While the reviewer understands that developing a theory for such "approximate myopicity" might be outside the scope of this paper, it would be great to know what are the challenges, and more importantly, how does approximate myopicity differ qualitatively from the exact case. Details in requested changes (8)

---

> ### Author Response · Authors · 2025-06-06
> **Authors' rebuttal**
>
> We sincerely appreciate the reviewer’s time and effort in providing feedback on our work. We have addressed all the requested changes and will discuss them in the order they appear. The changes can be categorized into two main areas: (1) revisiting the Related Work section and (2) addressing concerns about the claims of our experiments.
>
> For (1), we understand there was a misconception that all subspace clustering methods are subspace discovery methods, which is not accurate. To clarify, we have expanded our related work section to include all the requested details and citations. We would greatly appreciate the reviewer's feedback about this newer version of Section 2.
>
> For (2), we have added an explanation about hypothesis tests in Section 4.2 to clarify why our results are not “approximately myopic”, a claim we never made in our article.
>
> The changes made in the article are colored $\textcolor{cyan}{\text{blue}}$ for faster identification. We now discuss all of the individual requested changes one by one.
>
> ## R1
> We appreciate the reviewer’s suggestions for additional citations and references. We have reviewed them and incorporated extra citations to clarify our work. As the reviewer will likely agree, subspace clustering is a broad field with a large multitude of different methods.
> Our intention was not to imply that all subspace selection methods used in subspace clustering fall under subspace discovery. We categorized the associated subspace selection process in each subspace clustering algorithm, which can be regarded as subspace search or subspace discovery. In fact, we have categorized several subspace clustering methods, such as [1] and [2], as subspace search. This is a common approach in categorizing related work in subspace selection for outlier detection, as cited by [3] and [4]. We have improved the explanation in the Related Work section to prevent further confusion.
>
> ## R2
> We appreciate the reviewer’s suggestion for our related work. However, [5] and [6] select subspaces based on whether a cluster fits within the subspace, similar to other cited models like [1] and [2]. This means the models rely on the correctness of the cluster assumptions generated by the attached clustering algorithm. For a method like Enclus, this implies that datapoints gather on subspaces with low entropy, which does not align with simple scenarios like the one described in Figure 1.a. Additionally, these two clustering methods require specifying the number of clusters, and thus the subspaces. In Subspace Selection [3, 4], the number of subspaces is assumed unknown, and therefore, the subspace allocation methods associated with these models are not typically considered in the field. We have added in Section 2 a clearer explanation of what constitutes a subspace search and discovery method. We also included the discussion about the Subspace Selection methods considered in our work.
>
> ## R3
> As PCA transforms data from the space of definition of the data to a different singular one (ideally lower-dimensional), it is an embedding model. This was already part of Table 2, but we have now made sure to include extra information in Section 5.1.3 to avoid confusion.
>
> ## R4
> We appreciate the reviewer’s insightful comments and agree with their observation on the challenge of defining a “good cluster.” Our focus, however, is not on how subspace clustering methods form their clusters, but rather on how they select their subspaces. Subspace search models aim to identify subspaces that minimize a specific quality metric. Consequently, these subspaces are preserved in projections only if they minimize this quality metric. Using Enclus as an example, the subspaces assumed to represent the data are those with minimum entropy. We know that this is not always true in practice, as illustrated in Figure 1.a, and it can be difficult to test in practice. In contrast, our measure-theoretic formulation enables transformations that precisely preserve subspaces (see Figures 3 and 12, Example 2 and Definition 2), and there are methods to accurately test whether these assumptions hold in practice (see Section 4.2 and Test (10)). We have included this discussion in our Related Work to more clearly illustrate what we meant by the cited text in the reviewer’s comments.

---

> ### Author Response · Authors · 2025-06-06
> **Authors' rebuttal**
>
> ## R5
> We want to thank the reviewer for bringing this confusion to our attention. We meant that the mentioned subspace clustering methods do not provide a way to project a newly incoming datapoint to any of the derived subspaces. I.e., they do not provide particular projection functions to each subspace. We did not claim that subspace clustering methods do not project data. Furthermore, we do not equate the subspace selection process in all subspace clustering methods with subspace discovery, as suggested by the reviewer. We have cited several subspace clustering methods, such as [1] and [2], that provide a projection function $\pi$ for newly incoming data into each subspace. We are also aware of other methods like [7, 8, 9] that do the same. Since these models minimize a quality metric to define their subspaces, they all fall under subspace search.
>
> As we have hopefully clarified in our responses to the comments above, our goal is to classify subspace selection methods, not subspace clustering methods.
> As subspace clustering methods involve selecting subspaces, we study their selection processes. As mentioned in the article, methods such as [2, 5, 1, 8] minimize a quality metric and explicitly provide the subspace, allowing for the projection of new data. Other methods, like [4], obtain a self-representation of the data, which, as the reviewer noted, projects the clustered data. Specifically, subspace discovery in spectral clustering projects data into the $k$-first eigenspaces of the Laplacian of the similarity matrix. However, standard sparse and spectral clustering methods do not offer a way to project new data [11,12]. There are adaptations for this purpose, such as [11,12,13], but they fall under online/adaptive machine learning, which is beyond our scope. We have incorporated this discussion into the related work section to address any confusion.
>
> ## R6
> We want to thank the reviewer for bringing this research to our attention. Concentration in [14] denotes a property of the measure of subspaces $S$ of the domain $\mathcal{X}$ of a distribution $\mathbb{P}\_\mathbf{x}$. However, our theory does not make any assumptions regarding the concentration of particular regions of $\mathcal{X}$. When considering projections as the operators in $\Theta(\mathfrak{X})$, one can roughly interpret our approach as looking for regions $S_i$ such that $\mathcal{X}=\bigcup_i S_i$. In other words, a myopic distribution $\mathbb{P}_\mathbf{x}$ with lense operator $\mathbf{U}$ can also be concentrated and vice-versa. We have added this discussion to our related work for a more complete analysis.
>
> ## R7
> We want to thank the reviewer for their suggestion. Lens operators can be easily obtained from populations lying on non-parallel intersecting vector spaces by considering their parallel projections. We have included a related example in Example 2 in Appendix A.1 to demonstrate this. In fact, this can also be derived for all manifolds that intersect forming a lower-dimensional submanifold (a problem studied in differential topology, known as transversal intersection of manifolds [15]). This is also added as another example in Example 2.
> Additionally, we now introduce Example 2 after the introduction of Definition 2 in order to help to paint a clearer picture of how a myopic population looks like in practice. Example 2 now contains 5 theoretical examples of different myopic distributions, which we hope will satisfy the reviewer.

---

> ### Author Response · Authors · 2025-06-06
> **Author's rebuttal**
>
> ## R8
> We thank the reviewer for their feedback. We have not studied “approximately myopic” distributions. In our experimental analysis with myopic distributions, we employed operators such that $\text{MMD}(\mathbb{P}\_\mathbf{x}, \mathbb{P}\_\mathbf{Ux}) = 0,$ as our results of Test (10) indicate. If the reviewer was referring to our minimization of the sample MMD, please consider that it is a *sample statistic*. To assess its true value, we employ a statistical test (Test (10)). This test evaluates whether MMD$(\mathbb{P}\_\mathbf{x},\mathbb{P}\_{\mathbf{U}\mathbf{x}} ) = 0$ based on the observed test statistic, $\widehat{\text{MMD}}(\mathbb{P}\_\mathbf{x},\mathbb{P}\_{\mathbf{U}\mathbf{x}})$. As demonstrated in our experimental section, we derived lens operators for 21 datasets using Test (10) in Section 4.2. These lens operators are not “approximately lens”; the test statistic, $\widehat{\text{MMD}}(\mathbb{P}\_\mathbf{x},\mathbb{P}\_{\mathbf{U}\mathbf{x}})$, is so low that our test predicts a $100\%$ chance of ${\text{MMD}}(\mathbb{P}\_\mathbf{x},\mathbb{P}\_{\mathbf{U}\mathbf{x}}) = 0$ (see Table 11). Our prediction of $\mathbf{U}$ being a lens is contingent on the test’s specificity, which is influenced by the chosen kernel $\kappa$. This limitation is common to all MMD-based and kernel-based methods. We have included a discussion on the test in Section 4.2 to prevent potential future confusion and have addressed the limitations of the two-sample test in the limitations section.  Additionally, we have also included further clarifications in Section 5.3.1.
>
> ## R9
> As commented in an earlier reply, and as noted in [16], Sparse Subspace Clustering methods project clustered data using the data’s self-representation. This implies that for new incoming data, the subspace selection part of these methods requires dynamic updates to their subspaces [13]. I.e., when new data arrives, these methods have to update their subspaces. Consequently, no specific projection is available for new data, which makes these models unsuitable for our particular setting. We have expanded on our existing discussion in Section 2 to clarify this confusion.
>
> ------------
> #### References
>
> [1] Chun-Hung Cheng, Ada Waichee Fu, and Yi Zhang. Entropy-based subspace clustering for mining numerical data. In _Proceedings of the fifth ACM SIGKDD international conference on Knowledge discovery and data mining_, pages 84–93, San Diego California USA, August 1999. ACM.
>
> [2] Rakesh Agrawal, Johannes Gehrke, Dimitrios Gunopulos, and Prabhakar Raghavan. Automatic Subspace Clustering of High Dimensional Data. _Data Mining and Knowledge Discovery_, 11(1):5–33, July 2005.
>
> [3] Charu C. Aggarwal and Saket Sathe. _Outlier Ensembles_. Springer International Publishing, Cham, 2017.
>
> [4] Fabian Keller, Emmanuel Muller, and Klemens Bohm. HiCS: High Contrast Subspaces for Density-Based Outlier Ranking. In _2012 IEEE 28th International Conference on Data Engineering_, pages 1037–1048, April 2012. ISSN: 2375-026X.
>
> [5] Dingding Wang, Chris Ding, and Tao Li. K-Subspace Clustering. In Wray Buntine, Marko Grobelnik, Dunja Mladenić, and John Shawe-Taylor, editors, _Machine Learning and Knowledge Discovery in Databases_, pages 506–521, Berlin, Heidelberg, 2009. Springer.
>
> [6] Pankaj K. Agarwal and Nabil H. Mustafa. k-means projective clustering. In _Proceedings of the twenty third ACM SIGMOD-SIGACT-SIGART symposium on Principles of database systems_, PODS ’04, pages 155–165, New York, NY, USA, June 2004. Association for Computing Machinery.
>
> [7] Karin Kailing, Hans-Peter Kriegel, Peer Kröger, and Stefanie Wanka. Ranking Interesting Subspaces for Clustering High Dimensional Data. In Nada Lavrač, Dragan Gamberger, Ljupčo Todorovski, and Hendrik Blockeel, editors, _Knowledge Discovery in Databases: PKDD 2003_, pages 241–252, Berlin, Heidelberg, 2003. Springer.
>
> [8] C. Baumgartner, C. Plant, K. Railing, H.-P. Kriegel, and P. Kroger. Subspace selection for clustering high-dimensional data. In _Fourth IEEE International Conference on Data Mining (ICDM’04)_, pages 11–18, November 2004.
>
> [9] Emmanuel Müller, Ira Assent, Patricia Iglesias, Yvonne Mülle, and Klemens Böhm. Outlier Ranking via Subspace Analysis in Multiple Views of the Data. In _2012 IEEE 12th International Conference on Data Mining_, pages 529–538, December 2012. ISSN: 2374-8486.
>
> [10] Yu-Xiang Wang and Huan Xu. Noisy Sparse Subspace Clustering.
>
> [11] Rocco Langone, Oscar Mauricio Agudelo, Bart De Moor, and Johan A. K. Suykens. Incremental kernel spectral clustering for online learning of non-stationary data. _Neurocomputing_, 139:246–260, September 2014.
>
> [12] Tengteng Kong, Ye Tian, and Hong Shen. A Fast Incremental Spectral Clustering for Large Data Sets. In _2011 12th International Conference on Parallel and Distributed Computing, Applications and Technologies_, pages 1–5, October 2011. ISSN: 2379-5352.

---

> ### Author Response · Authors · 2025-06-06
> **Author's rebuttal**
>
> [13] Jinping Sui, Zhen Liu, Li Liu, Alexander Jung, and Xiang Li. Dynamic Sparse Subspace Clustering for Evolving High-Dimensional Data Streams. _IEEE Transactions on Cybernetics_, 52(6):4173–4186, June 2022.
>
> [14] Ambar Pal, Jeremias Sulam, and Rene Vidal. Adversarial Examples Might be Avoidable: The Role of Data Concentration in Adversarial Robustness. November 2023.
>
> [15] Victor Guillemin and Alan Pollack. _Differential Topology_. American Mathematical Soc., 2010. GoogleBooks-ID: FdRhAQAAQBAJ.
>
> [16] Chong You. _Sparse Methods for Learning Multiple Subspaces from Large-scale, Corrupted and Imbalanced Data_. PhD thesis, The Johns Hopkins University, 2018.

---

> > ### Comment · Reviewer_DTvK · 2025-07-09
> >
> > I appreciate the authors' efforts to include additional examples supporting the core contribution. I think these are very useful, and would encourage the authors to consider moving all these examples to the main paper, though this would need some additional thought as they currently go through Prop. 4, which only appears in the Appendix.
> >
> > Additionally, please run a spelling / grammar check through the text (e.g, there are several instances of Ie vs i.e.).
> >
> > Finally, and less importantly, I still philosophically disagree with the way the literature is presented in Table 1, but will leave it to the broader community to determine its utility. The core contribution of the present paper stands regardless of this Related work table.

---

> > > ### Author Response · Authors · 2025-07-09
> > >
> > > We thank the reviewer for their positive comment. We are pleased to see that we have addressed the issues raised. We will move Example 2 to the main text and will further fix the grammar/spelling errors as requested.

---

### Decision · Action_Editor_zn1V · 2025-07-09

**Recommendation:** Accept as is

**Additional Comments:**

Initially, reviewers raised concerns regarding the related work and limitations of the proposed method. However, through discussion, the authors improved the paper by enhancing their writing. While there is still some room for improvement, as pointed out by the reviewers, I expect these points to be addressed in the authors' final submission.

**Audience:**

Yes

**Audience Explanation:**

Outlier detection is a crucial application in machine learning, and this research will be of interest to a segment of the machine learning community.

**Claims And Evidence:**

Yes

**Claims Explanation:**

In this paper, the authors proposed a subspace-based outlier detection method. Their approach operates under the assumption that the data distribution exhibits a property called myopicity with respect to a certain operator. The authors then introduced V-GAN, a GAN-based technique designed to generate subspaces that are effective for outlier detection.